# An Iterative Algorithm for Differentially Private $k$-PCA with Adaptive Noise

**Johanna Düngler**
Department of Computer Science
University of Copenhagen
jodu@di.ku.dk

**Amartya Sanyal**
Department of Computer Science
University of Copenhagen
amsa@di.ku.dk

## Abstract

Given $n$ i.i.d. random matrices $A_i \in \mathbb{R}^{d \times d}$ that share common expectation $\Sigma$, the objective of *Differentially Private Stochastic PCA* is to identify a subspace of dimension $k$ that captures the largest variance directions of $\Sigma$, while preserving differential privacy (DP) of each individual $A_i$. Existing methods either (i) require the sample size $n$ to scale super-linearly with dimension $d$, even under Gaussian assumptions on the $A_i$, or (ii) introduce excessive noise for DP even when the intrinsic randomness within $A_i$ is small. Liu et al. [2022a] addressed these issues for sub-Gaussian data but only for estimating the top eigenvector ($k = 1$) using their algorithm DP-PCA. We propose the first algorithm capable of estimating the top $k$ eigenvectors for arbitrary $k \leq d$, whilst overcoming both limitations above. For $k = 1$, our algorithm matches the utility guarantees of DP-PCA, achieving near-optimal statistical error even when $n = \tilde{O}(d)$. We further provide a lower bound for general $k > 1$, matching our upper bound up to a factor of $k$, and experimentally demonstrate the advantages of our algorithm over comparable baselines.

## 1 Introduction

Principal Component Analysis (PCA) is a foundational statistical method widely utilized for dimensionality reduction, data visualisation, and noise filtering. Given $n$ data points $\{x_i\}_{i=1}^n$, classical PCA computes the top eigenvectors of the empirical covariance matrix $X := \sum_{i=1}^n x_i x_i^\top \in \mathbb{R}^{d \times d}$. This problem of extracting the top $k$ eigenvectors is commonly known as $k$-PCA. In this work, we consider the problem of Stochastic $k$-PCA, which differs from the standard setting as follows: instead of inputting a single matrix, we input a stream of matrices $A_1, \ldots, A_n$, that are sampled independently from distributions that share the same expectation $\Sigma$. Given this input, the goal of a Stochastic $k$-PCA algorithm is to approximate the dominant $k$ eigenvectors of $\Sigma$.

Differential privacy (DP) [Dwork et al., 2006] provides rigorous, quantifiable guarantees of individual data privacy and has been widely adopted in sensitive data contexts, such as census reporting [Abowd et al., 2020] and large-scale commercial analytics [Apple, 2017]. Despite extensive study of differentially private PCA [Blum et al., 2005, Chaudhuri et al., 2013, Hardt and Roth, 2013, Dwork et al., 2014b], existing methods in the stochastic setting suffer from sample complexity super-linear in $d$ or inject noise at a scale that ignores the underlying stochasticity in the data. When applied to the stochastic setting, these works generally yield suboptimal error rates of $O(\sqrt{dk/n} + d^{3/2}k/(\varepsilon n))$ where $\varepsilon$ is the DP parameter.

**Example 1** (Spiked Covariance). *In the spiked covariance model, we observe i.i.d. matrices $A_i \in \mathbb{R}^{d \times d}$ that contain both a deterministic (low-rank) signal and random noise, causing the $A_i$ to be full-rank. As a concrete illustration, consider data points $x_i = s_i + n_i$, composed of a signal $s_i \sim Unif(\{v, -v\})$ with $v$ a unit vector and $n_i \sim \mathcal{N}(0, \sigma^2 \mathbf{I}_d)$. Therefore $A_i := x_i x_i^\top$ consists of a*

*deterministic part $vv^\top$ and noise terms that scale with $\sigma^2$. One would hope that the privacy noise that is needed, shrinks as the noise variance $\sigma^2$ decreases. Instead, most differentially private PCA methods employ non-adaptive clipping thresholds, so their added privacy noise scales only with that threshold, resulting in unnecessarily large privacy noise for many distributions.*

Recent advances by Liu et al. [2022a] address these limitations for sub-Gaussian distributions, but only for the top eigenvector case ($k = 1$). Cai et al. [2024] achieve optimal performance specifically for the $k$-dimensional spiked covariance model, yet their privacy guarantees only apply under distributional assumptions on the data.

**Our Contributions.** *In this work, we propose $k$-DP-PCA, the first DP algorithm for stochastic PCA that simultaneously (1) achieves sample complexity $n = \tilde{O}(d)$ under similar assumptions as Liu et al. [2021], (2) adapts its privacy noise to the data's inherent randomness, (3) generalizes seamlessly to any target dimension $k \leq d$, and (4) is simple to implement.*

For $k = 1$, $k$-DP-PCA matches the risk of Liu et al. [2022a] under sub-Gaussian assumptions. For general $k$, we prove a nearly matching lower bound up to a linear factor in $k$, precisely characterising the cost of privacy in this general setting. Technically, we employ the *deflation* framework: iteratively estimate the top eigenvector, project it out, and repeat. We extend the recent deflation analysis of Jambulapati et al. [2024] to the stochastic setting via a novel *stochastic e-PCA oracle* (Definition 5), which may be of independent interest. We then adapt DP subroutines from Liu et al. [2022a] based on Oja's algorithm and finally, through a novel utility analysis of non-private Oja's algorithm, demonstrate that the adapted subroutines satisfy the oracle's requirements, yielding a simple to implement, memory-efficient method.

The remainder of this paper is structured as follows. We formally define our setting in Section 2, state main results in Section 3, present technical analyses in Section 4 and empirical evaluations demonstrating the effectiveness of our approach in Section 5. Finally, we end with a discussion and open questions in Section 6 and conclusion in Section 7.

## 2 Problem formulation

Let $A_1, \ldots, A_n \in \mathbb{R}^{d \times d}$ be independent random matrices with common expectation $\Sigma = \mathbb{E}[A_i]$. We assume $\Sigma$ is symmetric positive semi-definite (PSD) with eigenvalues $\lambda_1 \geq \lambda_2 \geq \cdots \geq \lambda_d \geq 0$. For a given $k < d$, we assume the eigengap $\Delta_k = \lambda_k - \lambda_{k+1} > 0$. The goal of *Stochastic PCA* is to produce a $U \in \mathbb{R}^{d \times k}$ whose orthonormal columns approximate the top-$k$ eigenspace of $\Sigma$. We measure the utility of $U$ by comparing it to $V_k$, the matrix containing the true top $k$ eigenvectors of $\Sigma$ as columns. Throughout, $\|\cdot\|_2$ denotes the operator norm, $\langle \cdot, \cdot \rangle$ the Frobenius inner product: $\langle A, B \rangle = \text{Tr}(A^\top B)$, and $\gtrsim$ and $\tilde{O}(\cdot)$ hide polylogarithmic factors.

**Definition 1** ($\zeta$-approximate Utility)**.** We say $U \in \mathbb{R}^{d \times k}$ is $\zeta$-approximate if $U$ has orthonormal columns and

$$\langle UU^\top, \Sigma \rangle \geq (1 - \zeta^2)\langle V_k V_k^\top, \Sigma \rangle.$$

Although several utility measures exist for PCA, our choice is motivated by the error measure used in Jambulapati et al. [2024]. This is a natural measure of usefulness, as $\langle UU^\top, \Sigma \rangle$ quantifies how much of the original "energy" of $\Sigma$ is retained when projecting onto the lower-dimensional subspace spanned by $U$, and by the Eckart-Young Theorem we know $V_k$ is the optimal rank-$k$ approximation of $\Sigma$.

Further, we use the add/remove model of differential privacy, namely

**Definition 2** (Differential Privacy ([Dwork et al., 2006]))**.** Given two multi-sets $S$ and $S'$, we say the pair $(S, S')$ is neighboring if $|S \setminus S'| + |S' \setminus S| \leq 1$. We say a stochastic query $q$ over a dataset $S$ satisfies $(\varepsilon, \delta)$-differential privacy for some $\varepsilon > 0$ and $\delta \in (0, 1)$ if

$$P(q(S) \in A) \leq e^\varepsilon P(q(S') \in A) + \delta$$

for all neighboring $(S, S')$ and all subsets $A$ of the range of $q$.

Before discussing the main results of our work, we first formalize the assumptions on the data in Assumption A. Note that Assumption A is only required for our utility guarantee and is not necessary for the privacy guarantee.

**Assumption A** $((\Sigma, \{\lambda_i\}_{i=1}^d, M, V, K, \kappa, a, \gamma^2)$-model)**.** *Let $A_1, \ldots, A_n \in \mathbb{R}^{d \times d}$ be sampled independently from distributions satisfying:*

**A.1** $\mathbb{E}[A_i] = \Sigma$, *where $\Sigma$ is PSD with eigenvalues $\lambda_1 \geq \cdots \geq \lambda_d \geq 0$, corresponding eigenvectors $v_1, \ldots, v_d$, $0 < \Delta = \min_{i \in [k]} \Delta_k$ and $\kappa' := \frac{\lambda_1}{\Delta}$.*

**A.2** $\|A_i - \Sigma\|_2 \leq \lambda_1 M$ *almost surely.*

**A.3** $\max\left\{\left\|\mathbb{E}\left[(A_i - \Sigma)(A_i - \Sigma)^\top\right]\right\|_2, \left\|\mathbb{E}\left[(A_i - \Sigma)^\top(A_i - \Sigma)\right]\right\|_2\right\} \leq \lambda_1^2 V.$

**A.4** *For all unit vectors $u, v$ and projection matrices $P$,*

$$\mathbb{E}\left[\exp\left(\left(\frac{|u^\top P(A_i - \Sigma)Pv|^2}{K^2 \lambda_1^2 \gamma^2}\right)^{1/2a}\right)\right] \leq 1.$$

*Define $H_u = \frac{1}{\lambda_1^2}\mathbb{E}[(A_i - \Sigma)u\, u^\top(A_i - \Sigma)^\top]$ and $\gamma^2 = \max_{\|u\|=1} \|H_u\|_2$.*

Assumptions A.1 to A.3 are standard for matrix concentration (e.g., under the matrix Bernstein inequality [Tropp, 2012]) and thus also required for the utility guarantees of Oja's algorithm even in the non-private setting. Assumption A.4 guarantees that for any unit vectors $u, v$, and projection $P$

$$|u^\top P(A_i - \Sigma)Pv|^2 \leq K^2 \lambda_1^2 \gamma^2 \log^{2a}(1/\vartheta)$$

with probability $1 - \vartheta$, for some sufficiently large constant $K$. This bound, which controls the size of the bilinear form, can be seen as a Gaussian-like tail bound, which tells us that the magnitude of the projection of the $A_i$ along any direction is bounded with high probability. It is an extension of the assumptions in [Liu et al., 2022a] to the higher dimensional case. Distributions that fulfill this assumption include bounded matrices and (sub-)gaussian outer product matrices:

**Example 2** (Gaussian Data, Remark 3.4 in Liu et al. [2022a])**.** *Let $A_i = x_i x_i^\top$ with $x_i \sim \mathcal{N}(0, \Sigma)$, then comparing to Assumption A we have that $M = O(d \log(n))$, $V = O(d)$, $K = 4$, $a = 1$, and $\gamma^2 = O(1)$*

Distributions that violate assumption 4 include heavy-tailed outer products, for example $r \sim$ Pareto$(\alpha)$, $x = ru$, $A_i = xx^\top$, or mixtures with rare but huge spikes:

**Example 3.** *Let $A_i = x_i x_i^\top$, with $x_i$ be sampled as follows:*

$$x_i = \begin{cases} x \sim \mathcal{N}(0, \mathbf{I}_d) & w.p.\ 1 - \alpha \\ x \sim \mathrm{Unif}\{\alpha^{-1/4}\mathrm{v}, -\alpha^{-1/4}\mathrm{v}\} & w.p.\ \alpha \end{cases}$$

*where $v$ is a unit vector and $0 < \alpha < 1$. Then the mean of this distribution is 0 and its covariance is $\Sigma = (1 - \alpha)\mathbf{I}_d + \sqrt{\alpha}vv^\top$. So for $u = v$ and $P = \mathbb{I}_d$, if $x = \pm\alpha^{-1/4}v$*

$$u^\top(A_i - \Sigma)u = v^\top(x_i x_i^\top - \Sigma)v = (v^\top x_i)^2 - v^\top\Sigma v$$
$$= \alpha^{-1/2} - v^\top\Sigma v = \alpha^{-1/2} - (1 - \alpha) + \sqrt{\alpha} \simeq \alpha^{-1/2}$$

*and for $\alpha \to 0$ this term blows up, so for any fixed $K, \lambda_1, \gamma$ the overall expectation will exceed 1, and hence violate Assumption A.4.*

## 3 Main Results

In this section, we first discuss our main proposed algorithm in Section 3.1. In Section 3.2 we then discuss our main upper bounds and complement that with lower bounds in Section 3.3

### 3.1 Our Algorithm

Our first proposed algorithm k-DP-PCA, defined in Algorithm 1, follows a classical deflation [Jambulapati et al., 2024] approach. The algorithm proceeds in $k$ rounds and in each of the $k$ rounds it invokes the sub-routine MODIFIEDDP-PCA (Line 3), to identify the current top eigenvector. Then, the algorithm removes its contribution by projecting out the direction of the eigenvector from the remaining data (Line 4), on which it carries out the next round.

---

**Algorithm 1** $k$-DP-PCA

---

**Input:** $\{A_1, \ldots, A_n\}$, $k \in [d]$ , privacy parameters $(\varepsilon, \delta)$, $B \in \mathbb{Z}_+$, learning rates $\{\eta_t\}_{t=1}^{\lfloor n/B \rfloor}$, and $\tau \in (0, 1)$

1: $m \leftarrow n/k$, $P_0 \leftarrow \mathbf{I}_d$
2: **for** $i \in [k]$ **do**
3: $\quad u_i \leftarrow \text{MODIFIED}\text{DP-PCA}(\{A_{m \cdot (i-1)+j}\}_{j=1}^m, P_{i-1}, (\varepsilon, \delta), B, \{\eta_t\}, \tau)$
4: $\quad\quad P_i \leftarrow P_{i-1} - u_i u_i^\top$
5: **end for**
6: **return** $U \leftarrow \{u_i\}_{i \in [k]}$

---

The MODIFIEDDP-PCA subroutine (Algorithm 2) itself is based on Oja's streaming Algorithm [Jain et al., 2016], but importantly replaces the vanilla gradient update in Oja's algorithm $\omega_T \leftarrow \omega_{t-1} + \eta_t A_{t-1}\omega_{t-1}$, with a two-stage algorithm: first, Line 3 privately estimates the range of a batch of $\{A_i \omega_{t-1}\}$, then Line 4 leverages that range to calibrate the added noise to privately compute the batch's mean. By tailoring the noise scale to the empirical spread of the data, we inject significantly less (privacy) noise whenever the batch concentrates tightly around its mean. Thanks to those additional steps the algorithm enjoys certain statistical benefits as discussed in the paragraph below Corollary 2.

Nevertheless, it is possible to replace the MODIFIEDDP-PCA subroutine with other simpler subroutines that can privately estimate the top eigenvector. We present one such algorithm in Algorithm 3. In Section 5, we present simulations with both of these algorithms highlighting their respective advantages.

---

**Algorithm 2** ModifiedDP-PCA

---

**Input:** $\{A_1, \ldots, A_m\}$, a projection $P$, privacy parameters $(\varepsilon, \delta)$, learning rates $\{\eta_t\}_{t=1}^{\lfloor n/B \rfloor}$, $B \in \mathbb{Z}_+$ and $\tau \in (0, 1)$

1: Choose $\omega_0'$ uniformly at random from the unit sphere, $\omega_0 \leftarrow P\omega_0'/\|P\omega_0'\|$
2: **for** $t = 1, 2, \ldots, T = \lfloor m/B \rfloor$ **do**
3: $\quad \hat{\Lambda} \leftarrow \text{PRIVRANGE}\left(\{PA_{B(t-1)+i}P\omega_{t-1}\}_{i=1}^{\lfloor B/2 \rfloor}, (\varepsilon/2, \delta/2), \tau/(2T)\right)$ (Algorithm 6)
4: $\quad \hat{g}_t \leftarrow \text{PRIVMEAN}\left(\{PA_{B(t-1)+i}P\omega_{t-1}\}_{i=1}^{\lfloor B/2 \rfloor}, \hat{\Lambda}, (\varepsilon/2, \delta/2), \tau/(2T)\right)$ (Algorithm 7)
5: $\quad \omega_t' \leftarrow \omega_{t-1} + \eta_t P\hat{g}_t$
6: $\quad \omega_t \leftarrow P\omega_t'/\|P\omega_t'\|$
7: **end for**
8: **return** $\omega_T$

---

### 3.2 Upper Bound

We now state the main privacy and utility guarantees of k-DP-PCA (Algorithm 1).

**Theorem 1** (Main Theorem). *Let $\varepsilon, \delta \in (0, 0.9)$ and $1 \le k < d$. Then k-DP-PCA satisfies the following:*

**Privacy:** *For any input sequence $\{A_i \in \mathbb{R}^{d \times d}\}$, the algorithm is $(\varepsilon, \delta)$-differentially private.*

**Utility:** *Suppose $A_1, \ldots, A_n$ are i.i.d. satisfying Assumption A with parameters $(\Sigma, M, V, K, \kappa', a, \gamma^2)$. If*

$$
n \gtrsim C \max \begin{cases} e^{\kappa'^2} + \dfrac{d\,\kappa'\,\gamma\,\sqrt{\ln(1/\delta)}}{\varepsilon} + \kappa' M + \kappa'^2 V + \dfrac{\sqrt{d}\,(\ln(1/\delta))^{3/2}}{\varepsilon}, \\[2ex] \lambda_1^2\,\kappa'^2\,k^3\,V, \\[2ex] \dfrac{\kappa'^2\,\gamma\,k^2\,d\,\sqrt{\ln(1/\delta)}}{\varepsilon} \end{cases} , \tag{1}
$$

*for a sufficiently large constant $C$, then with probability at least $0.99$, the output $U \in \mathbb{R}^{d \times k}$ is $\zeta$–approximate with*

$$\zeta = \tilde{O}\left(\kappa'\left(\sqrt{\frac{Vk}{n}} + \frac{\gamma dk\sqrt{\log(1/\delta)}}{\varepsilon n}\right)\right), \tag{2}$$

where $\tilde{O}(\cdot)$ hides factors polylogarithmic in $n, d, 1/\varepsilon, \ln(1/\delta)$ and polynomial in $K$.

*Remark.* The proof of our main Theorem can be found in Appendix E. For $k = 1$, Theorem 1 recovers the bound of Liu et al. [2022a] for DP-PCA. Moreover, the linear dependence on $d$ in $\zeta$ matches the lower bound in Liu et al. [2022a]. On the other hand, the additional linear factor in $k$ may be an artifact of our analysis: if one could reuse samples across deflation steps, this factor could potentially be improved. Further, in $\zeta$, the first term $\sqrt{Vk/n}$ is the non-private statistical error of PCA, while the second term $(\gamma dk\sqrt{\ln(1/\delta)})/(\varepsilon n)$ is the cost of privacy. Lastly, the sample-size condition (1) arises because (i) each batch must be large enough to accurately estimate the range in PRIVRANGE in Algorithm 2, and (ii) errors accumulate across the $k$ deflation steps (Line 4).

As a direct consequence of applying Theorem 1 to Examples 1 and 2, we obtain the following Corollaries:

**Corollary 1** (Upper bound, Gaussian distribution). *Under the same setting as Theorem 1, let $A_i = x_i x_i^\top$ with $x_i \sim \mathcal{N}(0, \Sigma)$. Then with high probability the output is $\zeta$-approximate with*

$$\zeta = \tilde{O}\left(\kappa'\left(\sqrt{\frac{dk}{n}} + \frac{dk\sqrt{\log(1/\delta)}}{\varepsilon n}\right)\right)$$

where $\tilde{O}(\cdot)$ hides poly-logarithmic factors in $n, d, 1/\varepsilon$, and $\log(1/\delta)$.

**Corollary 2** (Upper bound, Spiked Covariance). *If $A_i$ follows the spiked covariance model from Example 1, then $V = O(\sigma^2 d)$, $\gamma^2 = \sigma^2$, and $K = 1$. Hence, with high probability the output is $\zeta$-approximate with*

$$\zeta = \tilde{O}\left(\sigma \cdot \kappa'\left(\sqrt{\frac{dk}{n}} + \frac{dk\sqrt{\log(1/\delta)}}{\varepsilon n}\right)\right) \tag{3}$$

**Adaptive noise**: Our algorithm's advantage is most pronounced when $\gamma$ and $V$ grow with the data randomness, as in Corollary 2. Since for $\zeta = \tilde{O}\left(\sigma\kappa'\left(\sqrt{dk/n} + (dk\sqrt{\ln(1/\delta)})/(\varepsilon n)\right)\right)$, the approximation error decreases as the noise standard deviation $\sigma$ shrinks. Moreover, by comparison with Corollary 3, this bound is tight up to a factor of $k$.

### 3.3 Lower Bounds

In this section, we derive an information-theoretic lower bound for differentially private PCA under our setting. Formal proofs can be found in Appendix F.1. Recall that our utility metric $\zeta$ defined in Definition 1 measures the *relative* loss in captured variance compared to the optimal top-$k$ subspace of $\Sigma$. By contrast, most classical lower bounds for PCA (e.g., Cai et al. [2024], Liu et al. [2022a]) quantify error in terms of the squared Frobenius norm $\|\tilde{U}\tilde{U}^\top - V_k V_k^\top\|_F^2$. These two measures are fundamentally different: the ratio of captured variance directly reflects variance explained in $\Sigma$, whereas the Frobenius-norm loss measures subspace distance without respecting the eigenvalue gaps in $\Sigma$. To connect them, we first establish:

**Lemma 1** (Reduction to Frobenius norm). *Let $\Sigma$ be a PSD $d \times d$ matrix with top-$k$ eigenvectors $V_k \in \mathbb{R}^{d \times k}$ and eigenvalues $\lambda_1 \geq \cdots \geq \lambda_d$. Any $U \in \mathbb{R}^{d \times k}$ that satisfies $\|UU^\top - V_k V_k^\top\|_F^2 \geq \gamma$, must incur*

$$\zeta^2 \geq \frac{\gamma \Delta_k}{2\sum_{i=1}^k \lambda_i}$$

where $\Delta_k := \lambda_k - \lambda_{k+1}$.

Note that if all eigenvalues of $\Sigma$ are equal, every subspace captures the same variance so $\zeta = 0$ for any estimate, yet two such subspaces can be far apart in Frobenius norm. This gap in sensitivity to eigengaps is precisely why our reduction from Frobenius error to $\zeta$ incurs a factor of $\Delta_k$. With this reduction in hand, we prove the spiked-covariance lower bound by invoking standard Frobenius-norm minimax rates [Cai et al., 2024] for differentially private PCA in the spiked covariance model.

**Corollary 3** (Lower bound, Spiked Covariance). *Let the $d \times n$ data matrix $X$ have i.i.d. columns samples from a distribution $P = \mathcal{N}(0, U^\top \Lambda U^\top + \sigma^2 \mathbf{I}_d) \in \mathcal{P}(\lambda, \sigma^2)$ where $\mathcal{P}(\lambda, \sigma^2) = \{\mathcal{N}(0, \Sigma), \Sigma = U\Lambda U^\top + \sigma^2 \mathbf{I}_d, c\lambda \le \lambda_k \le \cdots \le \lambda_1 \le C\lambda\}$. Suppose $\lambda \le c_0' \exp\{e\varepsilon - c_0(\varepsilon\sqrt{ndk} + dk)\}$ for some small constants $c_0, c_0' > 0$. Then, there exists an absolute constant $c_1 > 0$ such that*

$$\inf_{\tilde{U} \in \mathcal{U}_{\varepsilon,\delta}} \sup_{P \in \mathcal{P}(\lambda, \sigma^2)} \mathbb{E}[\zeta] \ge c_1 \left( \left( \frac{\sigma\sqrt{\lambda_1 + \sigma^2}}{\sum_{i=1}^k (\lambda_i + \sigma^2)} \right) \left( \sqrt{\frac{dk}{n}} + \frac{dk}{n\varepsilon} \right) \bigwedge 1 \right).$$

Comparing to our upper bound (Corollary 2), we see matching dependence on $\sigma$, $d$, $n$, and $\varepsilon$, up to a multiplicative factor of $k$, $\sqrt{\lambda_1 + \sigma^2}$, and $\sqrt{\log(1/\delta)}$. The gap in $k$ arises from our sequential deflation approach, which currently requires independent batches at each step. Reusing samples across rounds could remove this up to a $\sqrt{k}$ factor [1].

**Special case $k = 1$.** When $k = 1$, k-DP-PCA reduces exactly to MODIFIEDDP-PCA. Theorem 9 guarantees that the sine of the angle between the privately estimated eigenvector of MODIFIEDDP-PCA and the true top eigenvector is small, which is equivalent to being close in the Frobenius norm. This matches the upper bound of Liu et al. [2022a] and thus also the lower bound up to a factor of $\log(1/\delta)$ (restated in Theorem 11 in the Appendix).

## 4 Technical Results

We now sketch the proof of Theorem 1 by first proving a more general "meta-theorem" that applies to any *stochastic ePCA oracle* (defined below in Definition 5). At a high level, k-DP-PCA uses the classical deflation strategy: 1. Extract the top eigenvector of the current residual using a 1-PCA subroutine. 2. Project this vector out of the data. 3. Repeat until $k$ components are obtained. In Theorem 1 we implement the 1-PCA step with MODIFIEDDP-PCA, but the same proof carries through for any algorithm satisfying the following guarantee.

**Definition 3** (stochastic ePCA oracle). An algorithm $O_{\text{ePCA}}$ is a $\zeta$–*approximate 1-ePCA oracle* if the following holds. On independent inputs $A_1, \ldots, A_n \in \mathbb{R}^{d \times d}$ with $\mathbb{E}[A_i] = \Sigma \in \mathbb{S}_{\succeq 0}^{d \times d}$ for all $i$ and any orthogonal projector $P \in \mathbb{R}^{d \times d}$, $O_{\text{ePCA}}$ returns a unit vector $u \in \text{Im}(P)$ such that, with high probability,

$$\langle uu^\top, P\Sigma P \rangle \ge (1 - \zeta^2)\langle vv^\top, P\Sigma P \rangle$$

where $v$ is the top eigenvector of the projected matrix $P\Sigma P$.

This notion was inspired by Jambulapati et al. [2024], who analyzed deflation in the non-stochastic setting. Their results do not extend the stochastic setting that we explore here.

**Theorem 2** (Meta Theorem). *Let $\Sigma \in \mathbb{S}_{\succeq 0}^{d \times d}$ and $A_1, \ldots, A_n$ be $n$ i.i.d. samples with $\mathbb{E}[A_i] = \Sigma$. Suppose we replace each 1-PCA step in Line 3 of Algorithm 1 by a $\zeta$–approximate stochastic ePCA oracle $O_{1\text{PCA}}$. Then the deflation algorithm outputs $U \in \mathbb{R}^{d \times k}$ satisfying*

$$\langle UU^\top, \Sigma \rangle \ge (1 - \zeta^2)\|\Sigma\|_k.$$

*Further, for any $\varepsilon > 0, \delta \in (0, 1)$, if $O_{1PCA}$ is $\varepsilon, \delta$-DP then the entire algorithm remains $(\varepsilon, \delta)$-DP.*

*Remark.* This Theorem is a consequence of the stochastic deflation method we prove in Appendix C and Parallel Composition (Lemma 15).

One important thing we would like to highlight in this section is that this proof strategy is not unique to MODIFIEDDP-PCA. In fact, our novel analysis of non-private Oja's algorithm (Theorem 7) shows that Algorithm 3 is also a stochastic ePCA oracle. We highlight the two results below.

**Theorem 3.** *Given $A_1, \ldots, A_n$ are i.i.d. and satisfy Assumption A, MODIFIEDDP-PCA and DP-Ojas as defined Algorithms 2 and 3 are stochastic ePCA oracles with $\zeta = \tilde{O}\left( \kappa' \left( \sqrt{\frac{V}{n}} + \frac{\gamma d\sqrt{\log(1/\delta)}}{\varepsilon n} \right) \right)$ and $\zeta = \tilde{O}\left( \kappa' \left( \sqrt{\frac{V}{n}} + \frac{(\gamma+1)d\sqrt{\log(1/\delta)}}{\varepsilon n} \right) \right)$ respectively.*

---

[1] Reusing will allow us to use all $n$ samples every round (instead of $n/k$), however we will incur an additional $\sqrt{k}$ factor due to privacy composition, which is why it will only lead to a total improvement of $\sqrt{k}$ and not $k$.

---

**Algorithm 3** DP-Ojas

---

**Input:** $\{A_1, \ldots, A_m\}$, a projection $P$, privacy parameters $(\varepsilon, \delta)$, learning rates $\{\eta_t\}_{t=1}^{\lfloor m \rfloor}$
 1: Set DP noise multiplier: $\alpha \leftarrow C' \log(n/\delta)/(\varepsilon \sqrt{n})$
 2: Set clipping threshold: $\beta \leftarrow C\lambda_1 \sqrt{d}(K\gamma \log^a(nd/\zeta) + 1)$
 3: Choose $\omega_0'$ uniformly at random from the unit sphere, $\omega_0 \leftarrow P\omega_0'/\|P\omega_0'\|$
 4: **for** $t = 1, 2, \ldots, m$ **do**
 5:     Sample $z_t \sim \mathcal{N}(0, \mathbf{I}_d)$
 6:     $\omega_t' \leftarrow \omega_{t-1} + \eta_t P \left( \text{clip}_\beta(PA_tP\omega_{t-1}) + 2\beta\alpha z_t \right)$
 7:     $\omega_t \leftarrow P\omega_t'/\|P\omega_t'\|$
 8: **end for**
 9: **return** $\omega_T$
where $\text{clip}_\beta(x) = x \cdot \min\{1, \frac{\beta}{\|x\|_2}\}$

---

*Remark.* In Appendix E, we establish that both MODIFIEDDP-PCA and k-DP-Ojas are valid ePCA oracles, with each result stated and proved as a separate theorem.

Note that we cannot plug in the DP-PCA algorithm of Liu et al. [2022a] in Theorem 2, since it only guarantees relative error on $\mathbb{E}[P]\Sigma\mathbb{E}[P]$:

$$\langle uu^\top, \mathbb{E}[P]\Sigma\mathbb{E}[P]\rangle \geq (1-\zeta)\langle vv^\top, \mathbb{E}[P]\Sigma\mathbb{E}[P]\rangle,$$

rather than on $P\Sigma P$, and $\mathbb{E}[P]$ need not be a projection matrix.

The proof of Theorem 3 follows directly from the utility proof of MODIFIEDDP-PCA (Theorem 9) and of DP-Ojas (Theorem 10). Combining this with Theorem 2 immediately gives us Theorem 1 and the following Corollary 4.

To proof the utility of MODIFIEDDP-PCA we proceed in three steps: 1. Prove non-private Oja's algorithm is a stochastic ePCA oracle via a Novel analysis in Appendix D 2. Show that with high probability, the update step (Line 5 in Algorithm 2) can be reduced to an update step of non-private Oja's algorithm with matrices $PC_tP$, where $C_t := \frac{1}{B}\sum_{i\in[B]} A_i + \beta_t G_t$ and $G_t$ is a scaled Gaussian matrix. 3. Bound the accumulated projection error across deflation steps (Lemma 23). Importantly, a similar argument also shows that DP-Ojas Algorithm 3 satisfies the same property with a slightly differently $\zeta$.

**Corollary 4** ($k$-DP-Ojas). *Under Assumption A, if $n$ is sufficiently large then using Algorithm 3 in each 1-PCA step returns $U \in \mathbb{R}^{d\times k}$ that is $\zeta$–approximate with*

$$\zeta = \tilde{O}\left( \frac{\lambda_1}{\Delta} \left( \sqrt{\frac{Vk}{n}} + \frac{(\gamma+1)dk\log(1/\delta)}{\varepsilon n} \right) \right)$$

*hiding poly-logarithmic factors in $n, d, 1/\varepsilon, \ln(1/\delta)$ and polynomial factors in $K$.*

*Remark.* This Corollary follows directly from Theorem 2 together with Theorem 3.

When comparing the utility bounds of MODIFIEDDP-PCA and k-DP-Ojas the difference is particularly apparent when considering Example 1, as for k-DP-Ojas when $\sigma \to 0$ the bound becomes $\tilde{O}\left(\frac{dk\log(1/\delta)}{\varepsilon n}\right)$, as due to the second term of the utility bound containing the multiplicative factor of $(\gamma + 1)$ (as opposed $\gamma$ as in MODIFIEDDP-PCA) it does not vanish. Therefore in the low-noise cases MODIFIEDDP-PCA will outperform k-DP-Ojas. However, for other cases such as (sub-)Gaussian data we expect them to perform similarly. In those cases it can be preferential to use k-DP-Ojas as due to its simplicity it requires less hyperparamters to be set and is more stable to changes in learning rates.

## 5   Experiments

In our experiments, we compare k-DP-PCA and k-DP-Ojas against two modified versions of the DP-Gauss algorithms of Dwork et al. [2014b] and a modified version of the noisy power method [Hardt and Price, 2014]. All of these works operate in a deterministic setting, and require some form of norm

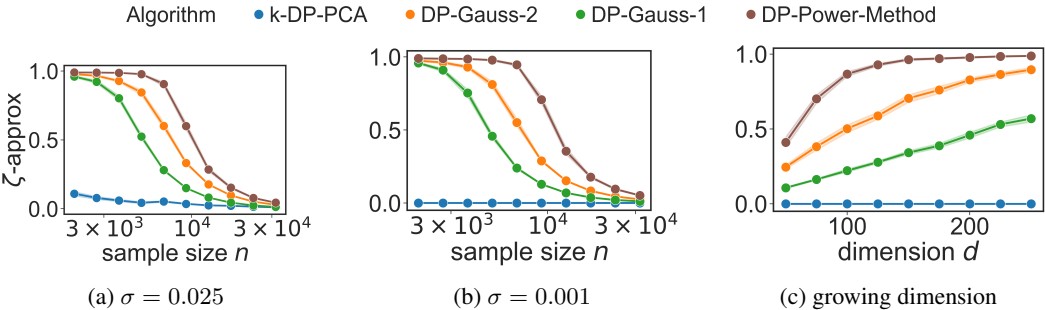

(a) $\sigma = 0.025$     (b) $\sigma = 0.001$     (c) growing dimension

Figure 1: Comparison of k-DP-PCA vs DP-Gauss-1 (input perturbation), DP-Gauss-2 (output perturbation), and DP-Power-Method on the spiked covariance model. We plot the mean over 50 trials, with shaded regions representing 95% confidence intervals. We set $k = 2$, $d = 200$, $\lambda_1 = 10$, $\varepsilon = 1$, and $\delta = 0.01$.

bound on the matrices to ensure differential privacy. Dwork et al. [2014b] requires each row of the data matrix $X \in \mathbb{R}^{n \times d}$ to be bounded in $\ell_2$-norm by 1 and they then estimate the top eigenvectors of $X^\top X$. The original noisy power method given a matrix $A \in \mathbb{R}^{d \times d}$, allowed only single entry changes by at most $\pm 1$, however more recent analysis [Nicolas et al., 2024, Florina Balcan et al., 2016] showed that it protects the privacy for changes of the form $A' = A + C$, with $\sqrt{\sum_{i=1}^n \|C_{i,:}\|_1^2} \leq 1$. By contrast, our setting is stochastic: we draw independent matrices $A_i$, without any norm constraint, and we estimate the top eigenvectors of $\mathbb{E}[A_i] = \Sigma$. Thus, we first adapt these algorithms to also guarantee privacy in our setting. Note that if we draw observations $x_i$ from a distribution with mean zero and covariance $\Sigma$, then $X^\top X = \sum_{i=1}^n x_i x_i^\top$ serves as an unbiased estimate of $n\Sigma$. A naive way to enforce the bounded norm requirement of Dwork et al. [2014b], is to define $\tilde{x}_i = x_i / \max\{\|x_i\|_2\}$. However, this non-private pre-processing step will violate privacy [Hu et al., 2024]: modifying a single $x_i$ can potentially change the maximum norm and thus affect all of the $\tilde{x}_i$. A natural next attempt is to scale each vector exactly to unit norm, i.e., $\tilde{x}_i = x_i / \|x_i\|_2$. However, this will result in a biased estimator as $\mathbb{E}\left[xx^\top // \|x\|^2\right] \neq \Sigma$ and thus does not enjoy meaningful utility guarantees. Instead, we clip each $x_i$ at $\beta$ so that with probability at least $1 - \vartheta$,s $\|x_i\|_2 \leq \beta$. Then scaling the Gaussian noise in the DP-Gauss mechanisms by $\beta$ maintains $(\varepsilon, \delta)$-DP guarantee. For the spiked covariance model this would mean $\beta = C\sqrt{\lambda_1} + \sigma\sqrt{d \log(n/\vartheta)}$. Using this strategy we modify Algorithm 1 and 2 in Dwork et al. [2014b] and refer to them as `DP-Gauss-1` and `DP-Gauss-2` respectively. `DP-Gauss-1` first clips each $x_i$, adds appropriately scaled Gaussian noise to the sum $\sum_i \tilde{x}_i \tilde{x}_i^\top$, and then performs standard (non-private) PCA. `DP-Gauss-2`, on the other hand, begins by privately estimating the eigengap of the clipped covariance matrix, runs non-private PCA on the clipped data, and finally perturbs the resulting top-$k$ eigenvectors with noise that scales with that that privately computed eigengap. Similarly to what we do for the `DP-Gauss` algorithms, to enforce the condition $\sqrt{\sum_{i=1}^n \|C_{i,:}\|_1^2} \leq 1$ by Nicolas et al. [2024] we define $A' = A + aa^\top$, meaning $C = aa^\top$, then the ith row of $C$ is equal to $|a_i| \|a\|_1$, which results in the requirement $\|a\|_2 \|a\|_1 \leq 1$. So we clip the matrices to $\|a\|_1 \leq \alpha$, and $\|a\|_2 \leq \beta$ (same $\beta$ as for DP-Gauss) and scale the privacy noise accordingly. For the spiked covariance model we choose $\alpha = \sigma d + \sqrt{\lambda_1 d} + \sigma\sqrt{d \log(n/\vartheta)}$, to achieve $\|x_i\|_1 \leq \alpha$ with probability $1 - \vartheta$. This makes their algorithm comparable to DP-Gauss in terms of utility guarantees with respect to k and d. However, as we will see, it is still outperformed by the DP-Gauss algorithms. In the rest of this section, Figure 1 compares k-DP-PCA with `DP-Gauss-1`, `DP-Gauss-2` and `DP-Power-Method` across various noise levels $\sigma$ and dimensions $d$. Figure 2 also incorporates the much simpler-to-implement k-DP-Ojas algorithm and shows that a simpler, more scalable algorithm can match or even outperform k-DP-PCA in practice, despite its slightly weaker theoretical guarantee.

**Experimental Results using Spiked Covariance Data** We evaluate all methods on the spiked-covariance model(see Example 1). Figures 1a and 1b show utility as a function of sample size for large and small noise levels, respectively. Our results show that across both regimes, k-DP-PCA consistently outperforms the baselines, with the gap widening when the noise level is significantly smaller than the signal strength ($\sigma \ll \lambda_1$). Figure 1c examines the effect of increasing ambient dimension $d$ at fixed $n$. As $d$ grows, the DP-Gauss methods' and Power-Method's utility degrades faster than k-DP-PCA 's, reflecting the fact that their theoretical utility scales like $O(d^{3/2}/n)$, whereas our guarantee only incurs a linear dependence on $d$.

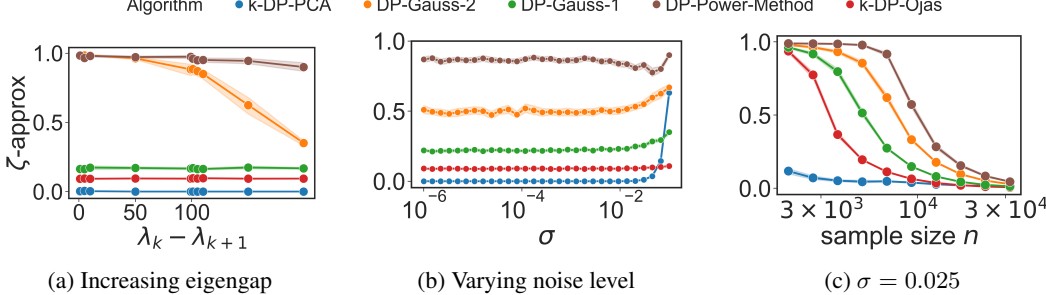

(a) Increasing eigengap      (b) Varying noise level      (c) $\sigma = 0.025$

Figure 2: Comparison of k-DP-PCA and k-DP-Ojas in a higher noise regime (also including DP-Gauss-1 (input perturbation), DP-Gauss-2 (object perturbation), and DP-Power-Method) on the spiked covariance model. We plot the mean over 50 trials, with shaded regions representing 95% confidence intervals. We set $k = 2$, $d = 200$, $\lambda_1 = 10$, $\varepsilon = 1$, and $\delta = 0.01$.

In Figure 2a, we plot the utility against the eigengap ($\lambda_k - \lambda_{k+1}$) for different algorithms. DP-Gauss-2, which is designed with large eigen-gaps in mind steadily improves in utility as the gap grows and nearly matches the utility of k-DP-PCA for very large eigengap. By contrast, DP-Gauss-1 which offers better scalability with dimension $d$ but is insensitive to the eigen-gap, maintains a nearly flat utility as the eigen-gap grows. Throughout, k-DP-PCA consistently outperforms both DP-Gauss algorithms.

Next, in Figure 2, we compare k-DP-PCA against the much simpler k-DP-Ojas algorithm. As predicted by Corollaries 2 and 4, k-DP-PCA clearly outperforms k-DP-Ojas in the low-noise regime ($\sigma \ll \lambda_1$). Conversely, at larger noise levels k-DP-Ojas often matches or even exceeds k-DP-PCA in practice, owing to its fewer hyperparameters and greater robustness to learning-rate choices (see Figure 2b and appendix G). Although both algorithms require knowledge of the eigenvalues of $\Sigma$ to set optimal step sizes, these can be obtained privately via the Gaussian mechanism. Nevertheless, it is interesting to note that k-DP-Ojas remains effective even when its step size is chosen without any explicit eigenvalue estimates (see Appendix G). Lastly, we want to note that in Appendix G we present more comprehensive results, using different $d$ and $k$. The results in this section are kept simple for illustrative purposes.

## 6 Related Work and Open problems

**Related Work** Differentially private PCA has been studied extensively [Blum et al., 2005, Chaudhuri et al., 2013, Hardt and Roth, 2013, Dwork et al., 2014b]. However, when applied to the stochastic setting, these methods typically suffer from sample complexity that scales super-linearly in $d$ or inject noise at a scale that ignores the underlying stochasticity in the data, resulting in suboptimal error rates of $O(\sqrt{dk/n} + d^{3/2}k/(\varepsilon n))$. The first to address these limitations were [Liu et al., 2022b, Cai et al., 2024]; however the results by [Liu et al., 2022b] only apply for $k = 1$ and Cai et al. [2024] provide an algorithm whose privacy guarantee is conditional on distributional assumptions on the data. In contrast, our algorithm applies to all $k \le d$, is private for all inputs, provides an error rate that scales linearly with $d$, and the injected noise scales with the inherent stochasticity in the data.

A complimentary line of work, [Singhal and Steinke, 2021, Tsfadia, 2024] obtains sample complexity that scales independently of the dimension $d$ but requires a strong multiplicative eigengap $(\lambda_k/\lambda_{k+1}) = O(\sqrt{d})$, which is a strictly stronger assumption than ours.

**Open Problems** Despite being a mild concentration requirement also seen in prior work [Liu et al., 2022a], Assumption A.4 is perhaps the most non-standard assumption in Assumption A. As observed by Liu et al. [2022a], this can be relaxed to a bounded $k$-th moment condition, at which point the second term in (23) grows to $O(d(\log(1/\delta)/\varepsilon n)^{1-1/k})$. Further, empirical improvements may also be possible from applying private robust mean estimation [Liu et al., 2021, Hopkins et al., 2022], as opposed to clipping around the mean of the gradients. Lastly, the current PRIVRANGE is optimal for spiked covariance data, however for other data distributions we expect different range estimators to work better. We leave this to future work.

The sample size condition in Equation (1) includes an exponential dependence on the spectral gap: $n \geq \exp(\kappa')$. While this is relatively harmless as there is no such exponential dependence in the utility guarantee Equation (2), we show in Appendix E.2 how to get rid of this exponential dependence by incurring an additional $\tilde{O}(\gamma d^2 \log(1/\delta)/(\varepsilon n))$ term in the utility guarantee.

As already mentioned in Section 3.3, our upper bounds are loose in their dependence in $k$ and $\delta$. We incur this additional $k$ factor, because each deflation step must use a fresh batch of samples, so that the projection matrices $P$ remain independent of the data matrices in Line 4 of Algorithm 1. If one could safely reuse the same $A_i$'s across rounds, this could be improved to $O(\sqrt{k})$ via adaptive composition. We think it is interesting future work to see whether we can obtain a $\sqrt{k}$ factor using the techniques from the robust PCA results in Jambulapati et al. [2024] or using our analysis but with "slightly" correlated data. However, even if one theses approaches turn out to be viable, a gap still remains between the resulting upper bound and our lower bound and it is an interesting question to resolve this. Finally, although inspired by the streaming analysis of Oja's method [Jain et al., 2016, Huang et al., 2021], our subroutines (MODIFIEDDP-PCA, PRIVRANGE, PRIVMEAN) are not directly streaming-compatible. Adapting them to the streaming setting is an interesting avenue for future work.

## 7 Conclusion

We have presented the first algorithm for stochastic $k$-PCA that is both differentially private and computationally efficient, supports any $k \leq d$, and achieves near-optimal error. Our analysis critically relies on our adaptation of the DP-PCA algorithm [Liu et al., 2022a], a stochastic deflation framework inspired by [Jambulapati et al., 2024], and our novel analysis of non-private Oja's algorithm [Jain et al., 2016]. Along with our novel results in the *Stochastic k-PCA* problem, we believe the above mentioned theoretical results are of independent interest, and may inspire the developement of new algorithms for this and related problems.

## 8 Acknowledgement

JD acknowledges support from the Danish Data Science Academy, which is funded by the Novo Nordisk Foundation (NNF21SA0069429) and VILLUM FONDEN (40516). AS acknowledges the Novo Nordisk Foundation for support via the Startup grant (NNF24OC0087820) and VILLUM FONDEN via the Young Investigator program (72069). The authors would also like to thank Rasmus Pagh for very insightful discussions.

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

## Appendix

The appendix is structured as follows. In Appendix A, we provide a more detailed overview of related work to complement the discussion in the main text. Appendix B introduces mathematical and privacy-related preliminaries that lay the groundwork for our analysis. In Appendices C and D, we present novel technical contributions: in C we extend the recent deflation analysis of Jambulapati et al. [2024] to the stochastic setting and also prove Theorem 2 in the main text, and in D we provide a new utility analysis of the non-private Oja's algorithm. These results are then used to prove our main theorem and establish the utility and privacy guarantees for our second proposed algorithm (Corollary 4), k-DP-Ojas, in Appendix E. In Appendix F, we prove our lower bound result from Section 3.3. We then provide additional experimental details in Appendix G, and conclude by restating the subroutines from Liu et al. [2022b], which are used in MODIFIEDDP-PCA, in Appendix H.

## A   Related Work

The problem of private $k$-PCA has been the subject of extensive research, with many works exploring it under various constraints. Several works address $k$-PCA in the standard setting, while assuming an additive eigengap [Blum et al., 2005, Chaudhuri et al., 2013, Hardt and Roth, 2013, Dwork et al., 2014b, Nicolas et al., 2024]. These works operate in a deterministic setting where each sample is assumed to be bounded ($\|x_i\| \leq \beta$). When applied to the stochastic setting, these works generally yield suboptimal error rates. This is partially due to the fact that all of these works assume a data independent bound ($\beta = 1$), which we cannot easily enforce in the stochastic setting (as discussed in Section 5). Considering Gaussian data with $x_i \sim \mathcal{N}(0, \Sigma)$, we know $\|x_i\| \leq \beta = O(\sqrt{\lambda_1 d \log(n/\zeta)}$ for all $i$ with probability $1 - \zeta$. [Blum et al., 2005, Dwork et al., 2014b, Nicolas et al., 2024] use the Gaussian mechanism, so when scaling the privacy noise with a factor $\beta$ we ensure $(\varepsilon, \delta)$-DP in the stochastic setting. The tightest of the previous discussed result then achieves

$$O\left(\sqrt{dk/n} + d^{3/2}k/(\varepsilon n)\right).$$

More recent work has considered the multiplicative eigengap setting [Tsfadia, 2024, Singhal and Steinke, 2021], though this is a strictly stronger assumption. Finally, there is a set of results without spectral gap assumptions [Chaudhuri et al., 2013, Kapralov and Talwar, 2013, Liu et al., 2022b]. However, these works either do not allow a tractable implementation or give utility bounds that are super-linear in their dependence on $d$.

A widely used strategy in the non-private PCA literature to mitigate the complexity of designing algorithms for $k$-PCA is to reduce the $k$-dimensional problem to a series of 1-dimensional problems using a technique known as the deflation method [Mackey, 2008, Allen-Zhu and Li, 2016]. Jambulapati et al. [2024] proved significantly sharper bounds on the degradation of the approximation parameter of deflation methods for $k$-PCA. While their analysis only catered to the standard non-stochastic setting and assumed access to the true covariance matrix $\Sigma$, their results serve as a conceptual foundation for this work. We extend similar arguments to the stochastic setting, where only access to sample matrices $A_i$ with shared expectation $\mathbb{E}[A_i] = \Sigma$ is available.

Our 1-PCA method builds upon Oja's algorithm [Oja, 1982], (see Algorithm 5), one of the oldest and most popular algorithms for streaming PCA. The first formal utility guarantees for Oja's algorithm in the $k = 1$ case were established by Jain et al. [2016], whose analysis inspired our proofs in Appendix D. Subsequent extensions to the $k > 1$ case were provided by Huang et al. [2021].

Lastly, our ePCA oracle MODIFIEDDP-PCA is largely inspired by the DP-PCA algorithm of Liu et al. [2022b]. Their result builds upon a series of advances in private SGD [Kamath et al., 2022, Bassily et al., 2014, 2019, Feldman et al., 2020, Kulkarni et al., 2021, Wang et al., 2020, Hu et al., 2022], and private mean estimation [Bun and Steinke, 2019, Karwa and Vadhan, 2017, Kamath et al., 2019, Biswas et al., 2020, Feldman and Steinke, 2018, Tzamos et al., 2020]. In this work, we use some of the techniques proposed by Liu et al. [2022b]: specifically their PRIVMEAN and PRIVRANGE algorithms. Replacing them with robust and private mean estimation [Liu et al., 2021, Kothari et al., 2022] could relax Assumption A.4, but at the cost of sub-optimal sample complexity.

# B   Preliminaries

In this section we list some mathematical and privacy preliminaries. A familiar reader is welcome to skip this section.

## B.1   Mathematics Preliminaries

**Lemma 2.** *Let $C, D \in \mathbb{R}^{d \times d}$ be symmetric matrices and let $A \in \mathbb{R}^{m \times d}$ be any conformable matrix. Then $C \preceq D \implies ACA^\top \preceq ADA^\top$*

*Proof.* Since $C \preceq D$, we have $C - D \preceq 0$. For any $x \in \mathbb{R}^m$, set $y = A^\top x$. Then
$$x^\top A(C - D)Ax = y^\top (C - D)y \leq 0.$$
which shows $ACA^\top \preceq ADA^\top$. $\qquad\square$

**Theorem 4** (Woodbury matrix identity). *Let $A \in \mathbb{R}^{n \times n}$ and $C \in \mathbb{R}^{k \times k}$ be invertible matrices, and let $U \in \mathbb{R}^{n \times k}$, $V \in \mathbb{R}^{k \times n}$. Then*
$$(A + UCV)^{-1} = A^{-1} - A^{-1}U(C^{-1} + VA^{-1}U)^{-1}VA^{-1}$$

**Theorem 5** (Pinsker's Inequality). *For $P$ and $Q$ two probability distributions on a measurable space then*
$$TV(P, Q) \leq \sqrt{\frac{1}{2}KL(P\|Q)}$$

**Lemma 3** (Lemma F.2 in [Liu et al., 2022a]). *Let $G \in \mathbb{R}^{d \times d}$ be a random matrix whose entries $G_{ij}$ are i.i.d. $\mathcal{N}(0, 1)$. Then there exists a universal constant $C > 0$ such that for all $t > 0$,*
$$\Pr\left[\|G\|_2 \leq C(\sqrt{d} + t)\right] \geq 1 - 2e^{-t^2}.$$

**Lemma 4** (Lemma F.5 in [Liu et al., 2022a]). *Under Assumptions A.1 to A.3, with probability at least $1 - \tau$*
$$\left\|\frac{1}{B}\sum_{i \in [B]} A_i - \Sigma\right\|_2 = O\left(\sqrt{\frac{\lambda_1^2 V \log(d/\tau)}{B}} + \frac{\lambda_1 M \log(d/\tau)}{B}\right)$$

**Lemma 5** (Adapted Version of Lemma F.3 in [Liu et al., 2022a]). *Let $G \in \mathbb{R}^{d \times d}$ be a random matrix where each entry $G_{ij}$ is i.i.d. sampled from standard Gaussian $\mathcal{N}(0, 1)$. Then we have*
$$\mathbb{E}[\|GG^\top\|_2] \leq Cd \qquad (4)$$

*Proof.*
$$\mathbb{E}[\|GG^\top\|_2] \leq \mathbb{E}[\|G\|_2^2]$$
$$= \int_0^\infty \mathbb{P}(\|G\|_2^2 > u)du = \int_0^\infty \mathbb{P}(\|G\|_2 > \sqrt{u})du$$
$$= \int_0^\infty \mathbb{P}(\|G\|_2 > r)2rdr$$

where we do the change of variable with $r := \sqrt{u}, u = r^2, du = 2rdr$. Next we split the integral into two parts using the "concentration radius" $r_0 = C_1\sqrt{d}$, as by Lemma 3 the exists a universal constant $C_1 > 0$ such that
$$\mathbb{P}(\|G\| \geq C_1(\sqrt{d} + s)) \leq e^{-s^2}, \forall s > 0$$
this gives us
$$\mathbb{E}[\|GG^\top\|_2] := \int_0^{r_0} \mathbb{P}(\|G\|_2 > r)2rdr + \int_{r_0}^\infty \mathbb{P}(\|G\|_2 > r)2rdr$$
$$\leq \int_0^{r_0} 2rdr + \int_{r_0}^\infty \mathbb{P}(\|G\|_2 > r)2rdr$$
$$= C_1^2 d + \int_{r_0}^\infty \mathbb{P}(\|G\|_2 > r)2rdr$$

for the second integral we use again Lemma 3 or rather its equivalent form

$$\mathbb{P}(\|G\| \geq r) \leq e^{-(\frac{r}{c_1} - \sqrt{d})^2}$$

which gives us

$$
\begin{aligned}
\int_{r_0}^{\infty} \mathbb{P}(\|G\|_2 > r) 2r dr &= \int_{r_0}^{\infty} e^{-(\frac{r}{c_1} - \sqrt{d})^2} 2r dr \\
&= \int_0^{\infty} C_1^2 (\sqrt{d} + s) e^{-s^2} ds \\
&= C_1^2 \sqrt{d} \int_0^{\infty} e^{-s^2} ds + C_1^2 \int_0^{\infty} s e^{-s^2} ds \\
&\leq C_2 \sqrt{d} + C_3
\end{aligned}
$$

where we used $r = C_1(\sqrt{d} + s)$ and $dr = C_1 ds$ in the second step, which finishes our proof.

$\square$

**Lemma 6** (Weyl's inequality [Horn and Johnson, 2012]). *Let $G_1$ and $G_2$ be two symmetric matrices with eigenvalues $\mu_1 \geq \cdots \geq \mu_d$ and $\nu_1 \geq \cdots \geq \nu_d$ respectively, then*

$$|\nu_i - \mu_i| \leq \|G_1 - G_2\|_2$$

**Lemma 7** (Conditional Markov Inequality). *Let $\mathcal{F}$ be a sigma-algebra, let $X > 0$ be a non negative random variable, and let $a > 0$. Then*

$$P(X \geq a | \mathcal{F}) \leq \frac{\mathbb{E}[X|\mathcal{F}]}{a}.$$

*Proof.* Define the indicator

$$
I_{\{X \geq a\}} = \begin{cases} 1, & X \geq a \\ 0, & \text{o.w.}. \end{cases}
$$

Then $X, I_{\{X \geq a\}} \geq a I_{\{X \geq a\}}$. Taking conditional expectation given $\mathcal{F}$ on both sides yields

$$\mathbb{E}\left[ X I_{\{X \geq a\}} \mid \mathcal{F} \right] \geq \mathbb{E}\left[ a I_{\{X \geq a\}} \mid \mathcal{F} \right] = a \Pr(X \geq a \mid \mathcal{F}).$$

Hence, $\Pr(X \geq a \mid \mathcal{F}) \leq \dfrac{\mathbb{E}[X \mid \mathcal{F}]}{a}$.

$\square$

**Lemma 8** (Conditional Chebyshev's Inequality). *Let $\mathcal{F}$ be a conditioning event (or a sigma-algebra), then for $a > 0$*

$$P(|X - \mathbb{E}[X|\mathcal{F}]| \geq a | \mathcal{F}) \leq \frac{Var[X|\mathcal{F}]}{a^2}$$

*where $Var[X|\mathcal{F}] = \mathbb{E}[(X - \mathbb{E}[X|\mathcal{F}])^2 | \mathcal{F}]$.*

*Proof.*

$$P(|X - \mathbb{E}[X|\mathcal{F}]| \geq a | \mathcal{F}) = P((X - \mathbb{E}[X|\mathcal{F}])^2 \geq a^2 | \mathcal{F})$$

$(X - \mathbb{E}[X|\mathcal{F}])^2$ is a non negative random variable, so we can use conditional Markov (Lemma 7), which gives us

$$P((X - \mathbb{E}[X|\mathcal{F}])^2 \geq a^2 | \mathcal{F}) \leq \frac{\mathbb{E}[(X - \mathbb{E}[X|\mathcal{F}])^2 | \mathcal{F}]}{a^2}$$

$\square$

**Lemma 9** (Distributional Equivalence). *Let $z \sim \mathcal{N}(0, \Sigma)$ be a $d$-dimensional Gaussian with covariance $\Sigma \succ 0$. Let $P \in \mathbb{R}^{d \times d}$ be an orthogonal projection matrix, and fix any unit vector $\omega \in \text{Im}(P)$. Then there exists a random matrix $G = \Sigma^{1/2} Y$, where each entry in $Y \in \mathbb{R}^{d \times d}$ is sampled i.i.d. from $\mathcal{N}(0, 1)$, such that*

$$Pz \stackrel{d}{=} PGP\omega.$$

*Proof.* Since $z \sim N(0, \Sigma)$ and $P$ is a projection matrix, we have $\text{Cov}(Pz) = P\Sigma P^\top = P\Sigma P$. On the other hand, let $G = \Sigma^{1/2} Y$. Then for any fixed $\omega \in \text{Im}(P)$ with $\|\omega\| = 1$,

$$\text{Cov}(G\omega) = \Sigma^{1/2} \text{Cov}(Y\omega) \Sigma^{1/2} = \Sigma^{1/2} I_d \Sigma^{1/2} = \Sigma,$$

because $Y\omega \sim \mathcal{N}(0, I_d)$ by rotational invariance of spherical gaussian (and $\|\omega\| = 1$). Hence $\text{Cov}(PGP\omega) = P\Sigma P = \text{Cov}(Pz)$. Since both $Pz$ and $PGP\omega$ are mean–zero Gaussians with the same covariance, we have

$$Pz \stackrel{d}{=} PGP\omega.$$

$\square$

**Lemma 10.** *For any matrix $A \in \mathbb{R}^{d \times d}$ and any projection matrix $P$,*

$$\|PAP\|_2 \le \|A\|_2$$

*Proof.* For any unit vector $x$, $\|PAPx\|_2 = \|P(APx)\|_2 \le \|APx\|_2 \le \|A\| \|Px\|_2 \le \|A\|_2$, where the last inequality follows as projection matrices have eigenvalues in $\{0, 1\}$. Taking ths supremum over all $x$ completes the proof. $\square$

**Lemma 11.** *Let $A \in \mathbb{R}^{d \times d}$ be a random matrix and $P$ a random projection matrix, independent of $A$. Then*

$$\|\mathbb{E}[PAPA^\top P]\|_2 \le \|\mathbb{E}[AA^\top]\|_2.$$

*Proof.* We first show that for any orthogonal projection $P$, we have $PAPA^\top P \preceq PAA^\top P$. Since $P$ is an orthogonal projection, $P = P^\top$ and $P^2 = P$. Consider the difference:

$$PAA^\top P - PAPA^\top P = PA(I)A^\top P - PAPA^\top P.$$

Using the identity $I = P + (I - P)$, the expression becomes

$$PA(P + I - P)A^\top P - PAPA^\top P = PA(I - P)A^\top P \succeq 0.$$

where the last step follows as $I - P$ is also an orthogonal projection. This implies

$$PAPA^\top P \preceq PAA^\top P.$$

Taking expectation (over both $P$ and $A$) then yields

$$\mathbb{E}\left[PAPA^\top P\right] \preceq \mathbb{E}\left[PAA^\top P\right]. \tag{5}$$

As $P$ is independent of $A$, one has

$$\mathbb{E}_{P,A}\left[PAA^\top P\right] = \mathbb{E}_P\left[\mathbb{E}_A\left[PAA^\top P \mid P\right]\right] = \mathbb{E}_P\left[P\mathbb{E}_A\left[AA^\top\right] P\right] = \mathbb{E}_P\left[PMP\right]$$

where $M := \mathbb{E}_A\left[AA^\top\right]$. Combining with previous step, we get

$$\mathbb{E}_{P,A}\left[PAPA^\top P\right] \preceq \mathbb{E}_P\left[PMP\right]. \tag{6}$$

Finally, we show

$$\|\mathbb{E}_P\left[PMP\right]\|_2 \le \|M\|_2. \tag{7}$$

Indeed, for any fixed projection $P$, the largest eigenvalue of $PMP$ can be written as

$$\lambda_{\max}(PMP) = \max_{\|x\|=1} x^\top (PMP) x = \max_{\|x\|=1} (Px)^T M (Px).$$

Since $\|P\,x\| \leq 1$ whenever $\|x\| = 1$, it follows

$$(Px)^\top M (Px) \leq \max_{\|y\| \leq 1} y^T M y = \|M\|_2 .$$

Taking the maximum over all $\|x\| = 1$ shows $\|PMP\|_2 \leq \|M\|_2$. Hence $\mathbb{E}_P [\|PMP\|_2] \leq \|M\|_2$. Because the operator norm $\|\cdot\|_2$ is convex,

$$\|\mathbb{E}_P [PMP]\|_2 \leq \mathbb{E}_P [\|PMP\|_2] \leq \|M\|_2 ,$$

which is Equation (7).

Now combine Equations (6) and (7), we have

$$\left\|\mathbb{E}\left[PAPA^\top P\right]\right\|_2 \leq \|\mathbb{E}_P [PMP]\|_2 \leq \|M\|_2 = \left\|\mathbb{E}\left[AA^\top\right]\right\|_2 .$$

This completes the proof. $\qquad\square$

**Lemma 12.** *Let $A$ and $B$ be independent random matrices in $\mathbb{R}^{d \times d}$. Then*

$$\mathbb{E}\left[ABA^\top\right] \preceq \|\mathbb{E}[B]\|_2 \, \mathbb{E}\left[AA^\top\right]$$

*Proof.* Since $A$ and $B$ are independent, we have $\mathbb{E}\left[ABA^\top\right] = \mathbb{E}\left[A\mathbb{E}[B]A^\top\right]$. Then, using $\mathbb{E}[B] \preceq \|\mathbb{E}[B]\|_2 \mathbf{I}_d$ and Lemma 2 we obtain the wished inequality. $\qquad\square$

**Lemma 13.** *Fix any projection matrix $P \in \mathbb{R}^{d \times d}$. Define, for each unit vector $u \in \mathbb{R}^d$,*

$$H_u^P = \frac{1}{\lambda_1^2 (P\Sigma P)} \mathbb{E}\left[P (A_i - \Sigma) P u u^\top P (A_i - \Sigma) P\right], \quad \gamma_P^2 = \max_{\|u\|=1} \left\|H_u^P\right\|_2 ,$$

*where $\lambda_1$ and $\gamma$ are as defined in Assumption A, and $\lambda_1^2 (P\Sigma P)$ refers to the top eigenvalue of $P\Sigma P$. Then*

$$\lambda_1^2 (P\Sigma P) \, \gamma_P^2 \leq \lambda_1^2 \gamma^2 .$$

*Proof.*

$$
\begin{aligned}
\|\mathbb{E}\left[P(A_i - \Sigma)Puu^\top P(A_i - \Sigma)P\right]\| &= \|\mathbb{E}_P\left[P\mathbb{E}[(A_i - \Sigma)Puu^\top P(A_i - \Sigma)|P]P\right]\| \\
&\leq \mathbb{E}_P\left[\|P\|\|\mathbb{E}[(A_i - \Sigma)Puu^\top P(A_i - \Sigma)|P]\|\|P\|\right] \\
&\leq \mathbb{E}_P\left[\|\mathbb{E}[(A_i - \Sigma)Puu^\top P(A_i - \Sigma)|P]\|\right]
\end{aligned}
$$

and further

$$\max_{\|u\|=1} \|\mathbb{E}[(A_i - \Sigma)Puu^\top P(A_i - \Sigma)|P]\| \leq \max_{\|u\|=1} \|\mathbb{E}[(A_i - \Sigma)uu^\top (A_i - \Sigma)|P]\| = \lambda_1^2 \gamma^2$$

as $Puu^\top P \preceq uu^\top$. So, all together this proves the Lemma. $\qquad\square$

**Definition 4.** Define $\mathbb{O}_{d,k}$ to denote the set of $d \times k$ matrices satisfying $U^\top U = \mathbf{I}_k$.

*Remark.* The Frobenius norm is equal to the Schatten-2 norm.

**Lemma 14** (Lemma 3 in [Jambulapati et al., 2024]). *Let $\Sigma \in \mathbb{S}_{\succeq 0}^{d \times d}$, $k \in [d]$. If $P \in \mathbb{R}^{d \times d}$ is a rank-$(d-k)$ orthogonal projection matrix, then $\|P\Sigma P\|_2 \geq \lambda_{k+1}(\Sigma)$.*

### B.2 Differential Privacy Preliminaries

**Lemma 15** (Parallel composition, [Dwork et al., 2014a]). *Suppose we have $K$ interactive queries $q_1, \ldots, q_K$, each acting on a disjoint subset $S_k$ of the database, and each query $q_k$ individually satisfies $(\varepsilon, \delta)$-DP on its subset $S_k$. Then the joint mechanism $(q_1(S_1), q_2(S_2), \ldots, q_K(S_K))$ is also $(\varepsilon, \delta)$-DP.*

**Lemma 16** (Advanced Composition, [Kairouz et al., 2015]). *Let $\varepsilon \leq 0.9$ and $0 < \delta < 1$. Suppose a database is accessed $k$ times, each time using a $\left(\varepsilon/(2\sqrt{2k\log(2/\delta)}), \delta/(2k)\right)$-DP mechanism. Then the overall procedure satisfies $(\varepsilon, \delta)$-DP.*

**Algorithm 4** BlackBoxPCA($\{A_i\}$, $k$, $O_{1\text{PCA}}$)  [Jambulapati et al., 2024]

---

**Input:** $n$ i.i.d. matrices $A_1, \ldots, A_n \in \mathbb{R}^{d \times d}$ with $\mathbb{E}[A_i] = \Sigma \succeq 0$,  target rank $k \in \{1, \ldots, d\}$, and $O_{1\text{PCA}}$ a stochastic 1-ePCA oracle which, inputs a batch of samples $A_{j_1}, \ldots, A_{j_\ell}$ and an orthogonal projector $P$, and returns a unit vector $u \in \text{Im}(P)$.

1: $P_0 \leftarrow I_d$
2: $B \leftarrow \lfloor n/k \rfloor$
3: **for** $i = 1, 2, \ldots, k$ **do**
4:  Draw the next batch $\big\{ A_{(i-1)B+1}, \ldots, A_{iB} \big\}$.
5:  $u_i \leftarrow O_{1\text{PCA}} \big( A_{(i-1)B+1}, \ldots, A_{iB}; P_{i-1} \big)$
6:  $P_i \leftarrow P_{i-1} - u_i u_i^\top$
7: **end for**
8: **return** $U \leftarrow \{u_i\}_{i \in [k]}$

---

## C Meta Algorithm for stochastic $k$-PCA

In this section we prove that any stochastic 1-ePCA oracle, when passed into Algorithm 4, yields a valid $k$-PCA algorithm. This is the basis for Theorem 2, as our argument applies to *any* randomized stochastic 1-ePCA oracle (not necessarily private). In particular, it generalizes the utility analysis of Jambulapati et al. [2024] to the stochastic setting where each call to the oracle sees only a fresh batch of i.i.d. matrices $A_i$, and must approximate the top eigenvector of $\mathbb{E}[A_i] = \Sigma$.

**Definition 5** (stochastic ePCA oracle)**.** An algorithm $O_{\text{ePCA}}$ is a $\zeta$–*approximate 1-ePCA oracle* if the following holds. On independent inputs $A_1, \ldots, A_n \in \mathbb{R}^{d \times d}$ with $\mathbb{E}[A_i] = \Sigma \in \mathbb{S}_{\succeq 0}^{d \times d}$ for all $i$ and any orthogonal projector $P \in \mathbb{R}^{d \times d}$, $O_{\text{ePCA}}$ returns a unit vector $u \in \text{Im}(P)$ such that, with high probability,

$$\langle uu^\top, P\Sigma P \rangle \geq (1 - \zeta^2)\langle vv^\top, P\Sigma P \rangle$$

where $v$ is the top eigenvector of the projected matrix $P\Sigma P$.

*Remark.* The DP-PCA algorithm in Liu et al. [2022a]) does not directly qualify as a stochastic 1-ePCA oracle, since it guarantees $\langle uu^\top, \mathbb{E}[P]\Sigma\mathbb{E}[P] \rangle \geq (1 - \zeta^2) \langle vv^\top, \mathbb{E}[P]\Sigma\mathbb{E}[P] \rangle$, rather than comparing to $P\Sigma P$ itself. It is not obvious in general how large $\mathbb{E}[P]\Sigma\mathbb{E}[P] - P\Sigma P$ can be.

We will now show that for this type of approximation algorithm we can obtain a utility guarantee and that it would be optimal for the spiked covariance setting. We now recall the energy formulation of approximate $k$-PCA from Jambulapati et al. [2024], which is the utility metric we will use here.

**Definition 6** (energy $k$-PCA, [Jambulapati et al., 2024])**.** Let $M \in \mathbb{S}_{\succeq 0}^{d \times d}$. A matrix $U \in \mathbb{R}^{d \times k}$ with orthonormal columns is a $\zeta$-approximate energy $k$-PCA of $M$ if

$$\big\langle UU^\top, M \big\rangle \geq (1 - \zeta^2) \, \|M\|_k$$

where

$$\|M\|_{(k)} := \max_{V \in \mathbb{R}^{d \times k}, V^\top V = I_k} \text{Tr}\big(VV^\top M\big).$$

The following lemma relates the angle between two unit vectors to the corresponding energy in $\Sigma$.

**Lemma 17.** *Let $v, w \in \mathbb{R}^d$ be unit vectors, let $\theta$ be the angle between them, and let $\Sigma \succeq 0$ be any PSD matrix with top-eigenvector $v$. Then*

$$\big\langle ww^\top, \Sigma \big\rangle \geq \big(1 - \sin^2(\theta)\big) \big\langle vv^\top, \Sigma \big\rangle$$

*Proof.* Observe

$$\big\langle ww^\top, \Sigma \big\rangle = \big\langle vv^\top, \Sigma \big\rangle - \big\langle vv^\top - ww^\top, \Sigma \big\rangle = \left(1 - \frac{\langle vv^\top - ww^\top, \Sigma \rangle}{\langle vv^\top, \Sigma \rangle}\right) \big\langle vv^\top, \Sigma \big\rangle \quad (8)$$

Note that since $v$ is the top eigenvector of $\Sigma$, we have

$$\langle vv^\top, \Sigma \rangle = \mathrm{Tr}\left(vv^\top \Sigma\right) = v^\top \Sigma v = \lambda_1$$

where $\lambda_1 \geq \cdots \geq \lambda_d$ denote the eigenvalues of $\Sigma$ and $v, v_2, \ldots, v_d$ the corresponding eigenvectors. Then, we can rewrite

$$\left(1 - \frac{\langle vv^\top - ww^\top, \Sigma \rangle}{\langle vv^\top, \Sigma \rangle}\right) = 1 - \left(1 - \frac{w^\top \Sigma w}{\lambda_1}\right) = 1 - \left(1 - \frac{w^\top vv^\top w}{\lambda_1} - \sum_{j=2}^{d} \frac{\lambda_j w^\top v_j v_j^\top w}{\lambda_1}\right)$$

$$= 1 - \left(1 - \langle w, v \rangle^2 - \sum_{j=2}^{d} \frac{\lambda_j \langle w, v \rangle^2}{\lambda_1}\right)$$

$$\geq 1 - \left(1 - \langle w, v \rangle^2\right) = \left(1 - \sin^2(\theta)\right)$$

Substituting this back in Equation (8) gives us

$$\langle ww^\top, \Sigma \rangle \geq \left(1 - \sin^2(\theta)\right) \langle vv^\top, \Sigma \rangle$$

and completes the proof. $\qquad \square$

We now prove that, if each $O_{1\mathrm{PCA}}$ call in Algorithm 4 approximates the top eigenvector of $P_{i-1} \Sigma P_{i-1}$, then the final $U$ is a $\zeta$-approximate energy $k$-PCA of $\Sigma$.

**Theorem 6** (Reduction from $k$-PCA to 1-ePCA). *Let $A_1, \ldots, A_n$ be i.i.d. samples in $\mathbb{R}^{d \times d}$ with $\mathbb{E}[A_i] = \Sigma \succeq 0$. Fix $\zeta \in (0, 1)$. Suppose $O_{1\mathrm{PCA}}$ is a $\zeta$-approximate stochastic 1-ePCA oracle as defined in Definition 5. If we run Algorithm 4 with $O_{1\mathrm{PCA}}$, then (with high probability) its output $U = \{u_i\}_{i=1}^{k}$ satisfies*

$$\langle UU^\top, \Sigma \rangle \geq \left(1 - \zeta^2\right) \|\Sigma\|_{(k)}.$$

*Proof.* Define $U_i := [u_1, \ldots, u_i] \in \mathbb{R}^{d \times i}$. We claim by induction on $i$ that

$$\mathrm{Tr}\left(U_i^\top \Sigma U_i\right) = \sum_{j=1}^{i} u_j^\top \Sigma u_j \geq \left(1 - \zeta^2\right) \|\Sigma\|_{(i)}.$$

**Base case** ($i = 1$) Since $P_0 = I_d$, by definition of the oracle, the first call returns $U_1$ satisfying

$$\mathrm{Tr}\left(U_1^\top \Sigma U_1\right) = u_1^\top \Sigma u_1 = \langle u_1 u_1^\top, \Sigma \rangle \geq \left(1 - \zeta^2\right) \lambda_{\max}(\Sigma) = \left(1 - \zeta^2\right) \|\Sigma\|_{(1)}.$$

**Inductive step.** Suppose after $i$ steps, $\mathrm{Tr}(U_i^\top \Sigma U_i) \geq (1 - \zeta^2) \|\Sigma\|_{(i)}$. Let $P_i = I_d - U_i U_i^\top$. Then, by definition of the oracle, the $(i+1)$-th call returns $u_{i+1} \in \mathrm{Im}(P_i)$ such that

$$\langle u_{i+1} u_{i+1}^\top, P_i \Sigma P_i \rangle \geq \left(1 - \zeta^2\right) \|P_i \Sigma P_i\|_2.$$

Since $\langle u_{i+1} u_{i+1}^\top, \Sigma \rangle \geq \langle u_{i+1} u_{i+1}^\top, P_i \Sigma P_i \rangle$, it follows

$$u_{i+1}^\top \Sigma u_{i+1} \geq \langle u_{i+1} u_{i+1}^\top, P_i \Sigma P_i \rangle \geq \left(1 - \zeta^2\right) \|P_i \Sigma P_i\|_2.$$

By Lemma 3 in Jambulapati et al. [2024] (restated as Lemma 14), we know $\|P_i \Sigma P_i\|_2 \geq \lambda_{i+1}(\Sigma)$. Hence,

$$\mathrm{Tr}\left(U_{i+1}^\top \Sigma U_{i+1}\right) = \mathrm{Tr}\left(U_i^\top \Sigma U_i\right) + u_{i+1}^\top \Sigma u_{i+1}$$

$$\geq \left(1 - \zeta^2\right) \|\Sigma\|_{(i)} + \left(1 - \zeta^2\right) \|P_i \Sigma P_i\|_2$$

$$\geq \left(1 - \zeta^2\right) \|\Sigma\|_{(i)} + \left(1 - \zeta^2\right) \lambda_{i+1}(\Sigma)$$

$$= \left(1 - \zeta^2\right) \|\Sigma\|_{(i+1)},$$

completing the induction.

Therefore, after $k$ steps, $\langle UU^\top, \Sigma \rangle = \mathrm{Tr}(U^\top \Sigma U) \geq \left(1 - \zeta^2\right) \|\Sigma\|_{(k)}.$ $\qquad \square$

---

**Algorithm 5** Oja's Algorithm

---

**Input:** $\{A_i\}_{i=1}^n$, learning rates $\{\eta_t\}_{t=1}^{\lfloor m \rfloor}$

  1: Choose $\omega_0$ uniformly at random from the unit sphere.
  2: **for** $t = 1, \ldots, n$ **do**
  3:     $\omega_t' \leftarrow \omega_{t-1} + \eta_t\, A_t\, \omega_{t-1}$
  4:     $\omega_t \leftarrow \omega_t' / \|\omega_t'\|_2$
  5: **end for**
  6: **return** $\omega_n$

---

**Theorem 2** (Meta Theorem). *Let $\Sigma \in \mathbb{S}_{\succeq 0}^{d \times d}$ and $A_1, \ldots, A_n$ be $n$ i.i.d. samples with $\mathbb{E}[A_i] = \Sigma$. Suppose we replace each 1-PCA step in Line 3 of Algorithm 1 by a $\zeta$–approximate stochastic ePCA oracle $O_{1\mathrm{PCA}}$. Then the deflation algorithm outputs $U \in \mathbb{R}^{d \times k}$ satisfying*

$$\langle UU^\top, \Sigma \rangle \geq (1 - \zeta^2)\|\Sigma\|_k.$$

*Further, for any $\varepsilon > 0, \delta \in (0, 1)$, if $O_{1PCA}$ is $\varepsilon, \delta$-DP then the entire algorithm remains $(\varepsilon, \delta)$-DP.*

*Proof.* Apply Theorem 6 to obtain the utility guarantee, and invoke Lemma 15 to conclude privacy under parallel composition. □

## D   Novel Analysis of non private Oja's Algorithm

Throughout this appendix, we condition on a fixed projection matrix $P$. All probability statements refer to randomness over the i.i.d. samples $\{A_i\}$, with $P$ held fixed. Whenever we write "with probability at least $1 - \delta$", it means $\Pr(\cdot \mid P) \geq 1 - \delta$. At the end, we apply a union bound to obtain an unconditional failure probability $\leq \delta$.

Let $A_1, \ldots, A_n$ be i.i.d. in $\mathbb{R}^{d \times d}$ with $\mathbb{E}[A_i] = \Sigma$. Denote the eigenvalues of $\Sigma$ with $\lambda_1 \geq \lambda_2 \geq \cdots \geq \lambda_d$ and corresponding eigenvectors $v_1, \ldots, v_d$. Let $P$ be a projection independent of $\{A_i\}_{i=1}^n$. Our goal is to approximate the top eigenvector of $P\Sigma P$.

When $P$ is deterministic, Jain et al. [2016] shows that Oja's algorithm outputs a vector close to the top eigenvector of $P\Sigma P$. However, in our setting $P$ itself is random, where $P$ is defined as $P = I - \sum_i u_i u_i^\top$ where each $u_i$ is computed using a prior independent sample of $\{A_i\}$. We cannot directly apply their result, since it would only guarantee closeness to the top eigenvector of $\mathbb{E}[P]\Sigma\mathbb{E}[P]$, and $\mathbb{E}[P]$ is generally not a projection matrix and may not preserve the spectral structure of interest.

To address this, we analyze Oja's algorithm on inputs $\{PA_iP\}$ and our main theorem shows that, under suitable conditions, the output is still an accurate approximation to the top eigenvector of $P\Sigma P$, even though $P$ is random and data-dependent. From here on, we write $\tilde{\lambda}_1 \geq \tilde{\lambda}_2 \geq \cdots \geq \tilde{\lambda}_d$ to denote the eigenvalues of $P\Sigma P$, and $\tilde{v}$ to denote its top eigenvector.

Assume scalars $\mathcal{M}, \mathcal{V}$ satisfy

$$\|A_i - \Sigma\|_2 \leq \mathcal{M} \text{ a.s.} \tag{9}$$

$$\max\left\{ \left\|\mathbb{E}\left[(A_i - \Sigma)(A_i - \Sigma)^\top\right]\right\|_2, \left\|\mathbb{E}\left[(A_i - \Sigma)^\top(A_i - \Sigma)\right]\right\|_2\right\} \leq \mathcal{V} \tag{10}$$

*Remark.* We intentionally use different notations $\mathcal{M}, \mathcal{V}$ here instead of $M, V$ than in Assumption A to simplify the expressions. Here $\mathcal{M} = \lambda_1 M$ and $\mathcal{V} = \lambda_1^2 V$ under Assumption A.

Next, define

$$B_n := (\mathbf{I} + \eta_n PA_nP)(\mathbf{I} + \eta_{n-1} PA_{n-1}P) \cdots \cdots (\mathbf{I} + \eta_1 PA_1P) \tag{11}$$

$$\omega_n := \frac{B_n\omega_0}{\|B_n\omega_0\|_2} \tag{12}$$

$$\bar{\mathcal{V}} := \mathcal{V} + \tilde{\lambda}_1^2 \tag{13}$$

where $\eta_i$ refers to the learning rate of Oja's Algorithm at step $i$, which in turn means $\omega_n$ is the output of Oja's Algorithm after $n$ steps given $\{PA_iP\}$ as input. We defined the variables like this in order to apply the following Lemma from Jain et al. [2016] to prove convergence of .

**Lemma 18** (One Step Power Method [Jain et al., 2016])**.** *Let $B \in \mathbb{R}^{d \times d}$, let $v \in \mathbb{R}^d$ be a unit vector, and let $V_\perp$ be a matrix whose columns form an orthonormal basis of the subspace orthogonal to $v$. If $\omega$ is sampled uniformly on the unit sphere then, with probability at least $1 - \delta$,*

$$\sin^2\left(v, \frac{Bw}{\|Bw\|_2}\right) = 1 - \left(v^\top B w\right)^2 \leq C \frac{\log(1/\delta)}{\delta} \frac{\mathrm{Tr}\left(V_\perp^\top B B^\top V_\perp\right)}{v^\top B B^\top v} \tag{14}$$

*where C is an absolute constant.*

Now we are ready to state the main theorem of this section.

**Theorem 7** (Main theorem of this section)**.** *Fix any $\delta > 0$ and set $\eta_t = \frac{\alpha}{(\tilde{\lambda}_1 - \tilde{\lambda}_2)(\beta + t)}$ for $\alpha > 1/2$, and define*

$$\beta := 20 \max \left( \frac{\mathcal{M}\alpha}{(\tilde{\lambda}_1 - \tilde{\lambda}_2)}, \frac{\bar{\mathcal{V}}\alpha^2}{(\tilde{\lambda}_1 - \tilde{\lambda}_2)^2 \log\left(1 + \frac{\delta}{100}\right)} \right).$$

*Suppose the number of iterations $n > \beta$. Then, with probability at least $1 - \delta$, the output $\omega_n$ of Algorithm 5 given inputs $\{PA_iP\}$ satisfies*

$$1 - \left(\omega_n^\top \tilde{v}\right)^2 \leq \frac{C \log(1/\delta)}{\delta^2} \left[ d\left(\frac{\beta}{n}\right)^{2\alpha} + \frac{\alpha^2 \mathcal{V}}{(2\alpha - 1)(\tilde{\lambda}_1 - \tilde{\lambda}_2)^2} \frac{1}{n} \right].$$

*Here C is an absolute numerical constant.*

*Remark.* Based on Lemma 18 to show Oja's algorithm (Algorithm 5) succeeds for our inputs we simply need that with high probability $\mathrm{Tr}\left(\tilde{V}_\perp^\top B_n B_n^\top \tilde{V}_\perp\right)$ is relatively large and $\tilde{v}^\top B_n B_n^\top \tilde{v}$ is relatively small, so that their ratio is large. Where $\tilde{v}$ refers to the top eigenvector of $P\Sigma P$ and $\tilde{V}_\perp$ is a matrix whose columns form an orthonormal basis of the subspace orthogonal to $\tilde{v}$. As long as we pick $\eta_i$ in Algorithm 5 sufficiently small, i.e. $\eta_i = O(1/\max M, \tilde{\lambda}_1)$ then $\mathbf{I} + \eta_i PA_iP$ is invertible, so in turn $B_n B_n^\top$, which guarantees $\tilde{v}^\top B_n B_n^\top \tilde{v} > 0$, so the RHS of the inequality will always be finite. In order to explicitly bound the RHS we will utilize conditional Chebychev's and Markov's, where the conditioning will serve to fix $P$.

*Proof of Theorem 7.* The proof is analogous to Theorem 4.1 in Jain et al. [2016], except we replace their Theorem 3.1 by our Theorem 8 stated and proved below. □

**Theorem 8.** *Given $A_1, \ldots, A_n$ that fulfill Assumptions A.1 to A.3 with parameters $\Sigma, M, V, \kappa$, a projection matrix $P$ independent of the $A_i$, $\tilde{v}$ the top eigenvector of $P\Sigma P$, and $B_n$ as in Equation (11), the output $\omega_n$ resulting from non-private Oja's Algorithm (Algorithm 5) on inputs $PA_1P, \ldots, PA_nP$ satisfies*

$$\sin\left(\tilde{v}, \frac{B_n \omega_n}{\|B_n \omega_n\|_2}\right) \leq \frac{1}{Q} \exp\left(\sum_{j=1}^t 5\eta_j^2 \bar{\mathcal{V}}\right) \left( d \exp\left(-2\left(\tilde{\lambda}_1 - \tilde{\lambda}_2\right) \sum_{j=1}^t \eta_j\right) \right),$$

*where $Q = \frac{\delta}{C\log(1/\delta)} \left(1 - \frac{1}{\sqrt{\delta}} \sqrt{\exp\left(\sum_{i=1}^n 18\eta_i^2 \bar{\mathcal{V}}\right) - 1}\right).$*

*Proof of Theorem 8.* By Lemma 18, applied after replacing $B$ with $B_n$, $v$ with $\tilde{v}$, and $V_\perp$ spanning $\tilde{v}^\perp$, we have with probability at least $1 - \delta$

$$\sin^2\left(\tilde{v}, \frac{B_n \omega}{\|B_n \omega\|_2}\right) \leq C \frac{\log(1/\delta)}{\delta} \frac{\mathrm{Tr}\left(V_\perp^\top B_n B_n^\top V_\perp\right)}{\tilde{v}^\top B_n B_n^\top \tilde{v}}. \tag{15}$$

It now remains to upper bound the numerator $\mathrm{Tr}\left(V_\perp^\top B_n B_n^\top V_\perp\right)$ and lower bound the denominator $\tilde{v}^\top B_n B_n^\top \tilde{v}$ separately.

**(i) Lower Bound the denominator** Using Conditional Chebychev's inequality (Lemma 8), we have

$$\mathbb{P}\left[\tilde{v}^\top B_n B_n^\top \tilde{v} \geq \mathbb{E}\left[\tilde{v}^\top B_n B_n^\top \tilde{v} \mid P\right] - \frac{1}{\sqrt{\delta}} \sqrt{\mathrm{Var}\left[\tilde{v} B_n B_n^\top \tilde{v} \mid P\right]}\right] < \delta. \tag{16}$$

Expand the variance expression as

$$\sqrt{\mathrm{Var}\left[\tilde{v}B_n B_n^\top \tilde{v} \mid P\right]} = \mathbb{E}\left[\tilde{v}B_n B_n^\top \tilde{v} \mid P\right]\sqrt{\Delta - 1}, \quad \text{where } \Delta = \frac{\mathbb{E}\left[\left(\tilde{v}B_n B_n^\top \tilde{v}\right)^2 \mid P\right]}{\mathbb{E}\left[\tilde{v}B_n B_n^\top \tilde{v} \mid P\right]^2}.$$

Then, we can rewrite Equation (16) to

$$\mathbb{P}\left[\tilde{v}^\top B_n B_n^\top \tilde{v} \geq \mathbb{E}\left[\tilde{v}^\top B_n B_n^\top \tilde{v} \mid P\right]\left(1 - \frac{1}{\sqrt{\delta}}\sqrt{\Delta - 1}\right)\right] < \delta. \tag{17}$$

Now, we need to bound the conditional expectation term and $\Delta$. Using Lemma 20, we bound the conditional expectation by

$$\mathbb{E}\left[\tilde{v}^\top B_n B_n^\top \tilde{v} \mid P\right] \geq \exp\left(\sum_{i=1}^n \left(2\eta_i \tilde{\lambda}_1 - 4\eta_i^2 \tilde{\lambda}_1^2\right)\right). \tag{18}$$

Then, using both Lemmas 20 and 21 we bound $\Delta$ as

$$\Delta = \frac{\mathbb{E}\left[\left(\tilde{v}^\top B_n B_n^\top \tilde{v}\right)^2 \mid P\right]}{\mathbb{E}\left[\tilde{v}^\top B_n B_n^\top \tilde{v} \mid P\right]^2} \leq \exp\left(\sum_{i=1}^n \eta_i^2\left(10\mathcal{V} + 8\tilde{\lambda}_1^2\right)\right) \leq \exp\left(\sum_{i=1}^n 18\eta_i^2\bar{\mathcal{V}}\right). \tag{19}$$

Plugging Equations (18) and (19) into Equation (17), bounds the denominator

$$\mathbb{P}\left[\tilde{v}^\top B_n B_n^\top \tilde{v} \geq \exp\left(\sum_{i=1}^n \left(2\eta_i \tilde{\lambda}_1 - 4\eta_i^2 \tilde{\lambda}_1^2\right)\right)\frac{Q}{\delta}\right] < \delta. \tag{20}$$

where

$$Q = \frac{\delta}{C\log\left(1/\delta\right)}\left(1 - \frac{1}{\sqrt{\delta}}\sqrt{\exp\left(\sum_{i=1}^n 18\eta_i^2\bar{\mathcal{V}}\right) - 1}\right).$$

**(ii) Upper Bound the numerator** Using Conditional Markov's inequality (Lemma 7) we have

$$\mathrm{Pr}\left[\mathrm{Tr}\left[\tilde{V}_\perp^\top B_n B_n^\top \tilde{V}_\perp\right] \geq \frac{\mathbb{E}\left[\mathrm{Tr}\left[\tilde{V}_\perp^\top B_n B_n^\top \tilde{V}_\perp\right] \mid P\right]}{\delta}\Bigg| P\right] \leq \delta \tag{21}$$

Using Lemma 22, we can bound the conditional expectation as

$$\mathbb{E}\left[\mathrm{Tr}\left[\tilde{V}_\perp^\top B_n B_n^\top \tilde{V}_\perp\right] \mid P\right] \leq \exp\left(\sum_{j\in[t]} 2\eta_j\tilde{\lambda}_2 + \eta_j^2\bar{\mathcal{V}}\right)\left(d + \mathcal{V}\sum_{i=1}^t \eta_i^2 \exp\left(\sum_{j\in[i]} 2\eta_j\left(\tilde{\lambda}_1 - \tilde{\lambda}_2\right)\right)\right)$$

$$\leq d\exp\left(\sum_{j\in[t]} 2\eta_j\tilde{\lambda}_2 + \eta_j^2\bar{\mathcal{V}}\right)$$

**(iii) Applying Union Bound** Using the above bounds, by applying a union bound over both the numerator and the denominator we have with probability $1 - 2\delta$, conditioned on $P$

$$\frac{\mathrm{Tr}\left(\tilde{V}_\perp^\top B_n B_n^\top \tilde{V}_\perp\right)}{\tilde{v}^\top B_n B_n^\top \tilde{v}} \leq Qd\exp\left(\sum_{j\in[t]} 2\eta_j\left(\tilde{\lambda}_2 - \tilde{\lambda}_1\right) + \eta_j^2\left(\bar{\mathcal{V}} + 4\tilde{\lambda}_1^2\right)\right)$$

Substituting this into Equation (15) completes the proof. $\qquad\square$

## D.1 Supporting Lemmas

We now state and prove several lemmas that together with Lemma 18 will allow us to prove Theorem 8 which in turn yields Theorem 7. The terms $\mathcal{M}, \mathcal{V}, \bar{\mathcal{V}}, B_t, \omega_n$ are defined in Equations (9) to (13). Further $\tilde{\lambda}_1 \geq \tilde{\lambda}_2 \geq \cdots \geq \tilde{\lambda}_d$ denote the eigenvalues of $P\Sigma P$, and $\tilde{v}$ to denotes its top eigenvector.

**Lemma 19.** $\left\| \mathbb{E}\left[B_t B_t^\top \mid P\right] \right\|_2 \leq \exp(\sum_{i \in [t]} 2\eta_i \tilde{\lambda}_1 + \eta_i^2 (\tilde{\lambda}_1^2 + \mathcal{V}))$

*Proof.* We denote $\alpha_t = \left\| \mathbb{E}\left[B_t B_t^\top \mid P\right] \right\|_2$, where $B_t = (\mathbf{I} + \eta_t P A_t P)(\mathbf{I} + \eta_{t-1} P A_{t-1} P) \cdots (\mathbf{I} + \eta_1 P A_1 P)$.

$$
\begin{aligned}
\mathbb{E}\left[B_t B_t^\top \mid P\right] &= \mathbb{E}\left[(\mathbf{I} + \eta_t P A_t P) B_{t-1} B_{t-1}^\top (\mathbf{I} + \eta_t P A_t P)^\top \mid P\right] \\
&\preceq \alpha_{t-1} \mathbb{E}\left[(\mathbf{I} + \eta_t P A_t P)(\mathbf{I} + \eta_t P A_t P)^\top \mid P\right] \qquad \text{(by Lemma 12)} \\
&= \alpha_{t-1} \mathbb{E}\left[\mathbf{I} + \eta_t P A_t P + \eta_t P A_t^\top P + \eta_t^2 P A_t P A_t^\top P \mid P\right] \\
&= \alpha_{t-1} \left(\mathbf{I} + 2\eta_t P \Sigma P + \eta_t^2 \mathbb{E}\left[P A_t P A_t^\top P \mid P\right]\right).
\end{aligned}
$$

We bound $P\Sigma P \preceq \tilde{\lambda}_1 \mathbf{I}$. Further,

$$
\begin{aligned}
\mathbb{E}\left[P A_t P A_t^\top P \mid P\right] &= P\Sigma P \Sigma P + \mathbb{E}\left[P(A_t - \Sigma) P (A_t - \Sigma)^\top P \mid P\right] \\
&= P\Sigma P \Sigma P + P \mathbb{E}\left[(A_t - \Sigma) P (A_t - \Sigma)^\top \mid P\right] P \\
&\preceq \tilde{\lambda}_1^2 \mathbf{I} + \mathbb{E}\left[(A_t - \Sigma)(A_t - \Sigma)^\top \mid P\right] \\
&= \tilde{\lambda}_1^2 \mathbf{I} + \mathbb{E}\left[(A_t - \Sigma)(A_t - \Sigma)^\top\right] \\
&\preceq \left\{\tilde{\lambda}_1^2 + \mathcal{V}\right\} \mathbf{I},
\end{aligned}
$$

where the third step follows as $\|P\|_2 \leq 1$, the 4th as $P$ is independent of $A_t$ and the last step by assumption on the $A_i$. Hence,

$$
\alpha_t \leq \alpha_{t-1} \left(1 + 2\eta_t \tilde{\lambda}_1 + \eta_t^2 \left(\tilde{\lambda}_1^2 + \mathcal{V}\right)\right).
$$

With $\alpha_0 = 1$ and $1 + x \leq e^x$,

$$
\alpha_t \leq \exp\left(\sum_{i \in [t]} \left(2\eta_i \tilde{\lambda}_1 + \eta_i^2 \left(\tilde{\lambda}_1^2 + \mathcal{V}\right)\right)\right). \qquad \square
$$

**Lemma 20.** $\mathbb{E}\left[\tilde{v}^\top B_t B_t \tilde{v} \mid P\right] \geq \exp\left(\sum_{i \in [t]} \left(2\eta_i \tilde{\lambda}_1 - 4\eta_i^2 \tilde{\lambda}_1^2\right)\right)$

*Proof.* Let $\beta_t := \mathbb{E}\left[\tilde{v}^\top B_t B_t^\top \tilde{v} \mid P\right]$, where $\tilde{v}$ is the top eigenvector of $P\Sigma P$ with eigenvalue $\tilde{\lambda}_1$. Since $B_t = (\mathbf{I} + \eta_t P A_t P) B_{t-1}$ and $A_t$ is independent of $B_{t-1}$ given $P$,

$$
\beta_t = \left\langle \mathbb{E}\left[B_{t-1} B_{t-1}^\top \mid P\right], \mathbb{E}\left[(\mathbf{I} + \eta_t P A_t P)\tilde{v}\tilde{v}^\top(\mathbf{I} + \eta_t P A_t P)^\top \mid P\right] \right\rangle.
$$

For the right hand side,

$$
\begin{aligned}
\mathbb{E}\left[(\mathbf{I} + \eta_t P A_t P)\tilde{v}\tilde{v}^\top(\mathbf{I} + \eta_t P A_t P)^\top \mid P\right] &= \tilde{v}\tilde{v}^\top + \eta_t P\Sigma P \tilde{v}\tilde{v}^\top + \eta_t \tilde{v}\tilde{v}^\top P\Sigma P \\
&\quad + \eta_t^2 \mathbb{E}\left[P A_t P \tilde{v}\tilde{v}^\top P A_t^\top P \mid P\right] \\
&\succeq \tilde{v}\tilde{v}^\top + 2\eta_t \tilde{\lambda}_1 \tilde{v}\tilde{v}^\top,
\end{aligned}
$$

because $P\Sigma P \tilde{v} = \tilde{\lambda}_1 \tilde{v}$. Hence $\beta_t \geq \left(1 + 2\eta_t \tilde{\lambda}_1\right)\beta_{t-1}$. With $\beta_0 = \|\tilde{v}\|_2^2 = 1$ and $1 + x \geq \exp\left(x - x^2\right)$ for $x \geq 0$,

$$
\beta_t \geq \exp\left(\sum_{i=1}^{t} \left(2\eta_i \tilde{\lambda}_1 - 4\eta_i^2 \tilde{\lambda}_1^2\right)\right). \qquad \square
$$

**Lemma 21.** $\mathbb{E}\left[\left(\tilde{v}^\top B_t B_t \tilde{v}\right)^2 \mid P\right] \leq \exp\left(\sum 4\eta_i \tilde{\lambda}_1 + 10\eta_i^2 \bar{\mathcal{V}}\right)$

*Proof.* We define $\gamma_s := \mathbb{E}[(\tilde{v}^\top W_{t,s} W_{t,s}^\top \tilde{v})^2 | P]$ where $W_{t,s} := (\mathbf{I} + \eta_t P A_i P) \cdot \ldots (\mathbf{I} + \eta_{t-s+1} P A_{t-s+1} P)$. So by this definition we see $W_{t,t} = B_t$ and $\gamma_t = \mathbb{E}[(\tilde{v}^\top B_t B_t^\top \tilde{v})^2 | P]$. As the trace of a scalar is the scalar itself, we can exploit the cyclic permutation properties of the trace:

$$\gamma_t = \text{Tr}(\mathbb{E}[W_{t,t}^\top \tilde{v}\tilde{v}^\top W_{t,t} W_{t,t}^\top \tilde{v}\tilde{v}^\top W_{t,t} | P])$$

$$= \text{Tr}(\mathbb{E}[(\mathbf{I} + \eta_1 A_1^\top) G_{t-1} (\mathbf{I} + \eta_1 A_1)(\mathbf{I} + \eta_1 A_1^\top) G_{t-1}(\mathbf{I} + \eta_1 A_1) | P])$$

where $G_{t-1} := W_{t,t-1}^\top v_1 v_1^\top W_{t,t-1}$. We first bound for an arbitrary $G_{t-1} = G$, and then take the expectation over only $A_1$ and finally over $G_{t-1}$.

$$\text{Tr}(\mathbb{E}[(\mathbf{I} + \eta_1 P A_1^\top P) G (\mathbf{I} + \eta_1 P A_1 P)(\mathbf{I} + \eta_1 P A_1^\top P) G (\mathbf{I} + \eta_1 P A_1 P) | P])$$

$$= \text{Tr}(\mathbb{E}[(G + \eta_1 P A_1^\top P G + \eta_1 G P A_1 P + \eta_1^2 P A_1^\top P G P A_1 P)^2 | P])$$

$$= \text{Tr}(G^2) + 4\eta_1 \text{Tr}(P\Sigma P G^2) + 2\eta_1^2 \text{Tr}(\mathbb{E}[P A_1 P A_1^\top P | P] G^2)$$

$$+ \eta_1^2 \text{Tr}(\mathbb{E}[P A_1^\top P G P A_1 P G | P]) + \eta_1^2 \text{Tr}(\mathbb{E}[P A_1^\top P G P A_1^\top P G | P])$$

$$+ \eta_1^2 \text{Tr}(\mathbb{E}[G P A_1 P G P A_1 P | P]) + \eta_1^2 \text{Tr}(\mathbb{E}[G P A_1^\top P G P A_1 P | P])$$

$$+ 4\eta_1^3 \text{Tr}(\mathbb{E}[P A_1^\top P G P A_1^\top P G P A_1 P | P])$$

$$+ \eta_1^4 \text{Tr}(\mathbb{E}[P A_1^\top P G P A_1 P A_1^\top P G P A_1 P | P]))$$

Let's begin with the first order term:

$$\text{Tr}(P\Sigma P G^2) \leq \|P\Sigma P\|_2 Tr(G^2) = \tilde{\lambda}_1 \text{Tr}(G^2)$$

then let's consider:

$$\text{Tr}(\mathbb{E}[P A_1 P A_1^\top P | P] G^2) \leq (\|\mathbb{E}[P(A_1 - \Sigma) P(A_1^\top - \Sigma) P]\|_2 + \|P\Sigma P\Sigma P\|_2)\text{Tr}(G^2) \leq (\mathcal{V} + \tilde{\lambda}_1^2)\text{Tr}(G^2)$$

where the last inequality follows by Lemma 11. Next we have 4 remaining second order terms:

$$\text{Tr}(\mathbb{E}[P A_1^\top P G P A_1 P G | P]) = \text{Tr}(\mathbb{E}[P A_1^\top P G P A_1^\top P G | P])$$

$$= \text{Tr}(\mathbb{E}[G P A_1 P G P A_1 P | P]) = \text{Tr}(\mathbb{E}[G P A_1^\top P G P A_1 P | P])$$

$$\leq \frac{1}{2}\mathbb{E}[\|P A_1^\top P G\|_F^2 + \|P A_1 P G\|_F^2 | P]$$

$$= \frac{1}{2}\text{Tr}(G\mathbb{E}[P A_1 P A_1^\top P | P] G + G\mathbb{E}[P A_1 P A_1^\top P | P] G)) \leq (\mathcal{V} + \tilde{\lambda}_1^2)\text{Tr}(G^2)$$

Third order terms we can bound as follows:

$$\text{Tr}(\mathbb{E}[P A_1^\top P G P A_1^\top P G P A_1 P | P]) \leq \|P A_1^\top P\|\text{Tr}(\mathbb{E}[P A_1^\top P G G P A_1 P) | P]$$

$$\leq (\|P(A_1 - \Sigma) P\|_2 + \|P\Sigma P\|_2)\text{Tr}(G\mathbb{E}[P A_1 P A_1^\top P | P] G)$$

$$\leq (\mathcal{M} + \tilde{\lambda}_1)(\mathcal{V} + \tilde{\lambda}_1)\text{Tr}(G^2)$$

Finally the fourth order term

$$\text{Tr}(\mathbb{E}[P A_1^\top P G P A_1 P A_1^\top P G P A_1 P | P])) \leq \|\mathbb{E}[P A_1 P A_1^\top P]\|_2 \text{Tr}(G\mathbb{E}[P A_1 P A_1^\top P | P] G)$$

$$\leq (\mathcal{M} + \tilde{\lambda}_1)^2 (\mathcal{V} + \tilde{\lambda}_1)\text{Tr}(G^2)$$

all of this together gives us

$$\text{Tr}(\mathbb{E}[(\mathbf{I} + \eta_1 P A_1^\top P) G (\mathbf{I} + \eta_1 P A_1 P)(\mathbf{I} + \eta_1 P A_1^\top P) G (\mathbf{I} + \eta_1 P A_1 P) | P])$$

$$\leq \text{Tr}(G^2) + 4\eta_1 \tilde{\lambda}_1 \text{Tr}(G^2) + 5\eta_1^2 \bar{\mathcal{V}}\text{Tr}(G^2) + 4\eta_1^3(\mathcal{M} + \tilde{\lambda}_1)\bar{\mathcal{V}}\text{Tr}(G^2) + \eta_1^4(\mathcal{M} + \tilde{\lambda}_1)^2 \bar{\mathcal{V}}\text{Tr}(G^2)$$

$$= (1 + 4\eta_1 \tilde{\lambda}_1 + 5\eta_1^2 \bar{\mathcal{V}} + 4\eta_1^3(\mathcal{M} + \tilde{\lambda}_1)\bar{\mathcal{V}} + \eta_1^4(\mathcal{M} + \tilde{\lambda}_1)^2 \bar{\mathcal{V}})\text{Tr}(G^2)$$

$$\leq (1 + 4\eta_1 \tilde{\lambda}_1 + 10\eta_1^2 \bar{\mathcal{V}})\text{Tr}(G^2)$$

$$\leq \exp(4\eta_1 \tilde{\lambda}_1 + 10\eta_1^2 \bar{\mathcal{V}})\text{Tr}(G^2)$$

where we used $\eta_i \leq \frac{1}{4\max\{\lambda_1, \mathcal{M}\}}$ and $1 + x \leq \exp(x)$. All of this give us

$$\gamma_t \leq \exp(4\eta_1 \tilde{\lambda}_1 + 10\eta_1^2 \bar{\mathcal{V}})\mathbb{E}[\text{Tr}(G_{t-1}^2) | P] = \exp(4\eta_1 \tilde{\lambda}_1 + 10\eta_1^2 \bar{\mathcal{V}})\gamma_{t-1}$$

then using $\gamma_0 = 1$ gives us the wished result. □

**Lemma 22.**

$$\mathbb{E}\left[\mathrm{Tr}(\tilde{V}_\perp^\top B_t B_t^\top \tilde{V}_\perp)|P] \le \exp\left(\sum_{j=1}^{t} 2\eta_j \tilde{\lambda}_2 + \eta_j^2 \bar{\mathcal{V}}\right)\left(d + \sum_{i\in[t]} \eta_i^2 \mathcal{V}\exp\left(\sum_{j\in[i]} 2\eta_j\left(\tilde{\lambda}_1 - \tilde{\lambda}_2\right)\right)\right)\right]$$

*Proof.* Let $\alpha_t := \mathbb{E}\left[\mathrm{Tr}\left(\tilde{V}_\perp^\top B_t B_t^\top \tilde{V}_\perp\right) \mid P\right]$. By cyclicity of trace and $\tilde{V}_\perp$ being fixed under $\mathbb{E}\left[\cdot \mid P\right]$,

$$\alpha_t = \left\langle \mathbb{E}\left[B_t B_t^\top \mid P\right], \tilde{V}_\perp \tilde{V}_\perp^\top \right\rangle$$
$$= \left\langle \mathbb{E}\left[B_{t-1} B_{t-1}^\top \mid P\right], \mathbb{E}\left[(\mathbf{I} + \eta_t P A_t P)\tilde{V}_\perp \tilde{V}_\perp^\top (\mathbf{I} + \eta_t P A_t P)^\top \mid P\right]\right\rangle.$$

For the right-hand matrix,

$$\tilde{V}_\perp \tilde{V}_\perp^\top + \eta_t P\Sigma P\,\tilde{V}_\perp \tilde{V}_\perp^\top + \eta_t \tilde{V}_\perp \tilde{V}_\perp^\top P\Sigma P + \eta_t^2 \mathbb{E}\left[P A_t P\,\tilde{V}_\perp \tilde{V}_\perp^\top\, P A_t P \mid P\right]$$
$$\preceq \left(1 + 2\eta_t \tilde{\lambda}_2 + \eta_t^2 \bar{\mathcal{V}}\right)\tilde{V}_\perp \tilde{V}_\perp^\top + \eta_t^2 \mathcal{V}\,\tilde{v}\tilde{v}^\top,$$

using $\tilde{V}_\perp$ orthogonal to the top eigenvector $\tilde{v}$ of $P\Sigma P$, and $\tilde{V}_\perp \tilde{V}_\perp^\top \preceq \mathbf{I}$. Therefore,

$$\alpha_t \le \left(1 + 2\eta_t \tilde{\lambda}_2 + \eta_t^2 \bar{\mathcal{V}}\right)\alpha_{t-1} + \eta_t^2 \mathcal{V}\left\langle \mathbb{E}\left[B_{t-1} B_{t-1}^\top \mid P\right], \tilde{v}\tilde{v}^\top\right\rangle.$$

Using $1 + x \le \exp(x)$ and $\langle X, \tilde{v}\tilde{v}^\top\rangle \le \|X\|_2$,

$$\alpha_t \le \exp\left(2\eta_t \tilde{\lambda}_2 + \eta_t^2 \bar{\mathcal{V}}\right)\alpha_{t-1} + \eta_t^2 \mathcal{V}\left\|\mathbb{E}\left[B_{t-1} B_{t-1}^\top \mid P\right]\right\|_2$$
$$\le \exp\left(2\eta_t \tilde{\lambda}_2 + \eta_t^2 \bar{\mathcal{V}}\right)\alpha_{t-1} + \eta_t^2 \mathcal{V}\exp\left(\sum_{i=1}^{t-1}\left(2\eta_i \tilde{\lambda}_1 + \eta_i^2 \bar{\mathcal{V}}\right)\right),$$

by Lemma 19. Unrolling the recursion,

$$\alpha_t \le \exp\left(\sum_{j=1}^{t}\left(2\eta_j \tilde{\lambda}_2 + \eta_j^2 \bar{\mathcal{V}}\right)\right)\alpha_0 + \sum_{i=1}^{t} \eta_i^2 \mathcal{V}\exp\left(\sum_{j=1}^{i}\left(2\eta_j \tilde{\lambda}_1 + \eta_j^2 \bar{\mathcal{V}}\right)\right)\exp\left(\sum_{j=i+1}^{t}\left(2\eta_j \tilde{\lambda}_2 + \eta_j^2 \bar{\mathcal{V}}\right)\right)$$
$$= \exp\left(\sum_{j=1}^{t}\left(2\eta_j \tilde{\lambda}_2 + \eta_j^2 \bar{\mathcal{V}}\right)\right)\left(\alpha_0 + \sum_{i=1}^{t}\eta_i^2 \mathcal{V}\exp\left(\sum_{j=1}^{i} 2\eta_j\left(\tilde{\lambda}_1 - \tilde{\lambda}_2\right)\right)\right).$$

Since $\alpha_0 = \mathrm{Tr}\left(\tilde{V}_\perp^\top \tilde{V}_\perp\right) = d - 1 \le d$, the claim follows. $\qquad\square$

# E   Proof of Main Theorem

**Theorem 1** (Main Theorem). *Let $\varepsilon, \delta \in (0, 0.9)$ and $1 \le k < d$. Then k-DP-PCA satisfies the following:*

**Privacy:** *For any input sequence $\{A_i \in \mathbb{R}^{d\times d}\}$, the algorithm is $(\varepsilon, \delta)$-differentially private.*

**Utility:** *Suppose $A_1, \ldots, A_n$ are i.i.d. satisfying Assumption A with parameters $(\Sigma, M, V, K, \kappa', a, \gamma^2)$. If*

$$n \gtrsim C\max\begin{cases} e^{\kappa'^2} + \dfrac{d\,\kappa'\gamma\,\sqrt{\ln(1/\delta)}}{\varepsilon} + \kappa' M + \kappa'^2 V + \dfrac{\sqrt{d}\,(\ln(1/\delta))^{3/2}}{\varepsilon}, \\ \lambda_1^2\,\kappa'^2\,k^3\,V, \\ \dfrac{\kappa'^2\,\gamma\,k^2\,d\,\sqrt{\ln(1/\delta)}}{\varepsilon} \end{cases} \qquad, \qquad (1)$$

*for a sufficiently large constant C, then with probability at least 0.99, the output $U \in \mathbb{R}^{d \times k}$ is $\zeta$–approximate with*

$$\zeta = \tilde{O}\left(\kappa'\left(\sqrt{\frac{Vk}{n}} + \frac{\gamma dk\sqrt{\log(1/\delta)}}{\varepsilon n}\right)\right), \tag{2}$$

*where $\tilde{O}(\cdot)$ hides factors polylogarithmic in $n, d, 1/\varepsilon, \ln(1/\delta)$ and polynomial in $K$.*

*Proof of Theorem 1.* The privacy proof of Algorithm 1 follows straight away from using Advanced Composition (Lemma 16) together with the privacy of MODIFIEDDP-PCA, which in turn follows by [Liu et al., 2022a]. For the utility proof we note that by Theorem 9 we know that when passing $m = n/k$ matrices $A_i$ at every step of our deflation method we obtain a vector $u_i$ fulfilling

$$\sin(u_i, v_i) \leq \tilde{O}\left(\frac{\lambda_1(P\Sigma P)}{\lambda_1(P\Sigma P) - \lambda_2(P\Sigma P)}\left(\sqrt{\frac{Vk}{n}} + \frac{\gamma dk\sqrt{\log(1/\delta)}}{\varepsilon n}\right)\right)$$

where $v_i$ is the top eigenvector of $P_{i-1}\Sigma P_{i-1}$. Which by Lemma 17 give us

$$\langle u_i u_i^\top, P_{i-1}\Sigma P_{i-1}\rangle \geq (1 - \zeta_i^2)\langle v_i v_i^\top, P_{i-1}\Sigma P_{i-1}\rangle$$

with $\zeta_i = \tilde{O}\left(\frac{\lambda_1(P\Sigma P)}{\lambda_1(P\Sigma P) - \lambda_2(P\Sigma P)}\left(\sqrt{\frac{Vk}{n}} + \frac{\gamma dk\sqrt{\log(1/\delta)}}{\varepsilon n}\right)\right)$. By our choice of $n$ we know by Lemma 23 that

$$\zeta_i \leq \tilde{O}\left(\frac{\lambda_1}{\Delta}\left(\sqrt{\frac{Vk}{n}} + \frac{\gamma dk\sqrt{\log(1/\delta)}}{\varepsilon n}\right)\right)$$

where we used that $(\Delta - \delta)\delta$ is maximized by $\delta = \Delta/2$. So finally Theorem 6 gives us that

$$\langle UU^\top, \Sigma\rangle \geq (1 - \zeta^2)\langle V_k V_k^\top, \Sigma\rangle \tag{22}$$

where $V_k$ is the matrix obtained by non private $k$-PCA. □

For the above utility proof we could not apply DP-PCA straight away, as this would only give us a guarantee that the vector $\tilde{v}$ we obtain is a good approximation of the top eigenvector of $\mathbb{E}[P]\Sigma\mathbb{E}[P]$. This is not sufficient for the deflation method, as we require $\tilde{v}$ to be a good approximation of $P\Sigma P$. We show that for MODIFIEDDP-PCA this is indeed the case in Theorem 9). We proof Theorem 9 by first showing that with high likelihood we can reduce the update step to an update step of non private Oja's Algorithm with matrices $PC_t P$. We then apply a novel result we establish in Appendix D, which shows that the non-private Oja's algorithm, when run on the projected matrices $\{PC_t P\}_t$, produces a good approximation of the top eigenvector of $P\mathbb{E}[C_t]P$, under certain assumptions on the sequence $\{C_t\}$. Lastly, we need to control the error we accumulate through approximate projections $P_i = I - \sum_{j=1}^{i} u_j u_j^\top$, which we do in Lemma 23.

**Theorem 9** (MODIFIEDDP-PCA). *Let $\varepsilon, \delta \in (0, 0.9)$, then*

**Privacy:** *For any input sequence $\{A_i \in \mathbb{R}^{d \times d}\}$ and projection matrix $P$ independent of the $\{A_i\}$ the algorithm is $(\varepsilon, \delta)$-differentially private.*

**Utility:** *Suppose $A_1, \ldots, A_n$ are i.i.d. satisfying Assumption A.1–Assumption A.4 with parameters $(\Sigma, M, V, K, \kappa', a, \gamma^2)$, if*

$$n \gtrsim C \cdot \left(e^{\kappa'^2} + \frac{d\kappa'\gamma(\log(1/\delta))^{1/2}}{\varepsilon} + \kappa' M + \kappa'^2 V + \frac{d^{1/2}(\log(1/\delta))^{3/2}}{\varepsilon}\right)$$

*for a large enough constant C, where $\kappa' = \frac{\lambda_1(\Sigma)}{\lambda_1(P\Sigma P) - \lambda_2(P\Sigma P)}$, $\delta \leq 1/n$ and*

$$0 < \lambda_1(P\Sigma P) - \lambda_2(P\Sigma P)$$

*then there exists a learning rate $\eta_t$ that depends on $(t, M, V, K, a, \lambda_1(\Sigma), \lambda_1(P\Sigma P) - \lambda_2(P\Sigma P), n, d\varepsilon, \delta)$ such that $T = \lfloor n/B \rfloor$ steps of ModifiedDP-PCA with choices of $\tau = 0.01$ and $B = c_1 n/(\log n)^3$ output $\omega_T$ that with probability 0.99 fulfills*

$$\sin(\omega_t, \tilde{v}) \leq \tilde{O}\left(\kappa'\left(\sqrt{\frac{V}{n}} + \frac{\gamma d \sqrt{\log(1/\delta)}}{\varepsilon n}\right)\right) \tag{23}$$

*where $\tilde{v}$ is the top eigenvector of $P\Sigma P$ and $\tilde{O}(\cdot)$ hides poly-logarithmic factors in $n, d, 1/\varepsilon$, and $\log(1/\delta)$ and polynomial factors in $K$.*

*Remark.* For readability we omitted the advanced composition details. If we choose $T = O(log^2 n)$, we can simply set $(\varepsilon', \delta') = (\varepsilon/(2\sqrt{2\log^2(n)log(2/\delta)}), \delta/(2\log^2(n)))$ in every step and then by advanced composition we get. And in our utility guarantee we would only occur additional $\log^2(n)$ factors which we omit. We also want to comment on why the utility bound only depends on $P$ in the parameter $\kappa'$: We can see the the utility bound of MODIFIEDDP-PCA depends on several constants originating from constraints on the data:

1. $\kappa = \frac{\lambda_1}{\lambda_1 - \lambda_2}$

2. $M$ so that $\|A_i - \Sigma\|_2 \leq \lambda_1 M$ almost surely

3. $V$ so that $\max\{\|\mathbb{E}[(A_i - \Sigma)(A_i - \Sigma)^\top]\|_2, \|, \|\mathbb{E}[(A_i - \Sigma)^\top(A_i - \Sigma)]\|_2\} \leq \lambda_1^2 V$

4. $\gamma^2 := \max_{\|u\|=1} \|H_u\|_2$

5. $K$ so that $\max_{\|u\|=1, \|v\|=1} \mathbb{E}\left[\exp\left((\frac{|u^\top(A_i^\top - \Sigma)v|^2}{K^2\lambda_1^2\|H_u\|_2})^{1/(2a)}\right)\right] \leq 1$

now if we replace $\{A_i\}$, the input to MODIFIEDDP-PCA, with $\{PA_iP\}$ (which is exactly what happens at iteration $i$ of Algorithm 1) where $P$ is a projection matrix, the constants $M, V, \lambda_1^2\gamma^2$ and $K$ will still remain upper bounds (see Lemma 10, Lemma 11, Lemma 13).

*Proof.* We choose the batch size $B = \Theta(n/\log^3 n)$ such that we access the dataset only $T = \Theta(\log^3 n)$ times. Hence we do not need to rely on amplification by shuffling. To add Gaussian noise that scales as the standard deviation of the gradients in each minibatch (as opposed to potentially excessively large mean of the gradients), DP-PCA first gets a private and accurate estimate of the range. Using this estimate PRIVMEAN returns an unbiased estimate of the empirical mean of the gradients, as long as no truncation has been applied. As we choose the truncation threshold so that with high probability there will be no truncation the update step will look as follows:

$$\omega_t' \leftarrow \omega_{t-1} + \eta_t P(\frac{1}{B}\sum_{i \in [B]} PA_iP\omega_{t-1} + \beta_t z_t)$$

where $z_t \sim \mathcal{N}(0, \mathbf{I})$ and $\beta_t = \frac{8K\sqrt{2\hat{\Lambda}_t}\log^a(Bd/\tau)\sqrt{2d\log(2.5/\delta)}}{\varepsilon B}$. The privacy follows by the privacy of the subroutines private eigenvalue and private mean estimation [Liu et al., 2022a]. So all that is left to do is show the utility guarantee. We will do that by showing we can reduce it the accuracy of the non private case. First we note that $P^2 = P$ so we get

$$\omega_t' = \omega_{t-1} + \eta_t(\frac{1}{B}\sum_{i in [B]} PA_iP\omega_{t-1} + \beta_t Pz_t)$$

Using rotation invariance of the spherical Gaussian random vectors and the fact that $\|\omega_{t-1}\| = 1$ and $\omega_{t-1} \in \text{Im}(P)$ (for details see Lemma 9), we can reformulate it as

$$\omega_t' \leftarrow \omega_{t-1} + \eta_t \left(\frac{1}{B}\sum_{i \in [B]} PA_iP + \beta_t PG_tP\right)\omega_{t-1}$$

we can further pull out the projection matrices to obtain

$$\omega_t' \leftarrow \omega_{t-1} + \eta_t P\left(\frac{1}{B}\sum_{i \in [B]} A_i + \beta_t G_t\right)P\omega_{t-1}$$

Where $G$ is a matrix whose entries are i.i.d. $\mathcal{N}(0,1)$ distributed. So we have a matrix

$$C_t := \frac{1}{B} \sum_{i \in [B]} A_i + \beta_t G_t$$

and we will now proof that $C_t$ fulfills all requirements for Theorem 7 (our version of the non private Oja's Algorithm utility guarantee), which will directly give us the wished utility guarantee. It is easy to see that $\mathbb{E}[C_t] = \Sigma$ as $z$ is a zero mean random variable and hence so is $G_t$. Next we show the upper bound of $\max\left\{ \left\| \mathbb{E}\left[(C_t - \Sigma)(C_t - \Sigma)^\top\right]\right\|_2, \left\| \mathbb{E}\left[(C_t - \Sigma)^\top (C_t - \Sigma)\right]\right\|_2 \right\}$

$$\left\| \mathbb{E}\left[(C_t - \Sigma)(C_t - \Sigma)^\top\right]\right\|_2 = \left\| \mathbb{E}\left[ \left(\frac{1}{B}\sum_{i \in [B]} A_i + \beta_t G_t - \Sigma\right) \left(\frac{1}{B}\sum_{i \in [B]} A_i + \beta_t G_t - \Sigma\right)^\top \right]\right\|_2$$

$$\leq \left\| \mathbb{E}\left[ \left(\frac{1}{B}\sum_{i \in [B]} A_i - \Sigma\right) \left(\frac{1}{B}\sum_{i \in [B]} A_i - \Sigma\right)^\top \right]\right\|_2 + \beta_t^2 \left\| \mathbb{E}\left[G_t G_t^\top\right]\right\|_2$$

$$\leq \frac{V\lambda_1^2}{B} + \beta_t^2 \left\| \mathbb{E}\left[G_t G_t^\top\right]\right\|_2$$

$$\leq \frac{V\lambda_1^2}{B} + \beta^2 C_2 d =: \tilde{V}$$

where the first inequality holds due to $G_t$ being independent to $A_i$, and $\mathbb{E}[G_t] = 0$. The second inequality follows due to having $B$ elements of $\frac{1}{B^2}\left\| \mathbb{E}\left[(A_i - \Sigma)^\top (A_i - \Sigma)\right]\right\|_2$ and Assumption 3. And the last inequality holds with high probability due to $G_t$ having i.i.d. Gaussian entries (Lemma 5), and by choosing

$$\beta := \frac{16K\gamma\lambda_1 \log^a(Bd/\tau)\sqrt{2d\log(2.5/\delta)}}{\varepsilon B}$$

we have $\beta \geq \beta_t$ for all $t$ as by Theorem 6.1 in [Liu et al., 2022a] and Assumption 4

$$\hat{\Lambda} \leq \sqrt{2}\lambda_1^2 \|H_u\|_2 \leq \sqrt{2}\lambda_1^2 \gamma$$

Lastly let us consider $\|C_t - \Sigma\|_2$. By Lemma 3 and Lemma 4 we know with proabability $1 - \tau$ for all $t \in [T]$

$$\|C_t - \Sigma\|_2 = \left\| \frac{1}{B}\sum_{i \in [B]} A_i + \beta_t G_t - \Sigma \right\|$$

$$\leq \left( \frac{M\lambda_1 \log\left(dT/\tau\right)}{B} + \sqrt{\frac{V\lambda_1^2 \log\left(dT/\tau\right)}{B}} + \beta\left(\sqrt{d} + \sqrt{\log\left(T/\tau\right)}\right) \right) =: \tilde{M}$$

so by Theorem 7 with stepsize $\eta_t := \frac{\alpha}{(\lambda_1 - \lambda_2)(\xi + t)}$ after $T$ steps with

$$T \geq 20 \max\left( \frac{\tilde{M}\alpha}{\left(\tilde{\lambda}_1 - \tilde{\lambda}_2\right)}, \frac{\left(\tilde{V} + \lambda_1^2\right)\alpha^2}{\left(\tilde{\lambda}_1 - \tilde{\lambda}_2\right)^2 \log\left(1 + \frac{\zeta}{100}\right)} \right) := \xi \tag{24}$$

with probability $1 - \zeta$

$$\sin^2(w_T, \tilde{v}) \leq \frac{C\log(1/\delta)}{\delta^2}\left( d\left(\frac{\xi}{T}\right)^{2\alpha} + \frac{\alpha^2 \tilde{V}}{(2\alpha - 1)(\tilde{\lambda}_1 - \tilde{\lambda}_2)^2 T} \right)$$

so if we fill in $\tilde{M}$, $\tilde{V}$, and $\beta$ into $\xi$ and use $n = BT$ we get

$$\frac{\xi}{T} := 20 \max \begin{cases} \frac{\lambda_1 M \log(dT/\tau\alpha)}{(\tilde{\lambda}_1 - \tilde{\lambda}_2)n} + \sqrt{\frac{V\log(dT/\tau)}{nT}} \cdot \frac{\lambda_1\alpha}{(\tilde{\lambda}_1 - \tilde{\lambda}_2)} + \frac{K\gamma\lambda_1 \log^a(nd/T\tau\sqrt{2\log(2.5/\delta)}\sqrt{\log(T/\tau}d\alpha}{\varepsilon n(\tilde{\lambda}_1 - \tilde{\lambda}_2)}, \\ \frac{V\lambda_1^2\alpha^2}{n(\tilde{\lambda}_1 - \tilde{\lambda}_2)^2 \log(1 + \frac{\zeta}{100})} + \frac{K^2\gamma^2\lambda_1^2 \log^{2a}(Bd/\tau d^2 \log(2.5/\delta)\alpha^2}{\varepsilon^2 n^2(\tilde{\lambda}_1 - \tilde{\lambda}_2)^2 \log(1 + \frac{\zeta}{100})} + \frac{\lambda_1^2\alpha^2}{(\tilde{\lambda}_1 - \tilde{\lambda}_2)^2 \log(1 + \frac{\zeta}{100})T} \end{cases}$$

in order for Theorem 7 to hold we need to force $\xi/T \leq 1$. Noting $\tau = O(1)$, $K = O(1)$ and selecting $\alpha = c \log n$, $T = c'(\log n)^3$ we get that

$$\frac{\xi}{T} \leq 20C \max \left\{ \begin{array}{l} \frac{\lambda_1 M \log(d \log(n)) \log n}{(\tilde{\lambda}_1 - \tilde{\lambda}_2) n} + \sqrt{\frac{V \log(d \log(n))}{n}} \cdot \frac{\lambda_1}{(\tilde{\lambda}_1 - \tilde{\lambda}_2)} + \frac{\gamma \lambda_1 \log^2(nd/\log(n)) \sqrt{\log(1/\delta) \log(\log(n))} \log(n) d}{\varepsilon(\tilde{\lambda}_1 - \tilde{\lambda}_2)} \\ \frac{V \lambda_1^2 (\log n)^2}{n(\tilde{\lambda}_1 - \tilde{\lambda}_2)} + \frac{\gamma^2 \lambda_1^2 \log^{2a}(nd/\log(n)) \log(1/\delta) d^2 \alpha^2}{\varepsilon^2 n^2 (\tilde{\lambda}_1 - \tilde{\lambda}_2)^2} + \frac{\lambda_1^2 (\log n)^2}{(\tilde{\lambda}_1 - \tilde{\lambda}_2)^2 T} \end{array} \right.$$

so $\frac{\xi}{T} \leq 1$ will be trivially fulfilled if each of the summand is smaller than $1/3$. For the last term we need

$$\frac{\lambda_1^2 (\log n)^2}{(\tilde{\lambda}_1 - \tilde{\lambda}_2)^2 T} \leq 1/3 \tag{25}$$

as $T = c'(\log(n))^3$ this means

$$\log n \geq 3 \frac{\lambda_1}{(\tilde{\lambda}_1 - \tilde{\lambda}_2)^2}$$

for the remaining terms we need

$$\frac{n}{\log^a(n/\log n) \log(n)} \geq 3 \frac{\gamma \lambda_1 \sqrt{\log(1/\delta)} d}{\varepsilon(\tilde{\lambda}_1 - \tilde{\lambda}_2)}$$

$$\frac{n}{(\log(n))^2} \geq 3 \frac{V \lambda_1^2}{(\tilde{\lambda}_1 - \tilde{\lambda}_2)^2}$$

$$\frac{n}{\log(\log(n))} \geq \sqrt{3} \sqrt{V \log(d)}$$

$$\frac{n}{\log(n) \log(\log(n))} \geq 3 \frac{\lambda_1 M \log(d)}{(\tilde{\lambda}_1 - \tilde{\lambda}_2)}$$

We note that to obtain $n/log(n) \geq a$, $n \simeq a \log(a) + a \log \log(a)$. So

$$n \gtrsim C' \left( \exp(\lambda_1^2/(\tilde{\lambda}_1 - \tilde{\lambda}_2)^2) + \frac{M \lambda_1}{(\tilde{\lambda}_1 - \tilde{\lambda}_2)} + \frac{V \lambda_1^2}{(\tilde{\lambda}_1 - \tilde{\lambda}_2)^2} + \frac{d \gamma \lambda_1 \sqrt{\log(1/\delta)}}{(\tilde{\lambda}_1 - \tilde{\lambda}_2)\varepsilon} \right)$$

with large enough constant suffices (where $\gtrsim$ is hiding log terms) to obtain $\xi/T \leq 1$ and $d(\xi/T)^{2\alpha} \leq 1/n^2$. And we get

$$\frac{\tilde{V}}{(\tilde{\lambda}_1 - \tilde{\lambda}_2)} \lesssim C'' \left( \frac{V \lambda_1^2}{n} + \frac{\gamma^2 \lambda_1^2 d^2 \log(1/\delta)}{\varepsilon n} \right)$$

(where $\lesssim$ is hiding log terms), so plugging this in our bound for $\sin(\omega_T, \tilde{v})$ we get

$$\sin(\omega_T, \tilde{v}) \leq \tilde{O} \left( \kappa' \left( \sqrt{\frac{V}{n}} + \frac{\gamma d \sqrt{\log(1/\delta)}}{\varepsilon n} \right) \right)$$

which finishes the proof. $\qquad \square$

**Lemma 23.** *If we are given matrices $\{A_i \in \mathbb{R}^{d \times d}\}_{i=1}^n$ fulfilling Assumption A with parameters $(\Sigma, M, V, K, \kappa', a, \gamma^2)$, a fixed $k \leq d$, $0 < \Delta = \min_{i \in [k]} \lambda_i - \lambda_{i+1}$ where $\lambda_i$ refers to the ith eigenvalue of $\Sigma$, $\delta$ so that $0 < \delta < \Delta$, and a sufficiently large constant $C > 1$ so that*

$$B_{n/k} \leq \frac{(\Delta - \delta)\delta}{Ck\lambda_1^2}$$

*then*

$$\xi_i \leq \frac{\lambda_1}{\delta} B_{n/k}$$

*where*

$$B_n = \tilde{O} \left( \sqrt{\frac{V}{n}} + \frac{\gamma d \sqrt{\log(1/\delta)}}{\varepsilon n} \right)$$

*and $\xi_i$ refers to the utility of the vector $u_i$ returned at iteration $i \in [k]$ of Algorithm 1.*

*Proof.* We will denote

$$\kappa_i := \frac{\lambda_1(P_{i-1}\Sigma P_{i-1})}{\lambda_1(P_{i-1}\Sigma P_{i-1}) - \lambda_2(P_{i-1}\Sigma P_{i-1})}.$$

Then by Theorem 9 we obtain

$$\xi_i \leq \kappa_i \cdot B_n$$

Lemma 25 will give us a utility bound independent of $P$ for $k = 2$, as it bounds $\kappa_2$. However, we want to obtain a utility guarantee for arbitrary $k < d$, so the goal is to upper bound $\kappa_i$ for general $i$.

If we iteratively apply Lemma 25 we get

$$\kappa_i \leq \frac{\lambda_i(\Sigma) + \sum_{j=1}^{i-1}\Delta_j}{\lambda_i(\Sigma) - \lambda_{i+1}(\Sigma) - 2\sum_{j=1}^{i-1}\Delta_j}$$

where $\Delta_j = c\lambda_1(P_{j-1}\Sigma P_{j-1})\xi_j$ ($\Delta_0 := 0$ for completeness). Now the problem is that $\Delta_j$ still depends on previous projections and it's not even clear in general if $\xi_j > \xi_{j+1}$ or the other way around. Ultimately we want to have an upper bound for all $\xi_j$, to get a utility bound for $U = \{u_i\}$. A natural approach is to try and choose $n$ big enough so that

$$\lambda_1(P_i\Sigma P_i) \leq \lambda_1 \tag{26}$$
$$\lambda_1(P_i\Sigma P_i) - \lambda_2(P_i\Sigma P_i) \geq \delta \tag{27}$$

for some $\delta > 0$. If we achieve this we are done, as this will guarantee that

$$\xi_i \leq \frac{\lambda_1}{\delta} B_n$$

We will proof that at every step Equation (26) and Equation (27) are fulfilled by induction. For $k = 1$ we have $P_0 = \mathbf{I}$ which straightaway gives us equation 26. And as $\delta$ is smaller then the minium eigengap equation 27, directly follows as well. For $k + 1$ we start with showing equation 26. By Lemma 24

$$\lambda_1(P_k\Sigma P_k) \leq \lambda_{k+1}(\Sigma) + \sum_{j=1}^{k}\Delta_j$$

first let's upper bound $\sum_{j=1}^{k}\Delta_j$. By definition we have

$$\Delta_j = c \cdot \lambda_1(P_{j-1}\Sigma P_{j-1})\xi_j$$

for some constant $c$. By induction assumption this gives us :

$$\sum_{j=1}^{k}\Delta_j = \sum_{j=1}^{k} c\frac{\lambda_1^2(P_{j-1}\Sigma P_{j-1})}{\lambda_1(P_{j-1}\Sigma P_{j-1}) - \lambda_2(P_{j-1}\Sigma P_{j-1})} \cdot B_n$$

$$\leq cB_n \cdot \sum_{j=1}^{k} \frac{\lambda_1^2}{\delta}$$

so equation 26 will be implied by

$$B_n \leq (\lambda_1 - \lambda_{k+1}) \cdot \frac{\delta}{ck\lambda_1^2}$$

which is surely fulfilled as by assumption

$$B_n \leq \frac{(\Delta - \delta)\delta}{ck\lambda_1^2}.$$

To show equation 27, we see that

$$\lambda_1(P_k\Sigma P_k) - \lambda_2(P_k\Sigma P_k) \geq \lambda_{k+1}(\Sigma) - \lambda_{k+2}(\Sigma) - 2\sum_{j=1}^{k}\Delta_j$$

$$\geq \Delta - 2\sum_{j=1}^{k}\Delta_j$$

where the first inequality follows by Lemma 25 and the second by definition of $\Delta := \min_{i \in [k]} \lambda_i - \lambda_{i+1}$. Using the upper bound on $\sum_{j=1}^{k} \Delta_j$ we established before we obtain

$$\lambda_1(P_k \Sigma P_k) - \lambda_2(P_k \Sigma P_k) \geq \Delta - 2c \frac{k B_n \lambda_1^2}{\delta}$$

so if we choose

$$B_n \leq \frac{(\Delta - \delta)\delta}{ck\lambda_1^2}$$

this shows equation 27 will be fulfilled. $\qquad \square$

We need Lemma 23 as the utility result for MODIFIEDDP-PCA depends on the eigenvalues of the input. After the first step of $k$-DP-PCA our input is of the form $PA_1P, \ldots, PA_nP$, so our utility bound depends on the eigenvalues of $P\Sigma P$. In general $\lambda_1(P\Sigma P) - \lambda_2(P\Sigma P)$ can be arbitrarily much smaller than the actual eigengap of $\Sigma$, and therefore it is not a sufficient utility bound as is, to proof Theorem 1. However, as we iteratively apply projection matrices of the form

$$P = I - uu^\top$$

where $u$ is a unit vector, and further $u$ is $\varepsilon$-close to the top eigenvector of the matrix we apply it to, we can actually relate the eigengap of $P\Sigma P$ to the one of $\Sigma$ using Weyl's Theorem.

**Lemma 24.** *Given* $\sin^2(\theta) \leq \xi$, *where* $\theta$ *refers to the angle between* $v_1$, *the top eigenvector of* $\Sigma$ *(psd), and the unit vector* $u$, *then we have*

$$\tilde{\lambda}_i \geq \lambda_{i+1} - \Delta$$
$$\tilde{\lambda}_i \leq \lambda_{i+1} + \Delta$$

*where* $\tilde{\lambda}_i$ *is the ith eigenvector of* $P\Sigma P$, *with* $P = \mathbf{I}_d - uu^\top$, $\lambda_i$ *the ith eigenvector of* $\Sigma$, *and* $\Delta = 8\lambda_1 \sqrt{\xi}(1 + \sqrt{\xi})$

*Proof.* We will use Weyl's Theorem (Lemma 6) to proof this, by defining

$$G_1 = (\mathbf{I} - v_1 v_1^\top)\Sigma(\mathbf{I} - v_1 v_1^\top)$$
$$G_2 = (\mathbf{I} - uu^\top)\Sigma(\mathbf{I} - uu^\top)$$

then for $\mu_i$ the eigenvalues of $G_1$, and $\nu_i$ the eigenvalues of $G_2$ we know $\lambda_2 = \mu_1, \lambda_3 = \mu_2, \ldots$ and $\tilde{\lambda}_1 = \nu_1, \tilde{\lambda}_2 = \nu_2, \ldots$ etc. Now we can use this as follows:

$$\tilde{\lambda}_i = \lambda_{i-1} + (\tilde{\lambda}_i - \lambda_{i-1})$$
$$\leq \lambda_{i-1} + |\tilde{\lambda}_i - \lambda_{i-1}|$$
$$\leq \lambda_{i-1} + \|G_1 - G_2\|$$

where the last inequality follows by Weyl's Theorem. Next we will bound $\|G_1 - G_2\|$

$$\|G_1 - G_2\| = \|(v_1 v_1^\top \Sigma - uu^\top \Sigma) + (\Sigma v_1 v_1^\top - \Sigma uu^\top) + (uu^\top \Sigma uu^\top - v_1 v_1^\top \Sigma v_1 v_1^\top)\|$$
$$\leq 4\|v_1 v_1^\top - uu^\top\|_2 \|\Sigma\|_2$$

where the last step follows as $(uu^\top \Sigma uu^\top - v_1 v_1^\top \Sigma v_1 v_1^\top = (uu^\top - v_1 v_1^\top)\Sigma uu^\top + v_1 v_1^\top \Sigma(uu^\top - v_1 v_1^\top$ and $\|v_1 v_1^\top\|_2 = \|uu^\top\|_2 = 1$. Further it turns out that we can bound $\|v_1 v_1^\top - uu^\top\|_2$ using $\sin^2(v_1, u) \leq \xi$: First we note that as $u$ and $v_1$ are unit vectors we can write

$$u = \cos \theta v_1 + \sin \theta v_1^\perp$$

so this means

$$uu^\top = \cos^2 \theta v_1 v_1^\top + \cos \theta (v_1 v_1^{\perp \top} + v_1^\perp v_1^\top) + \sin^2 \theta v_1^\perp v_1^{\perp \top}$$

and also gives us

$$\|uu^\top - v_1 v_1^\top\|_2 = \|(\cos^2 \theta - 1) v_1 v_1^\top + \cos \theta \sin \theta (v_1 v_1^{\perp \top} + v_1^\perp v_1^\top) + \sin^2 \theta v_1^\perp v_1^{\perp \top}\|_2$$
$$= \| - \sin^2 \theta v_1 v_1^\top + \cos \theta (v_1 v_1^{\perp \top} + v_1^\perp v_1^\top) + \sin^2 \theta v_1^\perp v_1^{\perp \top}\|_2$$
$$\leq |\sin^2 \theta| \|v_1 v_1^\top\| + |\cos \theta \sin \theta| \|v_1 v_1^{\perp \top} + v_1^\perp v_1^\top\|_2 + |\sin^2 \theta| \|v_1^\perp v_1^{\perp \top}\|_2$$
$$\leq 2|\sin^2 \theta| + 2|\sin \theta| \leq 2\sqrt{\xi}(1 + \sqrt{\xi})$$

$\qquad \square$

We can now use Lemma 24 to lowerbound the eigengap of $P\Sigma P$.

**Lemma 25.** *For $\Sigma \in \mathbb{R}^{d \times d}$ a matrix with eigenvalues $\lambda_1 \geq \lambda_2 \geq \cdots \geq \lambda_d$, $P = I - uu^\top$, with $u \in Im(\Sigma)$, and $\tilde{\lambda}_1 \geq \tilde{\lambda}_2 \geq \cdots \geq \tilde{\lambda}_{d-1}$ the eigenvalues of $P\Sigma P$*

$$\tilde{\lambda}_1 - \tilde{\lambda}_2 \geq \lambda_2 - \lambda_3 - 2\Delta$$

*where $\Delta = 8\lambda_1\sqrt{\xi}(1 + \sqrt{\xi})$ and $\xi \geq \sin^2(\theta)$ with $\theta$ the angle between $u$ and $v_1$, the top eigenvector of $\Sigma$.*

### E.1   Proof of Utility of DP-Ojas

**Theorem 3.** *Given $A_1, \ldots, A_n$ are i.i.d. and satisfy Assumption A,* MODIFIEDDP-PCA *and DP-Ojas as defined Algorithms 2 and 3 are stochastic ePCA oracles with $\zeta = \tilde{O}\left( \kappa' \left( \sqrt{\frac{V}{n}} + \frac{\gamma d\sqrt{\log(1/\delta)}}{\varepsilon n} \right) \right)$ and $\zeta = \tilde{O}\left( \kappa' \left( \sqrt{\frac{V}{n}} + \frac{(\gamma+1)d\sqrt{\log(1/\delta)}}{\varepsilon n} \right) \right)$ respectively.*

*Proof.* The proof follows by the utility proofs of MODIFIEDDP-PCA and DP-Ojas (Theorem 9 and Theorem 10) and Lemma 17. $\qquad\square$

**Theorem 10** (DP-Ojas). *Privacy: If $\varepsilon = O(\sqrt{\log(n/\delta)/n})$ then Algorithm 3 is $(\varepsilon, \delta)$-DP.*

*Utility:* *Given $n$ i.i.d. samples $\{A_i \in \mathbb{R}^{d \times d}\}_{i=1}^n$ satisfying Assumption A with parameters $(\Sigma, M, V, K, \kappa', a, \gamma^2)$, if*

$$n \gtrsim C \cdot \left( \kappa'^2 + \kappa M + \kappa'^2 V + \frac{d\kappa'(\gamma + 1)\log(1/\delta)}{\varepsilon} \right)$$

*with a large enough constant $C$, then there exists a choice of learning rate $\eta_t$ such that Algorithm 3 with a choice of $\zeta = 0.01$ outputs $w_n$ that with probability 0.99 fulfills*

$$\sin(w_n, v_1) \leq \tilde{O}\left( \kappa' \left( \sqrt{\frac{V}{n}} + \frac{(\gamma + 1)d\log(1/\delta)}{\varepsilon n} \right) \right)$$

*where $\kappa' = \frac{\lambda_1(\Sigma)}{\lambda_1(P\Sigma P) - \lambda_2(P\Sigma P)}$ and $\tilde{O}(\cdot)$ hides poly-logarithmic factors in $n, d, 1/\varepsilon$, and $\log(1/\delta)$ and polynomial factors in $K$.*

*Proof.* **Privacy:** The privacy proof follows by Lemma 3.1 in [Liu et al., 2022b].
**Utility:** By Assumption A.4 it follows analogously to Lemma 3.2 in [Liu et al., 2022b] that with probability $1 - O(\zeta)$ Algorithm 3 does not have any clipping. Under this event, the update rule becomes

$$w_t' \leftarrow w_{t-1} + \eta_t P(PA_t Pw_{t-1} + 2\beta\alpha z_t)$$
$$w_t \leftarrow Pw_t'/\|Pw_t'\|$$

where $\beta = C\lambda_1\sqrt{d}(K\gamma\log^2(nd/\zeta) + 1)$ and $z_t \sim \mathcal{N}(0, \mathbf{I})$. Just like in the proof of Theorem 9 we use that $P^2 = P$ and Lemma 9 to rewrite this as

$$w_t' \leftarrow w_{t-1} + \eta_t P \left( A_t + 2\beta\alpha G_t \right) Pw_{t-1}$$

where $G$ is a matrix whose entries are i.i.d. $\mathcal{N}(0, 1)$ distributed. So if we define

$$\tilde{A}_t := A_t + 2\beta\alpha G_t$$

this becomes

$$w_t' \leftarrow w_{t-1} + \eta_t P\tilde{A}_t Pw_{t-1}$$

so if we can show the $\tilde{A}_t$'s fulfill all requirements for Theorem 7, we will directly obtain the wished utility guarantee. Equivalently to the proof of Theorem 9 we can show

$$\|\mathbb{E}[(\tilde{A}_t - \Sigma)(\tilde{A}_t - \Sigma)^\top]\|_2 \leq V\lambda_1^2 + 4\alpha^2\beta^2 C_2 d =: \tilde{V}$$
$$\|\tilde{A}_t - \Sigma\|_2 \leq M\lambda_1 + 2C_3\alpha\beta(\sqrt{d} + \sqrt{\log(n/\zeta)}) =: \tilde{V}$$

Under the event that $\|\tilde{A}_t - \Sigma\|_2 \leq \tilde{M}$ for all $t \in [n]$, we apply Theorem 7 with a learning rate $\eta_t = \frac{h}{(\lambda_1 - \lambda_2)(\xi + t)}$ where

$$\xi = 20 \max\left( \frac{\tilde{M}h}{(\lambda_1 - \lambda_2)}, \frac{(\tilde{V} + \lambda_1)^2 h^2}{(\lambda_1 - \lambda_2)^2 \log(1 + \frac{\zeta}{100})} \right)$$

which tells us that with probability $1 - \zeta$, for $n > \xi$

$$\sin^2(w_n, v_1) \leq \frac{C \log(1/\zeta)}{\zeta^2} \left( d\left(\frac{\xi}{n}\right)^{2h} + \frac{h^2 \tilde{V}}{(2h - 1)(\lambda_1 - \lambda_2)^2 n} \right)$$

for some positive constant $C$. If we plug in $\alpha = \frac{C' \log(n/\delta)}{\varepsilon \sqrt{n}}$ (as defined in Algorithm 3), set $\zeta = O(1)$, $K = O(1)$, select $h = c \log(n)$ and assume

$$n \geq C\left( \frac{M\lambda_1 \log(n)}{\lambda_1 - \lambda_2} + \frac{V\lambda_1^2 (\log(n))^2}{(\lambda_1 - \lambda_2)^2} \frac{(K\gamma \log^2(nd/\zeta) + 1)\lambda_1 \log(n/\delta) \log(n)d}{(\lambda_1 - \lambda_2)\varepsilon} + \frac{\lambda_1^2 \log^2(n)}{(\lambda_1 - \lambda_2)^2} \right)$$

we are guaranteed $n \geq \xi$ and $d(\xi/n)^{2\alpha} \leq 1/n^2$, so we will obtain the wished bound. $\qquad\square$

*Remark.* An analogue to Lemma 23 holds as well for k-DP-Ojas, by simply setting

$$B_n = \tilde{O}\left( \sqrt{\frac{V}{n}} + \frac{(\gamma + 1)d \log(1/\delta)}{\varepsilon n} \right).$$

### E.2 Sample Size requirements

The sample size condition in Theorem 1:

$$n \geq C \max \begin{cases} e^{\kappa'^2} + \dfrac{d\,\kappa'\,\gamma\,\sqrt{\ln(1/\delta)}}{\varepsilon} + \kappa' M + \kappa'^2 V + \dfrac{\sqrt{d}\,(\ln(1/\delta))^{3/2}}{\varepsilon}, \\ \lambda_1^2\,\kappa'^2\,k^3\,V, \\ \dfrac{\kappa'^2\,\gamma\,k^2\,d\,\sqrt{\ln(1/\delta)}}{\varepsilon} \end{cases},$$

includes an exponential dependence on the spectral gap: $n \geq \exp(\kappa')$. While this is relatively harmless as there is no such exponential dependence in the utility guarantee of the Theorem, we are able to get rid of this exponential dependency in exchange for an additional term in the utility guarantee. When looking at the utility proof of MODIFIEDDP-PCA (Theorem 9) we see this term arises as we choose $T$ and $n$ so that $(\xi/T) < 1$, as this is one of the requirements of Theorem 7. The specific inequality that arose from bounding $(\xi/T)$ and that lead to this exponential dependency is

$$\frac{\lambda_1^2 (\log n)^2}{(\tilde{\lambda}_1 - \tilde{\lambda}_2)^2 T} \leq 1/3 \tag{28}$$

(see Equation (25)). As we selected $T = c'(\log n)^3$, we required $\log(n) \geq \lambda_1/(\lambda_1 - \lambda_2)$. By selecting a slightly larger $T = c\kappa \log^3 n$, we would get rid of this exponential dependence, however at the cost of getting an extra term of $\tilde{O}(\kappa^r \gamma^2 d^2 \log(1/\delta)/(\varepsilon n)^2)$ in the utility guarantee.

## F   Proof of Lower Bound

**Corollary 3** (Lower bound, Spiked Covariance). *Let the $d \times n$ data matrix $X$ have i.i.d. columns samples from a distribution $P = \mathcal{N}(0, U^\top \Lambda U^\top + \sigma^2 \mathbf{I}_d) \in \mathcal{P}(\lambda, \sigma^2)$ where $\mathcal{P}(\lambda, \sigma^2) = \{\mathcal{N}(0, \Sigma), \Sigma = U\Lambda U^\top + \sigma^2 \mathbf{I}_d, c\lambda \leq \lambda_k \leq \cdots \leq \lambda_1 \leq C\lambda\}$. Suppose $\lambda \leq c_0' \exp\{e\varepsilon - c_0(\varepsilon\sqrt{ndk} + dk)\}$ for some small constants $c_0, c_0' > 0$. Then, there exists an absolute constant $c_1 > 0$ such that*

$$\inf_{\tilde{U} \in \mathcal{U}_{\varepsilon, \delta}} \sup_{P \in \mathcal{P}(\lambda, \sigma^2)} \mathbb{E}[\zeta] \geq c_1 \left( \left( \frac{\sigma\sqrt{\lambda_1 + \sigma^2}}{\sum_{i=1}^k (\lambda_i + \sigma^2)} \right) \left( \sqrt{\frac{dk}{n}} + \frac{dk}{n\varepsilon} \right) \bigwedge 1 \right).$$

*Proof.* Combining Lemma 26 with Theorem 13 we obtain the lower bound in Corollary 3. □

**Lemma 26** (Reduction to Frobenius norm). *Let $\Sigma$ be a PSD $d \times d$ matrix with top-$k$ eigenvectors $V_k \in \mathbb{R}^{d \times k}$ and eigenvalues $\lambda_1 \geq \cdots \geq \lambda_d$. Any $U \in \mathbb{R}^{d \times k}$ that satisfies $\|UU^\top - V_k V_k\|_F^2 \geq \gamma$, must incur*

$$\zeta^2 \geq \frac{\gamma \Delta_k}{2 \sum_{i=1}^k \lambda_i}$$

*where $\Delta_k := \lambda_k - \lambda_{k+1}$.*

*Proof.* As

$$\langle UU^\top, X \rangle = \frac{\langle UU^\top, X \rangle}{\langle V_k V_k^\top, X \rangle} \langle V_k V_k^\top, X \rangle$$

$$= \frac{\mathrm{Tr}(UU^\top X)}{\mathrm{Tr}(V_k V_k^\top X)} \langle V_k V_k^\top, X \rangle$$

this implies that

$$\frac{\mathrm{Tr}(UU^\top X)}{\mathrm{Tr}(V_k V_k^\top X)} \geq 1 - \zeta^2. \tag{29}$$

So any upper bound on $\frac{\mathrm{Tr}(UU^\top X)}{\mathrm{Tr}(V_k V_k^\top X)}$ will give us a lower bound on $\zeta^2$. By Lemma 27 we know

$$\frac{\|UU^\top - VV\|_F^2 \Delta_k}{2} \leq \mathrm{Tr}(VV^\top X) - \mathrm{Tr}(UU^\top X)$$

which gives us that

$$\frac{\mathrm{Tr}(UU^\top X)}{\mathrm{Tr}(V_k V_k^\top X)} \leq 1 - \frac{\|UU^\top - V_k V_k\|_F^2 \Delta_k}{2\mathrm{Tr}(V_k V_k^\top X)}.$$

By equation 29 this gives us

$$\frac{\|UU^\top - V_k V_k\|_F^2 \Delta_k}{2 \sum_{i=1}^k \lambda_i} \leq \zeta^2.$$

□

**Lemma 27.** *For an orthonormal matrix $U \in \mathbb{R}^{d \times k}$ and a psd matrix $X \in \mathbb{R}^{d \times d}$ with eigengap $\Delta_k = \lambda_k - \lambda_{k+1}$ and top $k$ eigenvectors $V \in \mathbb{R}^{d \times k}$, we have*

$$\frac{\|UU^\top - VV\|_F^2 \Delta_k}{2} \leq Tr(VV^\top X) - Tr(UU^\top X)$$

*Proof.* We will proof this by proving the following two (in)equalities:

$$\Delta_k \|\sin\Theta(U,V)\|_F^2 \leq \mathrm{Tr}(VV^\top X) - \mathrm{Tr}(UU^\top X) \tag{30}$$

$$\|UU^\top - VV^\top\|_F = \sqrt{2}\|\sin\Theta(U,V)\|_F \tag{31}$$

Equation (30): We first note that

$$\mathrm{Tr}(VV^\top X) - \mathrm{Tr}(UU^\top X) = \mathrm{Tr}((VV^\top - UU^\top)(X - \lambda_{k+1})\mathbf{I}_d)$$

as

$$\mathrm{Tr}((VV^\top - UU^\top)\lambda_{k+1}) = \lambda_{k+1}\left(\mathrm{Tr}(VV^\top) - \mathrm{Tr}(UU^\top)\right) = 0$$

where the last equality follows as $\mathrm{Tr}(UU^\top) = k = \mathrm{Tr}(VV^\top)$. Now

$$\mathrm{Tr}((VV^\top - UU^\top)(X - \lambda_{k+1})\mathbf{I}_d) = \mathrm{Tr}((VV^\top + (\mathbf{I}_d - VV^\top))(VV^\top - UU^\top)(X - \lambda_{k+1})\mathbf{I}_d)$$

$$= \mathrm{Tr}(VV^\top(VV^\top - UU^\top)(X - \lambda_{k+1})) + \mathrm{Tr}((\mathbf{I} - VV^\top)(VV^\top - UU^\top)(X - \lambda_{k+1}))$$

$$\geq \mathrm{Tr}(VV^\top(VV^\top - UU^\top)(X - \lambda_{k+1}\mathbf{I}_d))$$

$$\geq \Delta_k \mathrm{Tr}((V_k V_k^\top - UU^\top)_+)$$

where $(A)_+$ is obtained by replacing each eigenvalue of the matrix $A$ with $\max\{\mu_i, 0\}$. Now we note that

$$\text{Tr}((V_k V_k^\top - UU^\top)_+) \geq \|\sin\Theta(U,V)\|_F^2$$

Hence, since the $\sin\theta_i$ are nonnegative (as the principal angles $\theta_i$ lie in $[0, \pi/2]$) we have $\text{Tr}((V_k V_k^\top - UU^\top)_+) = \sum_{i=1}^k \sin\theta_i$. Further, by definition we have

$$\|\sin\Theta(U,V)\|_F^2 = \sum_{i=1}^k \sin^2\theta_i.$$

So by noticing that for any angle $\theta \in [0, \pi/2]$, $\sin\theta \geq \sin^2\theta$ we have proved the first inequality.

Equation (31): $\|UU^\top - VV^\top\|_F^2 = \text{Tr}((UU^\top - VV^\top)^2)$. By expanding $(UU^\top - VV^\top)^2$ we see

$$(UU^\top - VV^\top)^2 = UU^\top - UU^\top VV^\top - VV^\top UU^\top + VV^\top$$

which gives us

$$\begin{aligned}\text{Tr}((UU^\top - VV^\top)^2) &= 2k - 2\text{Tr}(UU^\top VV^\top) \\ &= 2k - \text{Tr}(V^\top UU^\top V) \\ &= 2k - \|U^\top V\|_F^2\end{aligned}$$

Lastly, utilizing

$$\|U^\top V\|_F^2 = 2\sum_{i=1}^k \cos^2\theta_i = 2\sum_{i=1}^k(1 - \sin^2\theta_i)$$

the proof follows. $\qquad\square$

## F.1 Existing Lower Bounds

**Theorem 11** (Lower bound, Gaussian distribution, Theorem 5.3 in Liu et al. [2022a]). *Let $\mathcal{M}_\varepsilon$ be a class of $(\varepsilon, 0)$-DP estimators that map $n$ i.i.d. samples to an estimate $\hat{v} \in \mathbb{R}^d$. A set of Gaussian distributions with $(\lambda_1, \lambda_2)$ as the first and second eigenvalues of the covariance matrix is denoted by $\mathcal{P}_{(\lambda_1, \lambda_2)}$. There exists a universal constant $C > 0$ such that*

$$\inf_{\hat{v} \in \mathcal{M}_\varepsilon} \sup_{P \in \mathcal{P}_{(\lambda_1,\lambda_2)}} \mathbb{E}_{S\sim P^n}[\sin(\hat{v}(S), v_1)] \geq C\min\left(\kappa\left(\sqrt{\frac{d}{n}} + \frac{d}{\varepsilon n}\right)\sqrt{\frac{\lambda_2}{\lambda_1}}, 1\right)$$

**Theorem 12** (Lower bound without Assumption 4, Theorem 5.4 in Liu et al. [2022a]). *Let $\mathcal{M}_\varepsilon$ be a class of $(\varepsilon, \delta)$-DP estimators that map $n$ i.i.d. samples to an estimate $\hat{v} \in \mathbb{R}^d$. A set of distributions satisfying 1.-3. of Assumption A with $M = \tilde{O}(d + \sqrt{n\varepsilon/d})$, $V = O(d)$ and $\gamma = O(1)$ is denoted by $\tilde{\mathcal{P}}$. For $d \geq 2$, there exists a universal constant $C > 0$ such that*

$$\inf_{\hat{v} in \mathcal{M}_\varepsilon} \sup_{P \in \tilde{\mathcal{P}}} \mathbb{E}_{S\sim P^n}[\sin(\hat{v}(S), v_1)] \geq C\kappa\min\left(\sqrt{\frac{d \wedge \log((1-e^{-\varepsilon})/\delta)}{\varepsilon n}}, 1\right)$$

**Theorem 13** (Theorem 4.2 in Cai et al. [2024]). *Let the $d \times n$ data matrix $X$ have i.i.d. columns samples from a distribution $P = \mathcal{N}(0, U^\top \Lambda U^\top + \sigma^2 \mathbf{I}_d) \in \mathcal{P}(\lambda, \sigma^2)$. Suppose $\lambda \leq c_0' \exp\{e\varepsilon - c_0(\varepsilon\sqrt{ndk} + dk)\}$ for some small constants $c_0, c_0' > 0$. Then, there exists an absolute constant $c_1 > 0$ such that*

$$\inf_{\tilde{U} \in \mathcal{U}_{\varepsilon,\delta}} \sup_{P \in \mathcal{P}(\lambda, \sigma^2)} \frac{\mathbb{E}\|\tilde{U}\tilde{U}^\top - UU^\top\|_F}{\sqrt{k}} \geq c_1\left(\left(\frac{\sigma\sqrt{\lambda + \sigma^2}}{\lambda}\right)\left(\sqrt{\frac{d}{n}} + \frac{d\sqrt{k}}{n\varepsilon}\right)\bigwedge 1\right)$$

*where the infimum is taken over all the possible $(\varepsilon, \delta)$-DP algorithms, denoted by $\mathcal{U}_{\varepsilon,\delta}$ and the expectation is taken with respect to both $\tilde{U}$ and $P$ and*

$$\mathcal{P}(\lambda, \sigma^2) := \{\mathcal{N}(0, \Sigma) : \Sigma = U\Lambda U^\top + \sigma^2 \mathbf{I}_d, U \in \mathbb{O}_{d,k}, \Lambda = diag(\lambda_1, \ldots, \lambda_k), c_0\lambda \leq \lambda_k \leq \lambda_1 \leq C_0\lambda\}$$

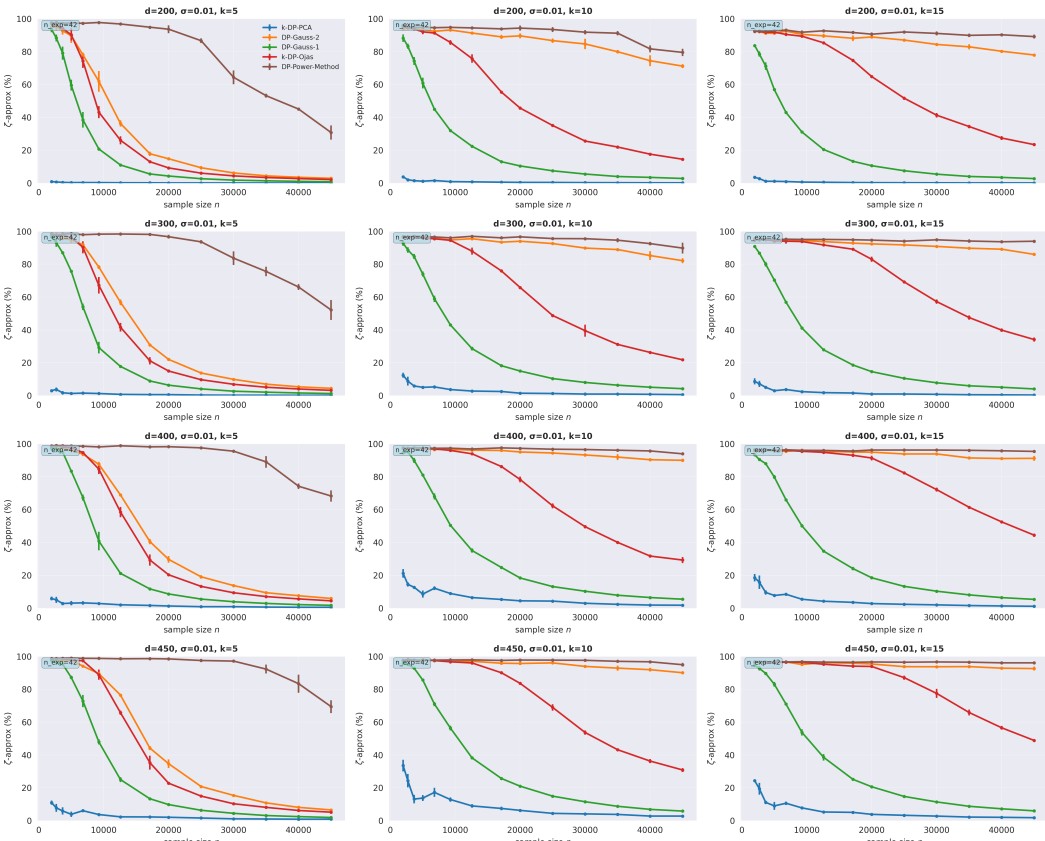

Figure 3: Comparison of k-DP-PCA and k-DP-Ojas for varying $k$ and $d$ (also including DP-Gauss-1 (input perturbation), DP-Gauss-2 (object perturbation), and DP-Power-Method) on the spiked covariance model. We plot the mean over 50 trials, with the bars representing the standard deviation.

## G   Experiments

In Section 5 we compare the performance of k-DP-PCA and k-DP-Ojas to two modified versions of the DP-Gauss algorithms of Dwork et al. [2014b], we refer to as `DP-Gauss-1` and `DP-Gauss-2` respectively, and a modified version of the noisy power method [Hardt and Price, 2014].

Given a stream of matrices $\{A_i\}$ and a clipping threshold $\beta$ (that is chosen based on the distribution of the input data), `DP-Gauss-1` first clips each matrix to have trace at most $\beta^2$: $\tilde{A}_i = A_i \cdot \min\{1, \beta^2/\mathrm{Tr}(A_i)\}$. In a second step it computes the sum of the $\tilde{A}_i$: $X = \sum_i \tilde{A}_i$ and then performs the gaussian mechanism: $X' = X + E$, where $E$ is a symmetric matrix with their upper triangle values (including its diagonal) i.i.d. sampled from $\mathcal{N}(0, \Delta_1^2 \mathbf{I}_d)$ and $\Delta_1 = \beta^2 \sqrt{2\log(1.25/\delta)}/\varepsilon$. Lastly, it performs an eigenvalue decomposition on $X'$, and releases the top $k$ eigenvectors.

`DP-Gauss-2` just like `DP-Gauss-1` clips the matrices and sums them up to obtain $X$. Next it extracts $V_k$ the top $k$ eigenvectors of $X$ via an eigenvalue decomposition and privatizes its eigengap: $g_k = \lambda_k - \lambda_{k+1} + z$, where $z \sim \mathrm{Lap}(2/\varepsilon)$. It then applies the Gaussian mechanism to $V_k$: $V_k' = V_k + E$, where $E$ is a symmetric matrix with their upper triangle values (including its diagonal) i.i.d. sampled from $\mathcal{N}(0, \Delta_2^2 \mathbf{I}_d)$ and

$$\Delta_2 = \frac{\beta^2(1 + \sqrt{2\log(1/\delta)}/\varepsilon)}{|g_k - 2(1 + \log(1/\delta)/\varepsilon)|}.$$

Finally, an additional eigenvalue decomposition is performed on $V_k'$, as the introduction of noise may result in a matrix whose columns are no longer orthogonal. The top $k$ eigenvectors obtained from

this decomposition are then released. It is important to note that if $g_k$ is not positive, the procedure is no longer differentially private, despite adherence to the algorithm described in the original paper (see Algorithm 2 in [Dwork et al., 2014b]). A fully compliant implementation—also discussed in the paper—would employ the PTR mechanism, albeit at the expense of increased privacy loss. For simplicity and to allow greater flexibility, we instead opt to resample fresh noise whenever $g_k \leq 0$.

`DP-Power-Method` clips the matrices with respect to the square root of the trace and the trace of the square root of the diagonal. For $A = aa^\top$ with $a \in \mathbb{R}^d$ the first corresponds to clipping with respect to $\|a\|_2 \leq \beta$, whereas the second to clipping with respect to $\|a\|_1 \leq \alpha$, where the clipping threshold $\beta$ is the same as we choose for the DP-Gauss algorithms. In a second step it then also computes the sum of the clipped matrices and then performs the noisy power method (find algorithm in [Nicolas et al., 2024]) where the gaussian noise that is being added at every iteration of the power method is scaled with an additional $\beta \cdot \alpha$ factor.

### G.1   Synthetic Data

We sample data from the spiked covariance model, meaning each matrix $A_i \in \mathbb{R}^{d \times d}$ consists of a deterministic rank-$k$ component, plus random noise that ensures $A_i$ is full-rank. For the case $k = 1$, we generate samples via $x_i = s_i + n_i$, where $s_i \sim \text{Unif}(\{\lambda_1 v, -\lambda_1 v\})$, with $v \in \mathbb{R}^d$ a unit vector and $\lambda_1 \in \mathbb{R}$ a scalar. The noise term is sampled as $n_i \sim \mathcal{N}(0, \sigma^2 \mathbf{I}_d)$. We then define $A_i = x_i x_i^\top$. Here, $\lambda_1$ and $\sigma$ are inputs to the sampling function, while $v$ is obtained by sampling a standard Gaussian vector of dimension $d$ and normalizing it to unit length. For $k > 1$, we proceed differently: we first sample a random matrix $V \in \mathbb{R}^{d \times k}$ with i.i.d. standard normal entries, then apply the Gram–Schmidt process to obtain $V_k \in \mathbb{R}^{d \times k}$, a matrix with $k$ orthonormal columns. We construct $A_i = V_k \Lambda V_k^\top + z_i z_i^\top$, where $z_i \sim \mathcal{N}(0, \sigma^2 \mathbf{I}_d)$, and $\Lambda \in \mathbb{R}^{k \times k}$ is a diagonal matrix whose entries are user-specified eigenvalues. We note that this construction for $k > 1$ is not a direct extension of the $k = 1$ case. In particular, independently sampling $k$ vectors as in the $k = 1$ case and summing their outer products would result in a mixture of Gaussians rather than a single spiked covariance structure. To avoid this and retain a well-defined rank-$k$ component, we instead fix the subspace and apply deterministic structure through $V_k \Lambda V_k^\top$.

We set $\beta = C\sqrt{\lambda_1} + \sigma\sqrt{d \log(n/\zeta)}$ for `DP-Gauss-1` and `DP-Gauss-2`, where $n$ is the number of samples, $1 - \zeta$ is the probability of not clipping. We set $\zeta = 0.01$ uniformly across all methods, including our algorithms (MODIFIEDDP-PCA and k-DP-Ojas) as well as both Gauss baselines. For both k-DP-PCA and k-DP-Ojas, the parameters $K$ and $a$ (as defined in Assumption A) must be provided as inputs. In the case of data generated as described above, we have $a = 1$ and $K = O(1)$, and thus we set $a = 1$ and $K = 1$ for our experiments. Additionally, k-DP-PCA requires specifying a batch size $B$, which is used in the PRIVMEAN algorithm. While the theoretical analysis suggests that the optimal choice is $B = n/\log^3(n)$, where $n$ is the sample size, we found empirically that setting $B = \sqrt{n}$ yielded improved performance in practice. Lastly, we need to set a learning rate for k-DP-PCA and k-DP-Ojas. For k-DP-PCA we set the learning rates to be

$$\eta_t^i = 1/(20\sigma\lambda_i + (\lambda_i - \lambda_{i+1}) \cdot t/\log(n))$$

where $t$ refers to the $t$th update step inside of MODIFIEDDP-PCA ($t \in [T]$ where $T = \lfloor n/B \rfloor$) and $i$ to the $i$th iteration of k-DP-PCA. For k-DP-Ojas we empirically found that simply choosing a decreasing learning rate (independent of eigenvalues) resulted in good performance, so we set the learning rate to be

$$\eta_j = 1/(1 + j)$$

for $j \in [n]$ for all $k$ iterations of k-DP-Ojas.

### G.2   Gaussian Data

For more general data distributions—that is, those not exhibiting a clean signal-plus-noise decomposition—Corollary 1 indicates that k-DP-PCA can still outperform existing state-of-the-art methods, primarily due to its favorable scaling with the ambient dimension $d$. However, our second algorithm, k-DP-Ojas, offers comparable utility guarantees in such settings (see Corollary 4).

While k-DP-PCA has strong theoretical properties, it requires careful tuning of the learning rate, which can be challenging in practice. Specifically, it depends on a step size parameter that must be adapted to the signal-to-noise ratio and spectrum of the data. In regimes where the noise level is

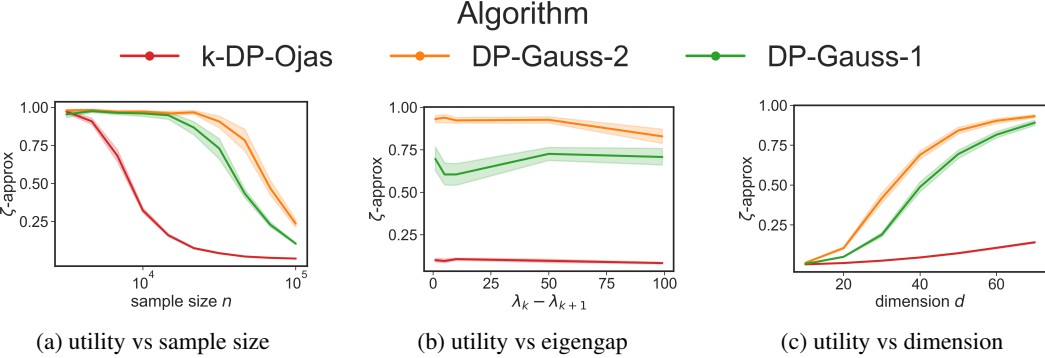

Figure 4: Comparison of k-DP-PCA vs DP-Gauss-1 (input perturbation) and DP-Gauss-2 (output perturbation) on gaussian data. We plot the mean over 50 trials, with shaded regions representing 95% confidence intervals. We set $k = 2$, $d = 200$, $\lambda_1 = 10$, $\varepsilon = 1$, and $\delta = 0.01$.

moderate or high, the theoretical gains of k-DP-PCA do not clearly outweigh the practical overhead of hyperparameter tuning and range estimation.

In contrast, k-DP-Ojas is simpler to deploy: it requires no hyperparameter tuning and exhibits robust performance across a range of learning rates. As shown in Figure 4, k-DP-Ojas consistently outperforms other state-of-the-art methods on data of the form $A_i = x_i x_i^\top$ with $x_i \sim \mathcal{N}(0, \Sigma)$. For these reasons, we recommend k-DP-Ojas as the preferred method in practical settings involving general data distributions.

### G.3 Further comments

Lastly, we comment on a potential modification to our algorithm. The subroutine PRIVRANGE is used to privately estimate a suitable truncation threshold around the mean for PRIVMEAN. In certain scenarios, however, it may be preferable to fix this threshold in advance or determine it through an alternative (non-private) mechanism. Doing so would eliminate the need to estimate the threshold from the data under differential privacy, thereby avoiding the substantial sample complexity that this estimation typically requires.

This consideration directly explains the lower bound on sample size in k-DP-PCA: a sufficient number of samples is necessary to ensure that the truncation threshold can be estimated both meaningfully and in a privacy-preserving manner. Interestingly, this also sheds light on why the algorithm may perform better in practice than its theoretical utility bounds suggest. In particular, even when using fewer samples than required for formal utility guarantees—i.e., below the threshold for reliable private estimation of the truncation point—k-DP-PCA can still exhibit strong empirical performance. In such cases, the algorithm retains its privacy guarantees, but the formal utility guarantees no longer apply.

More broadly, while our algorithm is provably asymptotically optimal, the choice of range finder or mean estimation method can significantly impact empirical performance depending on the data distribution. One of the key advantages of our iterative framework is its modularity. As demonstrated by k-DP-Ojas in Section 4, the algorithm can be viewed as a plug-and-play template: the private mean estimation subroutine can be replaced with alternative methods tailored to specific data characteristics. Crucially, Theorem 2 ensures that any such substitution carries over a corresponding utility guarantee, enabling both flexibility and theoretical rigor.

## H  Algorithms used in Modified DP-PCA

Below we describe the two subroutines that estimate the range and mean of the gradients in MODIFIEDDP-PCA.

---

**Algorithm 6** Top-Eigenvalue-Estimation, Algorithm 4 in [Liu et al., 2022a]

---

**Input:** $S = \{g_i\}_{=1}^{B}$ , privacy parameters $(\varepsilon, \delta)$, failure probability $\tau \in (0, 1)$

1: $\tilde{g}_i \leftarrow g_{2i} - g_{2i-1}$ for $i \in 1, 2, \ldots, \lfloor B/2 \rfloor$
2: $\tilde{S} = \{\tilde{g}_i\}_{=1}^{\lfloor B/2 \rfloor}$
3: Partition $\tilde{S}$ into $k = C_1 \log(1/(\delta\tau)/\varepsilon$ subsets and denote each dataset as $G_j \in \mathbb{R}^{d \times b}$ (where $b = \lfloor B/2k \rfloor$ is the size of the dataset)
4: $\lambda_1^{(j)} \leftarrow$ top eigenvalue of $(1/b)G_j G_j^\top$ for all $j \in [k]$
5: partition $[0, \infty)$ into $\Omega \leftarrow \{\ldots, [2^{-2/4}, 2^{-1/4}), [1, 2^{1/4}), \ldots\}$
6: run $(\varepsilon, \delta)$-DP histogram learner on $\{\lambda_1^{(j)}\}_{j=1}^{k}$ over $\Omega$
7: **if** all bins are empty **then**
8:     **return** $\perp$
9: **else**
10:     for $[l, r]$ the bin with the maximum number of points in the DP histogram
11:     **return** $\hat{\Lambda} = l$
12: **end if**

---

---

**Algorithm 7** Private-Mean-Estimation, Algorithm 5 in [Liu et al., 2022a]

---

**Input:** $S = \{g_i\}_{=1}^{B}$ , privacy parameters $(\varepsilon, \delta)$, target error $\alpha$, failure probability $\tau \in (0, 1)$, approximate top eigenvalue $\hat{\Lambda}$

1: let $\upsilon = 2^{1/4} K \sqrt{\hat{\Lambda}} \log^2(25)$
2: **for** $j = 1, 2, \ldots, d$ **do**
3:     Run $(\frac{\varepsilon}{4\sqrt{2d\log(4/\delta)}}$ , $\frac{\delta}{4d})$-DP histogram learner of Lemma on $\{g_{ij}\}_{i \in [B]}$ over $\Omega = \{\ldots, (-2\upsilon, -\upsilon], (-\upsilon, 0], (0, \upsilon], (\upsilon, 2\upsilon], \ldots\}$
4:     Let $[l, h]$ be the bucket that contains maximum number of points in the private histogram
5:     $\bar{g}_j \leftarrow l$
6:     Truncate the $j$-th coordinate of gradient $\{g_i\}_{i \in [B]}$ by $[\bar{g}_j - 3K\sqrt{\hat{\Lambda}} \log^a(BD/\tau), \bar{g}_j + 3K\sqrt{\hat{\Lambda}} \log^a(BD/\tau)]$.
7:     Let $\tilde{g}_i$ be the truncated version of $g_i$
8: **end for**
9: Compute empirical mean of truncated gradients $\tilde{\mu} = (1/B) \sum_{i=1}^{B} \tilde{g}_i$ and add Gaussian noise:

$$\hat{\mu} = \tilde{\mu} + \mathcal{N}\left(0, \left(\frac{12K\sqrt{\hat{\Lambda}} \log^a(BD/\tau)\sqrt{2d\log(2.5/\delta)}}{\varepsilon B}\right)^2 \mathbf{I}_d\right)$$

10: **return** $\hat{\mu}$

---

