# OpenReview forum: "An Iterative Algorithm for Differentially Private $k$-PCA with Adaptive Noise"
_NeurIPS.cc/2025/Conference — NeurIPS 2025 poster_

### Official Review · Reviewer_cGF1 · 2025-06-13

**Clarity:** 4
**Significance:** 3
**Originality:** 3
**Rating:** 4
**Confidence:** 3

**Summary:**

The paper consider differentialy private $k$-PCA, where the goal is to privately estimate the $k$ leading eigenvectors of the covariance matrix of data. The method and analysis is tailored to the stochastic setting, where the data vectors are drawn from a given distribution. The paper makes improvements over the prior works in essentially two ways: first, it improves the baseline method and analysis by Liu et al. (2022) which is for eastimating the leading eigenvectors of the covariance matrix, and second, the paper applies the deflation techniques presented by Jalumbati et al. (2024) to apply the single vector estimator $k$ times and carrying out deflation. Numerical comparisons to the baseline method by Liu et al. (2022) and also to the Analyze Gauss (Dwork et al., 2014) which works in the data-deterministic setting and where the whole covariance matrix is pertubed verify the effectiveness of the proposed method.

**Questions:**

- Can you comments on how the deflation method would compare to block iteration methods such as the noisy power method (Hardt and Price, 2014, Balcan et al., 2016) ? Both theoretically and numerically? As that is fundamentally a different approach (deflation vs. block iteration), I think it would be important to mention that approach.

- Why did you not consider other baselines than Analyze Gauss and the method by Liu et al. ?

**Ethical Concerns:**

["NO or VERY MINOR ethics concerns only"]

**Final Justification:**

This is a well-written and solid paper making clear contributions to DP PCA in the stochastic setting, including improvements over Liu et al. (2022) via application of deflation techniques to estimate multiple leading eigenvectors. The rebuttal addressed concerns about baselines by providing additional experiments with larger $k$ and by including comparisons to noisy power methods, supporting the claimed advantages of the proposed method. While the experimental section remains limited in scope (authors promise more experiments for the final version), the theoretical contributions seem sound.

**Limitations:**

yes

**Quality:**

3

**Strengths And Weaknesses:**

Strengths:

- Improvement of the single vector method by Liu et al. (2022)

- Novel analysis based on a novel analysis of the non-private method by Jain et al. (2016)

- Clear novelty: method to estimate $k$ leading vectors, enabled by the deflation (Jalumbati, 2024)

- The paper is very well written, it is nice that there are examples of data distributions that satisfy the assumptions of the analysis vs. exaples that do not satisfy.


Weaknesses:

- The experimental section is limited, as there are two examples with synthetic data and in both cases $k=2$. Comparisons with some real-world data with larger $k$ would strengthen the experimental section.

- The selection of baseline methods is limited. The method by [Cai et al. (2024)](https://arxiv.org/pdf/2401.03820) is not considered in the baselines. But most importantly, the noisy power iteration is not considered at all. There are recent works, e.g.

[Nicolas, J., Sabater, C., Maouche, M., Mokhtar, S. B., & Coates, M. (2024). Differentially private and decentralized randomized power method.](https://arxiv.org/pdf/2411.01931)

that are block iterations and consider computing simultaneously the leading $k$-eigensubspace for the covariance matrix. There is also the analysis of

[Balcan, M. F., Du, S. S., Wang, Y., & Yu, A. W. An improved gap-dependency analysis of the noisy power method. In Conference on Learning Theory 2016](http://proceedings.mlr.press/v49/balcan16a.pdf)

which gives an error analysis for the noisy power method, that could be directly compared to the results presented here.

The existing methods and analyses for the noisy power iteration, to the best of my knowledge, are for the data-deterministic setting. However, you could compare to that method using similar modifications that you make for the Analyze Gauss.

---

> ### Author Rebuttal · Authors · 2025-07-31
>
> We thank the reviewer for their comments. We will address them in order:
>
> > The experimental section is limited:
>
> The primary focus of the paper was on theoretical results for the problem of stochastic k-DP-PCA for $k >1$. However, we do appreciate the reviewers' concerns and during the rebuttal, we repeated our spiked data experiments on larger $d$ and $k$ as well as on different baselines and report the results below. We observe the same trends as highlighted in the paper. We show one of the experiments in the ASCII plot below and report the rest in tables.
>
> ```
>                             Mean Distance vs. Sample Size (k=10, d=450)
>     ┌──────────────────────────────────────────────────────────────────────────┐
> 0.98┤ •• DP-Gauss-1       ♰                    ♰                               │
>     │ ♰♰ DP-Gauss-2                                                      ♰     │
> 0.82┤ pp DP-Power-Method                                                       │
>     │ cc k-DP-Ojas                                                             │
>     │ ss k-DP-PCA                           c                                  │
> 0.66┤                                                                          │
>     │                                                                          │
> 0.50┤                                                                          │
>     │                    p                                                     │
> 0.34┤                   •                                             c        │
>     │                                                                          │
>     │                                                                          │
> 0.18┤    s                                   •p                                │
>     │                 s                                                •p      │
> 0.03┤                                      s                         s         │
>     └┬─────────────────┬──────────────────┬─────────────────┬─────────────────┬┘
>   1995.0            12746.2            23497.5           34248.8        45000.0
> distance                            sample size n
>
> ```
>
> |    Algorithm    |  $n$  |  $d$  |  $k$  |  distance  |
> |-----------------|-------|-------|-------|------------|
> |   DP-Gauss-1    | 20000 |  200  |   5   |   0.043    |
> |   DP-Gauss-2    | 20000 |  200  |   5   |   0.138    |
> | DP-Power-Method | 20000 |  200  |   5   |   0.068    |
> |    k-DP-Ojas    | 20000 |  200  |   5   |    0.1     |
> |   **k-DP-PCA**      | 20000 |  200  |   5   |   0.002    |
> |-----------------|-------|-------|-------|------------|
> |   DP-Gauss-1    | 20000 |  200  |  15   |   0.104    |
> |   DP-Gauss-2    | 20000 |  200  |  15   |   0.885    |
> | DP-Power-Method | 20000 |  200  |  15   |   0.246    |
> |    k-DP-Ojas    | 20000 |  200  |  15   |   0.666    |
> |    **k-DP-PCA**      | 20000 |  200  |  15   |   0.003    |
> |-----------------|-------|-------|-------|------------|
> |   DP-Gauss-1    | 40000 |  450  |  15   |   0.072    |
> |   DP-Gauss-2    | 40000 |  450  |  15   |   0.933    |
> | DP-Power-Method | 40000 |  450  |  15   |   0.099    |
> |    k-DP-Ojas    | 40000 |  450  |  15   |    0.57    |
> |    **k-DP-PCA**      | 40000 |  450  |  15   |    0.02    |
>
> > The selection of baseline methods is limited (why did you not consider Cai et al, noisy power method and other baselines than Analyze Gauss)
>
> * Cai et al. :  We note that the DP guarantees of Cai et al  are conditional on the data coming from a spiked covariance distribution. So, **if the data comes from a different distribution, Cai. et. al.'s algorithm is non-private**. Our algorithm is private for any dataset. Nevertheless, for spiked covariance data our algorithm performs similarly to Cai et al. (see table above)
>
> * Noisy power method: The original noisy power method paper (Hardt et al.) only gives DP guarantees in terms of a change in a single entry by 1 or a spectral change of norm 1. Whereas our algorithm gives DP guarantees for an arbitrarily large change of an entire row, which is a strictly stronger DP guarantee. When we converted their adjacency notion to ours via clipping and increasing the noise with a multiplicative factor (similarly to what we did for Analyze Gauss) their utility guarantee ended up being worse than Analyze Gauss, which is why we do not add it to the baseline.
>
> * Improved noisy power method: We do thank the reviewer for bringing the paper by Nicolas et al to our attention. We were not aware of this paper and plan to mention the Nicoals et al. as well as Balcan et al. in the final version of our paper. They give a stronger privacy guarantee than the original noisy power method paper, by defining the adjacency as $A' = A + C$, for $\sqrt{\sum_{i=1}^n \| C_{i,:} \|_1^2} \leq 1$. If we convert this to our setting (similarly to what we did for Analyze Gauss as suggested by the reviewer), by defining $A' = A + a a^{\top}$, then $C = a a^{\top}$ and the ith row of $C$ is equal to $| a_i | || a ||_1$ which results in the requirement that $|| a ||_2 || a ||_1 \leq 1$. We know $|| a ||_1 > || a||_2$ so simply clipping the matrices to $|| a ||_2 \leq \beta$ and then scaling accordingly, we add even less noise than we need to. Nevertheless, this makes their algorithm comparable to AnalyzeGauss in terms of utility guarantees with respect to k and d. We conducted experiments with their algorithm and reported it above. As the results show, our method outperforms them.
>
> > Can you comment on how the deflation method would compare to block iteration methods such as the noisy power method
>
> The deflation method is not inherently a DP method so we cannot directly compare it to the noisy power method. But we can compare block iteration methods such as the noisy power method to k-DP-PCA and k-DP-Ojas
> 1. Theoretically:
>     * **different notion of neighboring (for DP)**: as mentioned above the noisy power method considers a different notion of neighboring. If we modify it to comply with our notion this would lead to an additional $\sqrt{d}$ factor in the utility guarantee, and therefore would make their utility guarantee scale by $\sqrt{d}$ worse than k-DP-PCA and k-DP-Ojas
>     * **The privacy noise of k-DP-PCA adapts to the randomness in the data**: one problem we wanted to solve with our algorithm is that many DP algorithms introduce excessive noise for DP even when the intrinsic randomness within the data samples $A_i$ is small. Modified DP-PCA adapts to the data and hence allows us to add very little noise in data regimes such as the spiked data model.
> 2. Experimentally: The error of the original noisy power methods algorithm scales considerably worse than Analyze Gauss with respect to growing dimension d. The newer analysis of Nicolas et al. and Balcan et al. show that we can add less noise, which leads to an error scaling similar to Analyze Gauss for the neighboring definition we are considering.

---

> ### Comment · Reviewer_cGF1 · 2025-08-02
>
> Thank you for the reply, this clarifies. I am still slightly suprised, that the noisy power iteration would not compete with Analyze Gauss. Notice that you don't need to have the same adjacency notion as Hardt et al., you can pick any adjacency notion like change of one row where each step of the noisy power method has the privacy guarantees of a Gaussian mechanism and you get the total privacy guarantees via composition analysis. The privacy analysis is independent of the results by Hardt et al. Anyhow, I trust these results are correct, and I believe the ordering must depend on the data as well.

---

### Official Review · Reviewer_Ekub · 2025-06-29

**Clarity:** 3
**Significance:** 3
**Originality:** 2
**Rating:** 5
**Confidence:** 3

**Summary:**

The paper tackles differentially-private stochastic k-PCA. It combines a deflation (Algorithm 1 k-DP-PCA) with a new 1-PCA oracle (MODIFIEDDP-PCA). Under a sub-Gaussian data model, they authors prove $(\epsilon,\delta)$-DP and show that with only $n = \tilde{O}(d)$ samples their estimator recovers a $k$-dimensional subspace whose retained variance has near-optimal dependence on $k, n$. An information-theoretic lower bound for the case of the spiked covariance model matches dependence on factors except $k$. The authors further supplement their theoretical results with experiments to show the linear-scaling of the error with data dimension $d$.

**Questions:**

Please see weaknesses section above.

**Ethical Concerns:**

["NO or VERY MINOR ethics concerns only"]

**Final Justification:**

I thank the authors for their detailed explanations. I think my questions have been answered. I trust the authors will incorporate the explanation of challenges in their proof setting in the final revision. I would be happy to increase my score. I think extending Liu et al's work to k-PCA using deflation is an interesting line of work and this paper presents a good contribution.

**Limitations:**

Yes

**Quality:**

3

**Strengths And Weaknesses:**

Strengths :

A clear rigorous proof of differential privacy and utility for k-PCA with a nearly tight lower bound and efficient implementation. Nice combination and generalization of the ideas of Jambulapati et al. (2024) and Liu et al (2022). I have been keeping up with the literature and I think the paper is overall well-written and a nice contribution to the privacy and PCA literature.

Weaknesses/Questions :

I don't see any obvious weaknesses apart from the ones the authors acknowledged in Section 6, so I will ask state some questions/suggestions instead.

1. I would appreciate it if, apart from the lower bound, the authors could bring out their contributions over Jambulapati et al. (2024) and Liu et al (2022) and mention novel parts of their analysis, even as a sketch, as compared to these existing analysis.

2. Line 198-199 : The authors mention that Jambulapati et al's ePCA oracle definition doesn't extend to the stochastic case. However, based on my understanding, Jambulapati et al. do provide bounds for robust k-PCA where $1-\epsilon$ fraction of the data is drawn iid from some distribution. Therefore, I am a little confused as to what the authors meant with this claim.

3. Regarding dependence on $k$ : Again referencing Jambulapati et al., based on my understanding, for the case of robust heavy-tailed data, they show that in fact a union bound over the $k$ deflation steps is not necessary and a stability result in the first stage implies stability for the subsequent stages due to the nature of the deflation. This helps them save a $k$ factor in the sample complexity.

I am wondering if such a trick would work in the DP case as well?

4. Notion of $k$-PCA : The authors consider energy k-PCA. Another notion of PCA convergence is the correlation k-PCA which measures mis-alignment between estimated and true eigenspaces. I am just wondering if the authors believe that their results provide near-optimal guarantees for such a case as well.

---

> ### Author Rebuttal · Authors · 2025-07-31
>
> We thank the reviewer for their comments. We will address them in order:
>
> > I would appreciate it if, apart from the lower bound, the authors could bring out their contributions over Jambulapati et al. (2024) and Liu et al (2022) and mention novel parts of their analysis, even as a sketch, as compared to these existing analysis.
>
> Our algorithm indeed combines the DP-PCA method by Liu et al. with the deflation strategy from Jambulapati et al. Algorithmically, our techniques are reminiscient of these methods. However, the main contribution of our work lies in the theoretical arguments showing that this combination indeed leads to a working DP algorithm with a utility guarantee in the stochastic setting.
>
> Below we outline why this is not a straightforward result and what technical challenges we encountered.
> 1. Standard 1-PCA oracles (e.g. Liu et. al. 2021) guarantee that their (randomised) output eigenvector $u$ is close to the top eigenvector of the population covariance $\mathbb{E}[xx^\top] = \Sigma$. In deflation, one sets $P=I-u_1u_1^\top$ and runs the oracle on the residual data $Px$, hoping to approximate the top eigenvector of the residual covariance $P\Sigma P^{\top}$. However, because P itself is not only random but also correlated to the data (due to both the adaptive privacy noise and the randomness in the data used to construct $u_1$), the stochastic 1-PCA oracle only guarantees that $u_2$ is close to the top eigenvector of $\mathbb{E}[Pxx^tP]$, which is too weak for deflation (deflation needs closeness to the top eigenvector of $P\,\mathrm{E}[xx^t]\,P^\top$).
> To avoid this and apply DP to the deflation approach, we introduce the **stochastic ePCA oracle** (Definition 3), which by design (1) inputs a random projector $P$ and (2) outputs a vector $u$ guaranteeing that with high probability $u$ is close to the top-eigenvector of $P\mathbb{E}[xx^{\top}]P$.
>     * To the best of our knowledge,this is the first work to apply deflation in a DP context. We do this by introducing the stronger but sufficient concept of stochastic ePCA oracle.
>     * We prove that such an oracle with adaptive noise can yield the required adaptive utility bound. (Theorem 3)
>
> 2. Having identified what is sufficient for the problem, we next construct a DP algorithm that implements this stronger oracle. Starting from Liu et. al.’s DP-PCA (which is itself a modification of the classical Oja’s method), we introduce the following important modifications
>     * Naively feeding $Pxx^{\top}P^{\top}$ into DP-PCA fails because the gaussian noise added in PRIVMEAN (for DP) pushes the iterate outside of $\mathrm{Im}(P)$. Consequently successive outputs would not be orthogonal and deflation breaks. To fix this, we explicitly re-project back onto $\mathrm{Im}(P)$ (Step 5 of ModifiedDP‑PCA). This ensures every eigenvector remains in the correct subspace, preserving orthogonality.
>     * Even with the projection, as mentioned above, existing analyses only shows closeness to the top eigenvector of $\mathrm{E}[Pxx^{\top}P^{\top}]$, not to $P\mathrm{E}[xx^{\top}]P^{\top}$. To solve this,
>         1. we first reduce the private updates of modified DP-PCA to a non-private (projected and noisy) Oja’s algorithm on $P\tilde{A}P^{\top}$ where $\tilde{A}$ is a "noisy" version of $xx^{\top}$  (Theorem 9) and
>        2. prove a new utility bound for this projected and noisy version of Oja’s algorithm showing that its output indeed aligns with the top eigenvector of $P\mathrm{E}[xx^{\top}]P^{\top}$ (Theorem 8).
>     * Lastly we had to control the error terms resulting from a single call of ModifiedDP-PCA. To the best of our knowledge Jambulapati et al considered 1-PCA oracles whose utility guarantees do not contain constants affected by the projection matrix P. The utility guarantee of ModifiedDP-PCA (Theorem 9) however, depends on several constants originating from constraints on the data (see Assumption A). If we replace $\{ x_i x_i^{\top}\}$ the input to ModifiedDP-PCA with $\{ Px_i x_i^{\top}P\}$, we need to control how these constants change. We show that most of the constants will not incur a cumulative error (see Lemma 10, 11 and 13), except for $\kappa = \frac{\lambda_1}{\lambda_1 - \lambda_2}$ (= eigengap). The eigengap will indeed incur a cumulative error due to deflation (as at every step i we need to account for how far the top eigenvector of $P\Sigma P$ is removed from the true ith eigenvector of $\Sigma$). In Lemma 23, we characterize how to control this error. This might be of independent interest when using 1-PCA algorithms whose utility depends on the eigengap ($\kappa$) with the deflation method, as it was shown that the eigengap can play an important role in the performance of PCA algorithms.
>
> Taken together, these yield the first deflation-based DP-PCA algorithm for $k>1$ with near-optimal utility. Some of the new tools that we introduce e.g. stochastic-ePCA oracles, deflation with stochastic methods, as well as the new Oja’s analyses, may well be of independent interest outside DP.
>
> Apart from this we note that our proof strategy immediately gives us two additional insights:
> 1. We can swap in different mean/range estimation subroutines and via our Meta Theorem obtain utility guarantees for k>1 DP PCA straight away.
> 2. Our proof strategy generalizes beyond ModifiedDP-PCA to any stochastic ePCA oracle, leading to our second algorithm, k-DP-Oja (via the Meta Theorem) but can also be used for non-DP randomised PCA algorithms.
>
> > Line 198-199 : The authors mention that Jambulapati et al's ePCA oracle definition doesn't extend to the stochastic case.
>
> We thank the reviewer for bringing this to our attention and we were indeed a bit unclear. To the best of our understanding Jambulapati et al show that 1pca robust algorithms need the input dataset to contain a stable set and this stability is preserved under projections. While this stability condition i) is sufficient for robust-PCA ii) is preserved under projection and iii) (on a high level) leads to not having to worry about the dependence between $P $and $x_i$, it is unclear wether the same technique can be applied in our stochastic DP setting. We will make this clearer in our updated draft.
>
> > Regarding dependence on k:
>
> Jambulapati et al. show that by reusing the whole matrix $A = \sum_{i=1}^n a_i a_i^{\top}$ at every deflation step, we can obtain a bound independent of $k$ (without privacy). The first immediate observation is that as we protect the privacy of each $a_i a_i^{\top}$, reusing them would require adaptive composition resulting in a factor of $\sqrt{k}$. Thus, even if we were able to use their proof, using the deflation method with privacy would incur an inevitable $\sqrt{k}$ factor. Our paper incurs a factor of $k$ due to the fact that our analysis requires the projections $P_{i-1}$ at step $i$ of the deflation method to be independent of the matrices we pass at step $i$. To achieve this we split the data into $k$ equal subsets, resulting in the factor $k$ in our utility guarantee.
> Secondly, as mentioned in the answer above, their notion of stable set is uniquely suitable for the robust-PCA problem and it is not clear whether doing our DP-PCA algorithm as the inner sub-routine is also amenable to the stable set property. We think it is interesting future work to see whether we can get the $\sqrt(k)$ factor using the techniques from Jambulapati et. al. or using our analysis but with “slightly” correlated data.
>
> > Another notion of PCA convergence is the correlation k-PCA :
>
> Our notion of utility is denoted as energy PCA (ePCA) in Jambulapati et al, further they denote correlation k-PCA with cPCA (correlation PCA) and indeed show that any ePCA guarantee can be converted into a cPCA guarantee (see Lemma 12 in Jambulapati et al). However, to the best of our knowledge, a correlation k-PCA guarantee only gives an energy k-PCA in the case of $k=1$.

---

> > ### Comment · Reviewer_Ekub · 2025-08-02
> >
> > I thank the authors for their detailed explanations. I think my questions have been answered. I trust the authors will incorporate the explanation of challenges in their proof setting in the final revision. I would be happy to increase my score. I think extending Liu et al's work to k-PCA using deflation is an interesting line of work and this paper presents a good contribution.

---

### Official Review · Reviewer_e7Rs · 2025-06-30

**Clarity:** 2
**Significance:** 2
**Originality:** 1
**Rating:** 3
**Confidence:** 3

**Summary:**

The proposed stochastic k-PCA algorithm not only has differential privacy and computational efficiency, but also overcomes the following two limitations: (i) require the sample size n to scale super-linearly with dimension d, even under Gaussian assumptions on the Ai, or (ii) introduce excessive noise for DP even when the intrinsic randomness within Ai is small.

**Questions:**

1. In the experimental part, only k=2 was set to compare with other algorithms, and the advantages of the algorithm can not be demonstrated.
2. Line 230, “However, for other cases such as sub-Gaussian data we expect them to perform similarly.” This conclusion has not been verified.

**Ethical Concerns:**

["NO or VERY MINOR ethics concerns only"]

**Final Justification:**

The authors have addressed some of the concerns.

**Limitations:**

This paper extends the algorithm DP-PCA of Liu et al. [2022a] and is capable of estimating the top k eigenvectors of any k≤d. However, no new solutions were proposed to overcome the two limitations (i) and (ii).

**Quality:**

2

**Strengths And Weaknesses:**

This paper proposes the first algorithm for stochastic k-PCA that is both differentially private and computationally efficient, supports any k≤d, and achieves near-optimal error.
Weaknesses:
Liu et al. [2022a] addressed these issues for sub-Gaussian data but only for estimating the top eigenvector (k = 1) using their algorithm DP-PCA. This paper proposes a kind that can estimate the top k eigenvectors of any k≤d. For k = 1, the algorithm in this paper matches the utility guarantee of DP-PCA and improves the computational efficiency. It also provides a lower bound for the general k > 1. The essence of the paper can be regarded as an expansion of Liu et al. [2022a].

---

> ### Author Rebuttal · Authors · 2025-07-31
>
> We thank the reviewer for their comments. We address each of them below:
> > In the experimental part, only k=2 was set to compare with other algorithms, and the advantages of the algorithm can not be demonstrated.
>
> The primary goal of the paper is to provide a computationally efficient Differentially Private algorithm with theoretical utility guarantees for the problem of k>1 DP-PCA in the stochastic setting.  Despite the theoretical focus, we do appreciate the reviewers' concerns that larger results with larger d and k will boost the  contribution of our paper. Thus, during the rebuttal we repeated our experiments with larger d and k and observed the same trend as the experiments in the paper (see results below):
> ```
>              Mean Distance vs. Sample Size (k=10, d=450)
>     ┌────────────────────────────────────────────┐
> 0.98┤ •• DP-Gauss-1   ♰           ♰              │
> 0.82┤ ♰♰ DP-Gauss-2                              │
>     │ pp k-DP-Ojas                               │
> 0.66┤ cc k-DP-PCA           p                    │
> 0.50┤                                            │
>     │                                            │
> 0.34┤             •                        p     │
> 0.18┤                                            │
>     │ c       c                •                 │
> 0.03┤                     c              c    •  │
>     └┬──────────┬──────────┬─────────┬──────────┬┘
>   1995.0     12746.2    23497.5   34248.8 45000.0
> distance             sample size n
> ```
> |  Algorithm  |  $n$  |  $d$  |  $k$  |  distance  |
> |-------------|-------|-------|-------|------------|
> | DP-Gauss-1  | 20000 |  200  |   5   |   0.043    |
> | DP-Gauss-2  | 20000 |  200  |   5   |   0.138    |
> |  k-DP-Ojas  | 20000 |  200  |   5   |    0.1     |
> |  **k-DP-PCA**   | **20000** |  **200**  |   **5**   |   **0.002**    |
> |-------------|-------|-------|-------|------------|
> | DP-Gauss-1  | 40000 |  450  |  15   |   0.072    |
> | DP-Gauss-2  | 40000 |  450  |  15   |   0.933    |
> |  k-DP-Ojas  | 40000 |  450  |  15   |    0.57    |
> |  **k-DP-PCA**   | **40000** |  **450**  |  **15**   |    **0.02**    |
>
> > "However, for other cases such as sub-Gaussian data we expect them to perform similarly.” This conclusion has not been verified
>
> We also ran our experiments on gaussian-data and observed the same trends.
> ```
>              Mean Distance vs. Sample Size (k=10, d=200)
>     ┌────────────────────────────────────────────┐
> 1.00┤ •• DP-Gauss-1                              │
> 0.86┤ ♰♰ DP-Gauss-2   •  ♰          •   ♰        │
>     │ pp k-DP-Ojas                p             p│
> 0.73┤ cc k-DP-PCA                                │
> 0.60┤                                            │
>     │                                            │
> 0.46┤                                            │
> 0.33┤      c                                     │
>     │             c                              │
> 0.19┤                           c             c  │
>     └┬──────────┬──────────┬─────────┬──────────┬┘
>     100       11325      22550     33775    45000
> distance             sample size n
> ```
> |  Algorithm  |  $n$  |  $d$  |  $k$  |  distance  |
> |-------------|-------|-------|-------|------------|
> | DP-Gauss-1  | 45000 |  100  |   2   |   0.514    |
> | DP-Gauss-2  | 45000 |  100  |   2   |   0.808    |
> |  k-DP-Ojas  | 45000 |  100  |   2   |   0.477    |
> |  **k-DP-PCA**   | **45000** |  **100**  |   **2**   |   **0.057**    |
> |-------------|-------|-------|-------|------------|
> | DP-Gauss-1  | 45000 |  200  |  15   |    0.85    |
> | DP-Gauss-2  | 45000 |  200  |  15   |   0.912    |
> |  k-DP-Ojas  | 45000 |  200  |  15   |   0.878    |
> |  **k-DP-PCA**   | **45000** |  **200**  |  **15**  |    **0.29**    |
>
> We hope the above experiments resolves the reviewer’s concern regarding the performance of the algorithm on larger k and gaussian data and that the reviewer will consider increasing their score.
>
> > The essence of the paper can be regarded as an expansion of Liu et al. [2022a].
>
> While our work uses a modification of the DP-PCA method by Liu et al., we combine it with the deflation strategy (non-private version in Jambulapati et al).  Algorithmically, our techniques are reminiscient of these methods. However, the main contribution of our work lies in the theoretical arguments showing that this combination indeed leads to a working DP algorithm with a utility guarantee in the stochastic setting.
>
> Below we outline why this is not a straightforward result and what technical challenges we encountered.
> 1. Standard 1-PCA oracles (e.g. Liu et. al. 2021) guarantee that their (randomised) output eigenvector $u$ is close to the top eigenvector of the population covariance $\mathbb{E}[xx^\top] = \Sigma$. In deflation, one sets $P=I-u_1u_1^\top$ and runs the oracle on the residual data $Px$, hoping to approximate the top eigenvector of the residual covariance $P\Sigma P^{\top}$. However, because P itself is not only random but also correlated to the data (due to both the adaptive privacy noise and the randomness in the data used to construct $u_1$), the stochastic 1-PCA oracle only guarantees that $u_2$ is close to the top eigenvector of $\mathbb{E}[Pxx^tP]$, which is too weak for deflation (deflation needs closeness to the top eigenvector of $P\,\mathrm{E}[xx^t]\,P^\top$).
> To avoid this and apply DP to the deflation approach, we introduce the **stochastic ePCA oracle** (Definition 3), which by design (1) inputs a random projector $P$ and (2) outputs a vector $u$ guaranteeing that with high probability $u$ is close to the top-eigenvector of $P\mathbb{E}[xx^{\top}]P$.
>     * To the best of our knowledge, **this is the first work to apply deflation in a DP context**. We do this by introducing the stronger but sufficient concept of stochastic ePCA oracle.
>     * We prove that such an oracle with adaptive noise can yield the required adaptive utility bound. (Theorem 3)
>
> 2. Having identified what is sufficient for the problem, we next construct a DP algorithm that implements this stronger oracle. Starting from Liu et. al.’s DP-PCA (which is itself a modification of the classical Oja’s method), we introduce the following important modifications
>     * Naively feeding $Pxx^{\top}P^{\top}$ into DP-PCA fails because the gaussian noise added in PRIVMEAN (for DP) pushes the iterate outside of $\mathrm{Im}(P)$. Consequently successive outputs would not be orthogonal and deflation breaks. To fix this, we explicitly re-project back onto $\mathrm{Im}(P)$ (Step 5 of ModifiedDP‑PCA). This ensures every eigenvector remains in the correct subspace, preserving orthogonality.
>     * Even with the projection, as mentioned above, existing analyses only shows closeness to the top eigenvector of $\mathrm{E}[Pxx^{\top}P^{\top}]$, not to $P\mathrm{E}[xx^{\top}]P^{\top}$. To solve this,
>         1. we first reduce the private updates of modified DP-PCA to a non-private (projected and noisy) Oja’s algorithm on $P\tilde{A}P^{\top}$ where $\tilde{A}$ is a "noisy" version of $xx^{\top}$  (Theorem 9) and
>         2. prove a new utility bound for this projected and noisy version of Oja’s algorithm showing that its output indeed aligns with the top eigenvector of $P\mathrm{E}[xx^{\top}]P^{\top}$ (Theorem 8).
>
>     * Lastly we had to control the error terms resulting from a single call of ModifiedDP-PCA. To the best of our knowledge Jambulapati et al considered 1-PCA oracles whose utility guarantees do not contain constants affected by the projection matrix P. The utility guarantee of ModifiedDP-PCA (Theorem 9) however, depends on several constants originating from constraints on the data (see Assumption A). If we replace $\{ x_i x_i^{\top}\}$ the input to ModifiedDP-PCA with $\{ Px_i x_i^{\top}P\}$, we need to control how these constants change. We show that most of the constants will not incur a cumulative error (see Lemma 10, 11 and 13), except for $\kappa = \frac{\lambda_1}{\lambda_1 - \lambda_2}$ (= eigengap). The eigengap will indeed incur a cumulative error due to deflation (as at every step i we need to account for how far the top eigenvector of $P\Sigma P$ is removed from the true ith eigenvector of $\Sigma$). In Lemma 23, we characterize how to control this error. This might be of independent interest when using 1-PCA algorithms whose utility depends on the eigengap ($\kappa$) with the deflation method, as it was shown that the eigengap can play an important role in the performance of PCA algorithms.
> 3. Lastly we derive an information-theoretic lower bound for differentially private PCA under our setting.
>
> Taken together, these yield the first deflation-based DP-PCA algorithm for $k>1$ with near-optimal utility. Some of the new tools that we introduce e.g. stochastic-ePCA oracles, deflation with stochastic methods, as well as the new Oja’s analyses, may well be of independent interest outside DP.
> Apart from this we note that our proof strategy immediately gives us two additional insights:
> 1. We can swap in different mean/range estimation subroutines and via our Meta Theorem obtain utility guarantees for k>1 DP PCA straight away.
> 2. Our proof strategy generalizes beyond ModifiedDP-PCA to any stochastic ePCA oracle, leading to our second algorithm, k-DP-Oja (via the Meta Theorem) but can also be used for non-DP randomized PCA algorithms.

---

> > ### Comment · Reviewer_e7Rs · 2025-08-07
> > **comments for the manuscript**
> >
> > Thank the authors for their explanations. Some of my concerns have been answered. I can raise the score.

---

> > > ### Author Response · Authors · 2025-08-07
> > >
> > > We are glad that our rebuttal addressed some of the concerns of the reviewer and that the reviewer will increase their score. Please let us know if you want any further clarifications from us.

---

### Official Review · Reviewer_KnV6 · 2025-06-30

**Clarity:** 3
**Significance:** 3
**Originality:** 3
**Rating:** 4
**Confidence:** 4

**Summary:**

The paper studies the problem of k-PCA in the differential privacy settings. Prior work Liu et al.[2022a] address the problem only when k=1. Recently Jambulapati et al. [2024] provides a k-to-1-PCA reduction. This paper extended the deflation analysis of Jambulapati et al 2024 to the stochastic setting. The proposed algorithm repeatedly applies the DP-PCA subroutine in Liu et al. 2022a to recover the top eigenvector and filter out the component in the recovered space. A complementing lowerbound showed that the algorithm achieves near matching statistical error upto a factor of k.

**Questions:**

1. How is ModifiedDP-PCA different from DP-PCA in Liu et al. [2022a]? If we simply project each matrix with PAP^T and apply DP-PCA, do we have the same result and guarantee?
2. Following the first question. Line 210 on page 7 says “Note that we cannot plug in the DP-PCA algorithm...since it only guarantees relative error on E[P] \Sigma E[P]”. What does this mean? Isn’t P an input to the algorithm which is deterministic?

**Ethical Concerns:**

["NO or VERY MINOR ethics concerns only"]

**Final Justification:**

I will maintain my recommendation of Borderline Accept due to limited technical innovation of the paper.

**Limitations:**

Yes

**Quality:**

3

**Strengths And Weaknesses:**

Strengths: The paper addresses an important problem in the DP data analysis. The algorithm is simple to implement and achieves a near optimal error upto a factor of k.
Weakness: It is not clear there is significant technical novelty or contribution, at least not that I am aware of upon reading the paper. It seems to be a straightforward combination of Liu et al. [20222a] and Jambulapati et al. [2024]. The guarantee of the algorithm is off by a k factor, which might be improved to sqrt(k) if data is reused.

---

> ### Author Rebuttal · Authors · 2025-07-31
>
> We thank the reviewer for their comments. We hope our responses below answers the reviewer's concerns.
>
> > How is ModifiedDP-PCA different from DP-PCA? ... If we simply project each matrix with PAP^T and apply DP-PCA, do we have the same result and guarantee?
>
> While, we agree with the reviewer that Modified DP-PCA and DP-PCA (by Liu et al.) are algorithmically similar, they differ in step 5 of the algorithm. We highlight that by simply inputting the matrices $PAP^{\top}$ into DP-PCA, we would not obtain the same results. The additional projection ($P$) we add in step 5 of ModifiedDP-PCA is crucial, as the noise the PRIVMEAN algorithm (subroutine of DP-PCA) adds to achieve DP would lead the updates to no longer lie in Im($P$), which in turn will cause the deflation to not converge.
>
> However, even if one were to include this additional projection in Step 5, the utility guarantee in Liu et al  still does not apply. Their guarantee shows that the approximate eigenvector u is close to the top eigenvector of $\mathbb{E}[P A P]$. While this would suffice if $P$ were deterministic, in our setting $P$ is random, as it is dependent n the eigenvectors $u_j$ obtained in earlier steps of the deflation method. Due to the inherent randomness of our algorithm (due to DP) and the data-dependent nature of this randomness, the projection matrix $P =  I - \sum_{j < i} u_j u_j^{\top}$ may introduce bias. This bias can hinder convergence, which is why we require a new analysis accounting for this. Consequently, this also explains why it is not sufficient to apply an arbitrary DP algorithm within the deflation framework and we need it to satisfy a property like the stochastic ePCA oracle.
>
> > “It is not clear there is significant technical novelty or contribution”
>
> We agree with the reviewer that our algorithm indeed combines the PCA method by Liu et al. with the deflation strategy from Jambulapati et al. Algorithmically, our algorithm is similar to these two techniques. However, the main contribution of our work lies in the theoretical arguments showing that this combination leads to a working DP algorithm with a utility guarantee in the stochastic setting.
>
> Below we outline why this is not a straightforward result and what technical challenges we encountered.
> 1. Standard 1-PCA oracles (e.g. Liu et. al. 2021) guarantee that their (randomised) output eigenvector $u$ is close to the top eigenvector of the population covariance $\mathbb{E}[xx^\top] = \Sigma$. In deflation, one sets $P=I-u_1u_1^\top$ and runs the oracle on the residual data $Px$, hoping to approximate the top eigenvector of the residual covariance $P\Sigma P^{\top}$. However, because P itself is not only random but also correlated to the data (due to both the adaptive privacy noise and the randomness in the data used to construct $u_1$), the stochastic 1-PCA oracle only guarantees that $u_2$ is close to the top eigenvector of $\mathbb{E}[Pxx^tP]$, which is too weak for deflation (deflation needs closeness to the top eigenvector of $P\,\mathrm{E}[xx^t]\,P^\top$).
> To avoid this and apply DP to the deflation approach, we introduce the **stochastic ePCA oracle** (Definition 3), which by design (1) inputs a random projector $P$ and (2) outputs a vector $u$ guaranteeing that with high probability $u$ is close to the top-eigenvector of $P\mathbb{E}[xx^{\top}]P$.
>     * To the best of our knowledge, **this is the first work to apply deflation in a DP context**. We do this by introducing the stronger but sufficient concept of stochastic ePCA oracle.
>     * We prove that such an oracle with adaptive noise can yield the required adaptive utility bound. (Theorem 3)
>
> 2. Having identified what is sufficient for the problem, we next construct a DP algorithm that implements this stronger oracle. Starting from Liu et. al.’s DP-PCA (which is itself a modification of the classical Oja’s method), we introduce the following important modifications
>     * Naively feeding $Pxx^{\top}P^{\top}$ into DP-PCA fails because the gaussian noise added in PRIVMEAN (for DP) pushes the iterate outside of $\mathrm{Im}(P)$. Consequently successive outputs would not be orthogonal and deflation breaks. To fix this, we explicitly re-project back onto $\mathrm{Im}(P)$ (Step 5 of ModifiedDP‑PCA). This ensures every eigenvector remains in the correct subspace, preserving orthogonality.
>     * Even with the projection, as mentioned above, existing analyses only shows closeness to the top eigenvector of $\mathrm{E}[Pxx^{\top}P^{\top}]$, not to $P\mathrm{E}[xx^{\top}]P^{\top}$. To solve this,
>         1. we first reduce the private updates of modified DP-PCA to a non-private (projected and noisy) Oja’s algorithm on $P\tilde{A}P^{\top}$ where $\tilde{A}$ is a "noisy" version of $xx^{\top}$  (Theorem 9) and
>         2. prove a new utility bound for this projected and noisy version of Oja’s algorithm showing that its output indeed aligns with the top eigenvector of $P\mathrm{E}[xx^{\top}]P^{\top}$ (Theorem 8).
>     * Further we had to control the error terms resulting from a single call to ModifiedDP-PCA. To the best of our knowledge Jambulapati et al considered 1-PCA oracles whose utility guarantees do not contain constants affected by the projection matrix P. The utility guarantee of ModifiedDP-PCA (Theorem 9) however, depends on several constants originating from constraints on the data (see Assumption A). If we replace $\{ x_i x_i^{\top}\}$ the input to ModifiedDP-PCA with $\{ Px_i x_i^{\top}P\}$, we need to control how these constants change. We show that most of the constants will not incur a cumulative error (see Lemma 10, 11 and 13), except for $\kappa = \frac{\lambda_1}{\lambda_1 - \lambda_2}$ (= eigengap). The eigengap will indeed incur a cumulative error due to deflation (as at every step i we need to account for how far the top eigenvector of $P\Sigma P$ is removed from the true ith eigenvector of $\Sigma$). In Lemma 23, we characterize how to control this error. This might be of independent interest when using 1-PCA algorithms whose utility depends on the eigengap ($\kappa$) with the deflation method, as it was shown that the eigengap can play an important role in the performance of PCA algorithms.
> 3. Lastly we derive an **information-theoretic lower bound for differentially private PCA under our setting**.
>
> Taken together, these yield the first deflation-based DP-PCA algorithm for $k>1$ with near-optimal utility. Some of the new tools that we introduce e.g. stochastic-ePCA oracles, deflation with stochastic methods, as well as the new Oja’s analyses, may well be of independent interest outside DP.
>
> Apart from this we note that our proof strategy immediately gives us two additional insights:
> 1. We can swap in different mean/range estimation subroutines and via our Meta Theorem obtain utility guarantees for k>1 DP PCA straight away.
> 2. Our proof strategy generalizes beyond ModifiedDP-PCA to any stochastic ePCA oracle, leading to our second algorithm, k-DP-Oja (via the Meta Theorem) but can also be used for non-DP randomised PCA algorithms.

---

> > ### Comment · Reviewer_KnV6 · 2025-08-01
> >
> > "While this would suffice if $P$ were deterministic, in our setting $P$ is random, as it is dependent n the eigenvectors $u_j$ obtained in earlier steps of the deflation method." I am still quite confused by this argument. Yes P is random but given P, the distribution of A in the next round does not change, since every time new set of data is used. How does this introduce bias?

---

> ### Author Response · Authors · 2025-08-03
>
> You are right about the statistical object $P\Sigma P^t$. At step i, the projection matrix P is fixed from prior data, and the new data batch A is independent. The statistical object for this step is indeed the eigenspace of P\SigmaP^T. The issue is not one of statistical bias, and we will edit the manuscript to ensure this is clear.
>
> The main analytical challenge arises because the utility guarantee of any 1-PCA oracle is a function of its input distribution's properties (see Assumption A). Consequently, the error bound at step i depends on $P_{i-1}$, making that error bound itself a random variable. **From the perspective of an analyst proving the final utility guarantee, $P_{i-1}$ is a random variable, and they must account for all randomness up to that point**.
>
> First we establish a conditional guarantee for our subroutine ModifiedDP-PCA this is not a given from Liu et al., because of the additional projection at step 5. As mentioned in our previous answer, this required a new utility analysis, specifically we gave a reduction of ModifiedDP-PCA to a projected and noisy version of non-private Oja’s method (Theorem 9). This gives us a utility result for ModifiedDP-PCA where all the parameters in Assumption A are accounted for. While other paths to constructing a stochastic ePCA oracle from existing work might exist, they would still require a similar, non-trivial proof.
>
> Second, having established a reliable subroutine, we then analyze the entire $k$-step algorithm. This requires treating $P$ as a random variable to bound the total accumulated error (Theorem 1). The **error bound at step i is a function of the random eigengap of $P_{i-1}\Sigma P_{i-1}^T$. The core of this stage is proving that these intertwined errors do not blow up catastrophically** (see specifically Lemma 23, Lemma 10, Lemma 11, Lemma 13). Our analysis provides, to our knowledge, the first such bounds for a stochastic DP deflation method, ensuring the global stability (not just per-iteration) of the final estimate.
>
> In short, our contribution is in navigating these analytical challenges to obtain the first computationally efficient, iterative, deflation-based DP algorithm for stochastic PCA with a noise-adaptive guarantee.

---

> > ### Comment · Reviewer_KnV6 · 2025-08-03
> >
> > Thanks a lot for the clarification.

---

### Official Review · Reviewer_cqe7 · 2025-07-23

**Clarity:** 3
**Significance:** 3
**Originality:** 3
**Rating:** 5
**Confidence:** 3

**Summary:**

The paper considers the problem of DP $k$-PCA. The goal is to privately approximate the sub-space corresponding to the $k$ largest eigenvalues of a matrix that is the expected value of a distribution over random matrices. The problem has been addressed in the past in the work of Liu et al. (2022) for the case where $k = 1$, i.e., the case where we want to perform top eigenvector estimation.
The present paper gives the first algorithm for private $k$-PCA under approx-DP for which it suffices for the number of samples to scale roughly linearly with $d$ (the dimension of the data) to obtain low statistical error. A core element of the approach involves the use of the deflation framework of Jambulapati et al. (2024). That work demonstrates how, given black-box access to an oracle for top eigenvector estimation, it becomes possible to estimate the top-$k$ subspace by using the oracle iteratively. The present work combines a stochastic version of that reduction with variants of the subroutines given in Liu et al. (2022), which leads to their algorithm for $k$-PCA. This necessitates using the notion of a stochastic e-PCA oracle, and showing that the well-known Oja's algorithm satisfies the requirements of the oracle's definition. Upper bounds are given when the input is constructed from Gaussian random vectors, as well as when it comes from the spiked covariance model. Additionally, a lower bound is given for the spiked covariance model, which exhibits a gap with the upper bound in the dependence on $k$. Finally, an empirical evaluation of the method is conducted.

Arun Jambulapati, Syamantak Kumar, Jerry Li, Shourya Pandey, Ankit Pensia, and Kevin Tian. Black-box k-to-1-pca reductions: Theory and applications. arXiv:2403.03905, 2024.

Xiyang Liu, Weihao Kong, Prateek Jain, and Sewoong Oh. Dp-pca: Statistically optimal and differentially private pca. In Neural Information Processing Systems (NeurIPS), 2022.

**Questions:**

Can the authors provide a more detailed discussion and comparison with the results of Liu et al. (2022) and Jambulapati et al. (2024)?

**Ethical Concerns:**

["NO or VERY MINOR ethics concerns only"]

**Final Justification:**

I have read the authors' response carefully. It clarifies my questions about the technical challenges of the work, so I have decided to increase my score accordingly.

**Limitations:**

Yes

**Paper Formatting Concerns:**

No concerns

**Quality:**

3

**Strengths And Weaknesses:**

I think that this is a good paper, but there's a number of weaknesses too in my opinion. I am going to discuss both positive and negative aspects in detail below.

Strengths:
1) This is the first work that gives a computationally efficient private algorithm for top-$k$ PCA with roughly linear sample complexity and low error. I feel this alone gives the work significant merit.
2) The use of the deflation method from the work of Jambulapati et al. (2024) in the context of privacy was something that I found nice at a conceptual level.

Weaknesses:
1) One issue has to do with the gap in the $k$-dependence between the upper and the lower bounds. This is a weakness, but I don't think it significantly affects the value of the paper, since this is the first algorithm for DP $k$-PCA, and I think it's not damning if it is not statistically optimal.
2) By reading the main body, it was not entirely clear to me what the technical challenges were for modifying the subroutines of Liu et al. (2022), and to show that the requirements of a stochastic e-PCA oracle are satisfied. I have not read Liu et al. (2022) and Jambulapati et al. (2024), so I was hoping for a more detailed discussion, which would make the evaluation of the merits of the present work easier.

Overall, I think this is a decent paper, but I want to wait for the discussion period before expressing strong opinions about whether it should be accepted or not. I am voting borderline accept for the time being.

---

> ### Author Rebuttal · Authors · 2025-07-31
>
> We thank the reviewer for their comments. We address their perceived weaknesses below.
>
> >  Gap in the k-dependence:
>
> We acknowledge the gap and note that we incur a factor of $k$ because our analysis requires that the projections $P_{i-1}$ at step $i$ of the deflation method are independent of the matrices $A_1, …, A_n$ we pass at step $i$. Therefore we split the data into $k$ equal subsets so that each deflation step obtains a fresh independent batch of data. This ensures that at iteration $i$ of the deflation method: $P_{i-1} = I - \sum_{j < i} u_j u_j^{\top}$, is independent of the data matrices. However, this requirement might be an artifact of our proof, and if we were able to do the analysis over correlated batches, we could reuse all data samples at every deflation step and only incur a factor of $\sqrt{k}$ (due to adaptive composition). Reducing this gap in $k$ or understanding  if the $\sqrt{k}$ factor is inherently necessary for deflation methods with DP, could be an exciting direction for future work.
>
>
> > Technical contributions and challenges:
>
> Our algorithm indeed combines the DP-PCA method by Liu et al. with the deflation strategy from Jambulapati et al. Algorithmically, our techniques are reminiscient of these methods. However, the main contribution of our work lies in the theoretical arguments showing that this combination indeed leads to a working DP algorithm with a utility guarantee in the stochastic setting.
>
> Below we outline why this is not a straightforward result and what technical challenges we encountered.
> 1. Standard 1-PCA oracles (e.g. Liu et. al. 2021) guarantee that their (randomised) output eigenvector $u$ is close to the top eigenvector of the population covariance $\mathbb{E}[xx^\top] = \Sigma$. In deflation, one sets $P=I-u_1u_1^\top$ and runs the oracle on the residual data $Px$, hoping to approximate the top eigenvector of the residual covariance $P\Sigma P^{\top}$. However, because P itself is not only random but also correlated with the data (due to both the adaptive privacy noise and the randomness in the data used to construct $u_1$), the stochastic 1-PCA oracle only guarantees that $u_2$ is close to the top eigenvector of $\mathbb{E}[Pxx^tP]$, which is too weak for deflation (deflation needs closeness to the top eigenvector of $P\,\mathrm{E}[xx^t]\,P^\top$).
> To avoid this and apply DP to the deflation approach, we introduce the **stochastic ePCA oracle** (Definition 3), which by design (1) inputs a random projector $P$ and (2) outputs a vector $u$ guaranteeing that with high probability $u$ is close to the top-eigenvector of $P\mathbb{E}[xx^{\top}]P$.
>     * To the best of our knowledge,this is the first work to apply deflation in a DP context. We do this by introducing the stronger but sufficient concept of stochastic ePCA oracle.
>     * We prove that such an oracle with adaptive noise can yield the required adaptive utility bound. (Theorem 3)
>
> 2. Having identified what is sufficient for the problem, we next construct a DP algorithm that implements this stronger oracle. Starting from Liu et. al.’s DP-PCA (which is itself a modification of the classical Oja’s method), we introduce the following important modifications
>     * Naively feeding $Pxx^{\top}P^{\top}$ into DP-PCA fails because the gaussian noise added in PRIVMEAN (for DP) pushes the iterate outside of $\mathrm{Im}(P)$. Consequently successive outputs would not be orthogonal and deflation breaks. To fix this, we explicitly re-project back onto $\mathrm{Im}(P)$ (Step 5 of ModifiedDP‑PCA). This ensures every eigenvector remains in the correct subspace, preserving orthogonality.
>     * Even with the projection, as mentioned above, existing analyses only shows closeness to the top eigenvector of $\mathrm{E}[Pxx^{\top}P^{\top}]$, not to $P\mathrm{E}[xx^{\top}]P^{\top}$. To solve this,
>
>         1. we first reduce the private updates of modified DP-PCA to a non-private (projected and noisy) Oja’s algorithm on $P\tilde{A}P^{\top}$ where $\tilde{A}$ is a "noisy" version of $xx^{\top}$  (Theorem 9) and
>         2. prove a new utility bound for this projected and noisy version of Oja’s algorithm showing that its output indeed aligns with the top eigenvector of $P\mathrm{E}[xx^{\top}]P^{\top}$ (Theorem 8)
>     * Further we had to control the error terms resulting from a single call of ModifiedDP-PCA. To the best of our knowledge Jambulapati et al considered 1-PCA oracles whose utility guarantees do not contain constants affected by the projection matrix P. The utility guarantee of ModifiedDP-PCA (Theorem 9) however, depends on several constants originating from constraints on the data (see Assumption A). If we replace $\{ x_i x_i^{\top}\}$ the input to ModifiedDP-PCA with $\{ Px_i x_i^{\top}P\}$, we need to control how these constants change. We show that most of the constants will not incur a cumulative error (see Lemma 10, 11 and 13), except for $\kappa = \frac{\lambda_1}{\lambda_1 - \lambda_2}$ (= eigengap). The eigengap will indeed incur a cumulative error due to deflation (as at every step i we need to account for how far the top eigenvector of $P\Sigma P$ is removed from the true ith eigenvector of $\Sigma$). In Lemma 23, we characterize how to control this error. This might be of independent interest when using 1-PCA algorithms whose utility depends on the eigengap ($\kappa$) with the deflation method, as it was shown that the eigengap can play an important role in the performance of PCA algorithms.
> 3. Lastly we derive an **information-theoretic lower bound for differentially private PCA under our setting**.
>
> Taken together, these yield the first deflation-based DP-PCA algorithm for $k>1$ with near-optimal utility. Some of the new tools that we introduce e.g. stochastic-ePCA oracles, deflation with stochastic methods, as well as the new Oja’s analyses, may well be of independent interest outside DP.
>
> Apart from this we note that our proof strategy immediately gives us two additional insights:
> 1. We can swap in different mean/range estimation subroutines and via our Meta Theorem obtain utility guarantees for k>1 DP PCA straight away.
> 2. Our proof strategy generalizes beyond ModifiedDP-PCA to any stochastic ePCA oracle, leading to our second algorithm, k-DP-Oja (via the Meta Theorem) but can also be used for non-DP randomised PCA algorithms.
>
> We hope our answer clarifies the technical challenges in obtaining our results. We are happy to include these details in the updated draft as well as answer any questions you may have during the rebuttal period.

---

### Note · Authors · 2025-08-14

We are glad that all reviewers appreciate our contribution as the first algorithm to achieve near-optimal error—within a factor of k-in the stochastic setting for any k >= 1. In particular, we are pleased that reviewers **cqe7 and Ekub** liked the idea of using the deflation method in the context of privacy, reviewers **e7Rs and  Ekub** highlighted our lower bound,  reviewers **cGF1 and  Ekub** valued our presentation, and lastly reviewer **KnV6** appreciated that our approach is simple to implement

We are also grateful for the reviewers’ thoughtful engagement and believe the discussions will help strengthen our paper. Specifically:
1. As suggested by **Reviewer cGF1**, we will add the noisy power method as an additional baseline.
2. Based on **Reviewer Ekub's** feedback, we realized that our clarity was a bit lacking on discussing the exact technical challenges in applying the deflation method to a differential privacy context; we will address this in the revision.
3. Based on **Reviewer e7Rs’s** feedback we ran additional experiments to show that varying k and d empirically has the same effect as the main theorem suggests. We will add those experiments as additional plots to the appendix.

---

### Decision · Program_Chairs · 2025-09-17

**Decision:**

Accept (poster)

**Comment:**

Overall, the reviewers agree that this is a nice result on DP k-PCA problem, extending the work of Liu et al. [2022a] from k=1 to general k, and leveraging various techniques including the reduction in Jambulapati et al. [2024]. The concerns about the novelty of the paper compared with Liu et al. [20222a] and Jambulapati et al. [2024] were addressed in the rebuttal, but the authors are expected to add these discussions to the main paper. Also, the additional experimental results provided in the rebuttal are expected to be added to the camera ready.